# A Implies B: Circuit Analysis in LLMs for Propositional Logical Reasoning

**Guan Zhe Hong**[*1]    **Nishanth Dikkala**[2]    **Enming Luo**[2]    **Cyrus Rashtchian**[2]
**Xin Wang**[2]    **Rina Panigrahy**[2]
[1]Purdue University    [2] Google Research
hong288@purdue.edu,
{nishanthd, enming, cyroid, wanxin, rinap}@google.com

## Abstract

Due to the size and complexity of modern large language models (LLMs), it has proven challenging to uncover the underlying mechanisms that models use to solve reasoning problems. For instance, is their reasoning for a specific problem localized to certain parts of the network? Do they break down the reasoning problem into modular components that are then executed as sequential steps as we go deeper in the model? To better understand the reasoning capability of LLMs, we study a minimal propositional logic problem that requires combining multiple facts to arrive at a solution. By studying this problem on Mistral and Gemma models, up to 27B parameters, we illuminate the core components the models use to solve such logic problems. From a mechanistic interpretability point of view, we use causal mediation analysis to uncover the pathways and components of the LLMs' reasoning processes. Then, we offer fine-grained insights into the functions of attention heads in different layers. We not only find a sparse circuit that computes the answer, but we decompose it into sub-circuits that have four distinct and modular uses. Finally, we reveal that three distinct models – Mistral-7B, Gemma-2-9B and Gemma-2-27B – contain analogous but not identical mechanisms.

## 1 Introduction

LLMs can solve many tasks in a few-shot manner (Brown et al., 2020; Radford et al., 2019b). One emergent ability of these models is solving problems that require planning and reasoning. In particular, LLMs seem to have internal components that judiciously process and transform the provided information. Unfortunately, it is not clear whether these models implement step-by-step algorithms or whether they perform opaque computations to arrive at their answers.

The area of circuit analysis has emerged as a way to understand how transformers use their internal components. While the definition of a circuit varies across different works, in this paper, our definition follows Wang et al. (2023a). A *circuit* within a transformer is a collection of model components (attention heads, neurons, etc.) with the *edges* in the circuit indicating the information flow between the components in the forward pass. Circuit analysis can inform how to improve models, how to debug errors, and how to explain patterns in their behavior. Despite the importance of these goals, there has been a limited number of large-scale studies of interpreting LLM reasoning. Within mechanistic interpretability (Geva et al., 2021; Vig et al., 2020; Hou et al., 2023), existing studies either only provide partial evidence for the underlying circuits (Meng et al., 2022) or are limited to small models like a 3-layer transformer or GPT-2 sized models (Merullo et al., 2024a; Rauker et al., 2023; Wang et al., 2023a). Our goal is to go beyond this, uncovering the distinct roles of different

---

[*]Part of this work was done as a student researcher at Google Research.

sets of components for solving a canonical propositional reasoning problem in frontier models with as many as 27B parameters. Specifically, we aim to identify separate sub-circuits for distinct steps like retrieving information from the context, processing this information, and arriving at an answer.[2]

## 1.1 A Suitable Set of Propositional Logical Reasoning Problems

Performing circuit analysis for LLMs requires a careful selection of what problem to study. If it is too simple (e.g., just copying part of the input), then we will learn little about reasoning. If it is too complex (e.g., generic math word problems), then we will have no way to systematically apply causal analysis and we will struggle to localize and delineate the model's internal components. The problem also needs to be sufficiently natural, such that it is a common subproblem of more complex reasoning problems in the wild, such as those found in natural logic (MacCartney & Manning, 2014).

Keeping the above considerations in mind, we study the following propositional logic template, which requires reasoning over distinct parts of the input. The problem involves five boolean variables. Given two propositions as "Rules" and truth values of three variables as "Facts" we wish to infer the unknown truth value of a different variable (which we call the "query"). For concreteness, here are two possible instantiations:

| | |
|---|---|
| **Rules**: $A$ or $B$ implies $C$. $D$ implies $E$.
**Facts**: $A$ is true. $B$ is false. $D$ is true.
**Question**: what is the truth value of $C$? | **Rules**: $V$ and $W$ implies $X$. $Y$ implies $Z$.
**Facts**: $V$ is false. $W$ is true. $Y$ is false.
**Question**: what is the truth value of $Z$? |

Consider the left problem, which queries the LogOp (disjunction) chain. One concise solution to the question is "A is true. A or B implies C; C is true." This problem, while simple-looking on the surface, requires the model to perform actions that are essential to more complex reasoning problems. Even writing down the first answer token (A in "A is true") takes multiple steps.[3] The model must resolve the ambiguity of which *rule* is being queried. In this case, it is "A or B implies C" not "D implies E". Then, the model needs to determine which *facts* are relevant and to process the information "A is true" and "B is false". Finally, it decides that "A is true" is relevant and implies that "C is true" due to the nature of disjunction. In contrast, the problem on the right queries the linear chain, requiring the model to locate different rules ($Y$ implies $Z$) and process the facts ($Y$ is false).

Our logic problem is also inspired by some of the ones in reasoning benchmarks such as GSM8k (Cobbe et al., 2021) or ProofWriter (Tafjord et al., 2021). The issue with using existing benchmarks directly is that there is too much syntactic and semantic variation in the questions. This makes it challenging to use the tools of mechanistic interpretability, which require changing parts of the input in a systematic way to minimize confounding factors. One can amplify the difficulty of our problem by increasing the number of variables or the length of the reasoning chain. However, even on this simple problem, Mistral and Gemma models only achieve 70% to 86% accuracy in writing the correct proof and determining the query value, even with few-shot prompting.

## 1.2 Interpreting How LLMs Solve the Logic Problems

Given the above problem template, we design systematic causality-based experiments to understand how LLMs solve it. We aim to localize and delineate circuits, as a way to interpret how contemporary LLMs solve propositional logic problems (e.g., how they determine relevant parts of the input, how they move information around, how they process facts, and how they make decisions). Finding an interpretable circuit amounts to identifying a small number of components that map to natural operations and together implement an intuitive reasoning algorithm. It is important to note that, a priori, a non-interpretable outcome is also possible, where the LLM internally merges steps in reasoning or distributes the steps across all its parameters. One of our contributions is strong evidence that the mechanisms for LLMs to solve such reasoning problems are indeed modular and localized.

Figure 1 shows the main ideas of our approach, including the data model, the interpretability tools, and some findings that are shared across all three models: Gemma-2-27B, Gemma-2-9B (Gemma Team et al., 2024) and Mistral-7B-v0.1 (Jiang et al., 2023). Our main findings are:

---

[2]Our code is available at `https://github.com/guanzhehong/prop-logic-transformer-circuit`
[3]Each word (or capital letter representing a boolean variable) gets tokenized as an individual token.

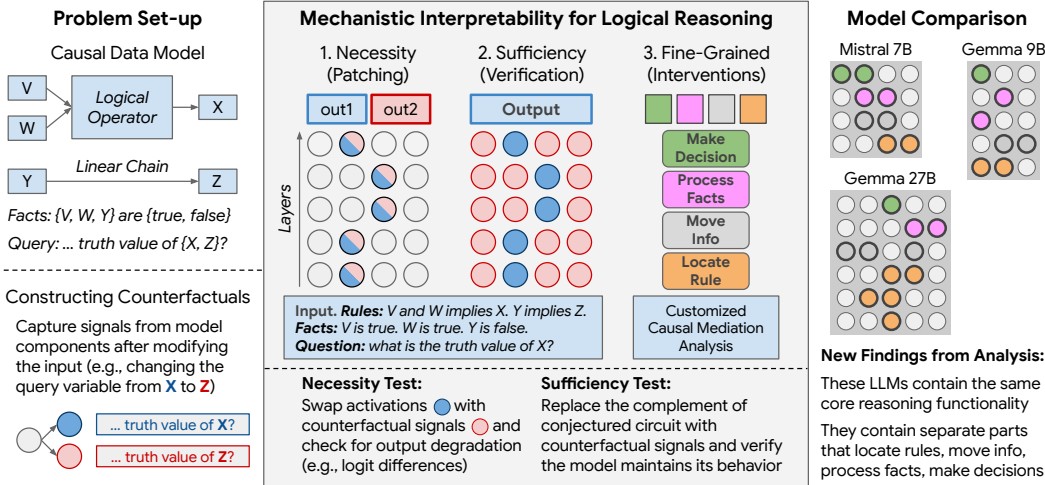

Figure 1: **Left: Problem set-up.** Top left shows the data model. The causal structure has two chains: one with a logical operator (LogOp, "and" or "or") and the other with a linear causal chain. Bottom left shows how we get counterfactual signals for the model components (e.g., attention heads). We vary parts of the input, such as the query variable, truth values, and order of rules. **Middle: Key steps in our interpretability approach.** We use causal mediation analysis to localize components in LLMs that are used to solve the logic problem (circles represent attention heads, with layers stacked vertically). We derive evidence about the functionality of components using multiple methods. For necessity, we patch with counterfactual signals (represented by blue and red nodes). Then we check the impact of each component by looking at the output of the network (e.g., checking if it outputs very different distributions "out1" vs. "out2" after patching). For sufficiency, we replace the complement of the circuit with counterfactuals, and we verify that the output does not change too much. For fine-grained analysis, we use key-value-query interventions to identify the role each component plays. **Right: New findings.** Three LLMs use similar reasoning steps when solving these problems. They use four sparse and distinct sets of attention heads in a step-by-step fashion to execute rule locating, rule moving, fact processing, and decision making (depicted by the colored nodes).

- All three models contain **four** families of attention heads with specialized roles in solving different steps in the problem: queried-rule locating heads, queried-rule mover heads, fact-processing heads, and decision heads (§3). This is surprising, as it indicates the model solves the logic problem in a sequential manner, without merging the steps. We further verify the roles of these four families by analyzing their behavior separately by patching after various types of counterfactual prompts, e.g., swapping rule locations or fact values (§3.3).

- Intriguingly, the three LLMs all perform a type of *lazy* reasoning. The core circuit does not immediately pre-process the Rules and Facts in the input. Rather, it primarily activates after seeing the Question and the ultimate query variable (§3). Moreover, the LLMs tend to reuse some of their sub-circuits for different parts of the argument when possible, consistent with Merullo et al. (2024a). We verify this for Gemma-2-9B and Mistral-7B (§3.4).

- We provide a partial analysis of Gemma-2-27B, which is somewhat limited due to computational contraints (§3.4). While all three models contain these four families components, Gemma-2-27B also possesses certain logical-operator heads that do not seem to exist or have strong causal roles in the 9B version. Gemma-2-27B's circuit appears to be slightly more parallel than the 9B version, which is more sequential. This adds nuances to a study showing that circuits are consistent across scale by Tigges et al. (2024). Indeed, from Gemma-2-9B to Gemma-2-27B, while the core algorithm appears to stay the same, there are mechanical differences and additional functional components which likely contribute to the larger model's higher proof accuracy.

- Finally, we contrast pre-trained LLMs with 3-layer models trained on our task. The small models perform intermingled reasoning, linking together their attention blocks and residual streams in subtle ways. These small models have nearly perfect accuracy, but their reasoning mechanisms are less modular, as the functional roles of model's internal components tend

to be situation-dependent (§3.5). In contrast, Mistral and Gemma seem to consistently use specialized components for different parts of the task.

Our analysis is, to our knowledge, the first to characterize the circuits employed by LLMs *in the wild* for solving a logic problem that involves distracting clauses and requires multi-step reasoning. While this reasoning is limited compared to the abilities of today's *thinking* models (Guo et al., 2025a; Arcuschin et al., 2025), we still go beyond single-hop tasks like token copying or addition from recent studies, e.g., Feucht et al. (2025); Hu et al. (2025). As another outcome, we have uncovered fundamental differences between small (3-layer), medium (7B/9B), and large (27B) models, indicating only some mechanistic insights generalize across scales.

## 2    Preliminaries and Methodology

We explain our data model and then give an overview of the methodology we use for our analysis.

**Input Prompts.** We query pre-trained LLMs in a few-shot manner on propositional logic problems defined in the introduction. We show 4 or 6 examples of questions and their minimal proofs (see Appendix B.1 for details). Then, we append a new problem that asks for the truth value of one variable, in the "Question" part. We refer to this variable as the QUERY token. A *minimal* reasoning chain consists of invoking the relevant fact(s) and rule to answer the query. Given the in-context examples, the models we consider always output a proof followed by the value of QUERY from {true, false, undetermined}. Interestingly, for the disjunction problem (e.g., invoking $A$ or $B$ implies $C$) when the model makes errors, it always starts with an incorrect *first* token (e.g., saying $B$ is true, where relevant fact is that $A$ is true). Hence, we can analyze the model's components when predicting the first token. In other words, to produce the first answer token in the minimal proof, the model must execute the causal chain "QUERY→Relevant Rule→Relevant fact(s)→Decision" over the context.

**Causal Mediation Analysis.** Following prior work, we primarily rely on Causal Mediation Analysis (CMA) (Pearl, 2001) to characterize the reasoning circuits. In our setting, CMA is primarily concerned with measuring the (natural) direct and indirect effects (DE and IE) of a model component. Consider the following classic causal diagram of CMA, in Figure 2.

Suppose we wish to understand whether a certain mediator $M$ plays an important role in the causal path from the input $X$ to the outcome $Y$. We decompose the "total effect" of $X$ on $Y$ into the sum of *direct* and *indirect* effects, as shown in the figure. The indirect effect (IE) measures how important a role the mediator $M$ plays in the causal path $X \to Y$. To measure it, we compute $Y$ given $X$, except that we artificially hold $M$'s output to its "corrupted" version (called an intervention), which is obtained by computing $M$ on a counterfactual ("corrupted") version of the input. A significant change in $Y$ indicates a strong IE, which implies that $M$ is important in the causal path. On the other hand, a weak IE (so a strong DE) implies that $M$ is unimportant.

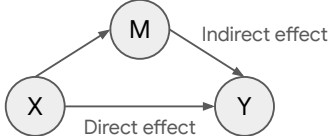

Figure 2: CMA causal graph.

**Using CMA for Circuit discovery in LLMs – Necessity and Sufficiency.** We can apply the above CMA methodology to identify the reasoning circuits in LLMs. We look at both the proof and the final answer output by the model. We perform our interpretability analysis before the model has produced the first token. We believe the first token demands the greatest number of latent reasoning steps by the model, so it is the most interesting place to examine. Consider the conjunction problem in §2. The first step in our analysis is the construction of a 'counterfactual' prompt given a base prompt (we refer to the base prompt as the normal prompt in the rest of the paper). For the first answer token, when we flip the QUERY token from C to E, the LLM must execute a fully different causal chain, from "*C→A and B implies C→A is true, B is false→B*" to "*E→D implies E(→D is true)→D*". In other words, flipping the QUERY token alone generates a "counterfactual" prompt which will help reveal the how the LLM latently executes the causal chain of "QUERY→Relevant rule→Relevant fact(s)→Decision". The outcome $Y$ to measure is also naturally defined here: the softmax layer's logit difference between the two possible answers, $B$ and $D$. Therefore, by generating normal-counterfactual prompt pairs in this fashion, LLM components with strong IEs, i.e., those pushing the LLM from the normal to the counterfactual answer, must belong to the reasoning circuit. To verify the sufficiency of the circuit, we just need to set the *complement* of the circuit as the mediator, and show that the DE is strong.

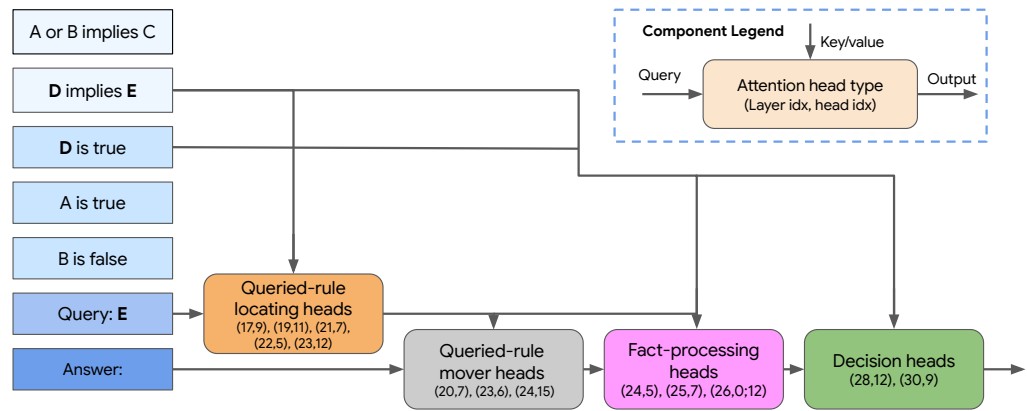

Figure 3: Gemma-2-9B's reasoning circuit for the first answer token: The (chunks of) input tokens are on the left, which are passed into the residual stream and processed by the attention heads. We use $(\ell, h)$ to denote an attention head. When referencing multiple heads in the same layer, we write $(\ell, h_1; h_2; ...; h_n)$ for brevity. We illustrate the information flow manipulated by the different types of attention heads we identified to be vital to the reasoning task.

**Fine-grained analysis: Understanding sub-component functionality**. After obtaining the full circuit, we can theorize about the role each component in the circuit plays in the reasoning chain. However, we need to verify such interpretations with empirical causal evidence. To do so, we again perform CMA with specific sub-components of the circuit held as the mediator, and we measure their IEs. For instance, if we hypothesize that an attention head uses the QUERY token to locate the rule being queried (the first reasoning step), then we perform two experiments. First, we set the mediator as the key activations of this head in the Rule section, and generate the counterfactual prompt by swapping the *position* of the rules, without touching anything else in the prompt. We should then observe a strong IE. Second, we generate the counterfactual prompt by flipping QUERY. Then, the query activation of this head, when held as the mediator, should have strong IE. In general, testing a hypothesis of a reasoning circuit component's role requires careful control on where the input prompt is "corrupted", where we need control over how the answer changes or does not change.

For the model to solve our logic problem, it has to perform multiple steps of reasoning, going from the Question to the Rules to writing down the proof. This leads to a more complicated process for systematically investigating the internal mechanisms in the LLMs. Other tools commonly used in mechanistic interpretability such as activation patching, causal tracing (Vig et al., 2020; Meng et al., 2022; Hase et al., 2024; Heimersheim & Nanda, 2024; Zhang & Nanda, 2024) are types of CMA.

# 3 Discovering Modular Reasoning Circuits in LLMs

To make the problem easier for the smaller models, we only examine Gemma-2-9B and Mistral-7B on the disjunction version (the LogOp is "or"); but, we study Gemma-2-27B on the full problem. For the "or" problem, Gemma-2-9B and Mistral-7B have proof accuracy around 83% and 70% respectively. Their accuracy with "and" is lower, while Gemma-2-27B has accuracy of 86% on the full problem.

We focus on Gemma-2-9B in the main text, which illustrates our analysis process. This example elucidates the common procedure we apply to the other models as well. Appendix B contains the full results for Gemma-2-9B, Mistral-7B and Gemma-2-27B models.

## 3.1 Discovering the Necessary Circuit: QUERY-based Search for Model Components

We discover the core reasoning circuit by performing QUERY-based intervention experiments. After performing attention head output patching and measuring logit differences, we locate a small set of attention heads that are central to the reasoning circuit of the LLM. The MLPs have much lower intervened logit difference (mostly $< 0.1$) showing they play a limited role in the reasoning process. However, we observe MLP-0 has a higher intervened logit difference. Prior work has observed that MLP-0 acts more as a "nonlinear token embedding" than a complex high-level processing unit (Wang et al., 2023a). Hence, in the rest of this section, we focus our analysis on the attention heads.

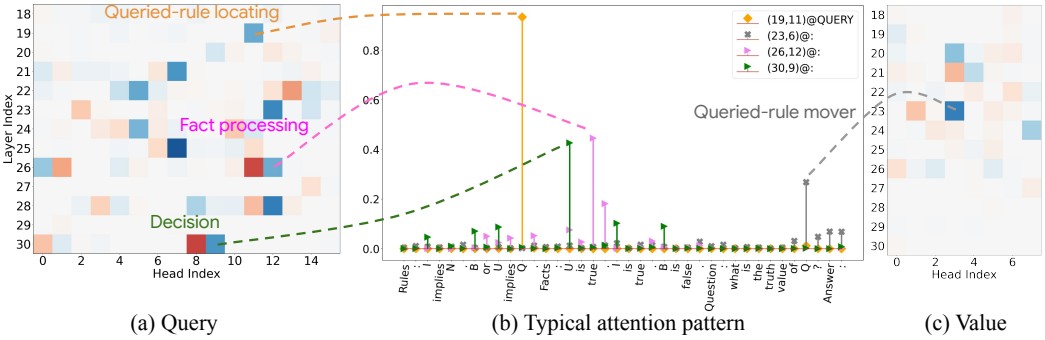

|     |     |     |
| --- | --- | --- |
| (a) Query | (b) Typical attention pattern | (c) Value |

Figure 4: Patching of query and value activations in (a) and (c). We show in (b) the typical attention patterns of a representative set of the attention heads, which are important in the intervention experiments shown in (a) and (c). We make several observations from (b). Queried-rule locating head (19,11): observe that it correctly locates the queried rule that ends with **Q**. Queried-rule mover head (23,6): it primarily focuses on the QUERY token **Q**, with some small amount of attention on the tokens following it. Fact processing head (26,12): attention concentrates in the fact section. Decision head (30,9): attention focused on the correct first answer token **U**. In this experiment, intervening on "key" activations yields trivial scores, so we focus on "value" and "query.".

*Remark.* Unless otherwise specified, we adopt a calibrated version of the logit difference (see Appendix B): the closer to 1, the stronger the indirect effect of that component is in the causal chain, and the greater a role that component plays in the reasoning circuit.

**Attention-head sub-component patching (QUERY-based patching).** We now aim to understand why the attention heads identified in the last sub-section are important. For now, we continue with altering the QUERY in the prompt as our causal intervention. We intervene on the sub-components of each attention head, namely their value, key, and query, and we examine details of their attention weights. We find that there are four types of attention heads. We show the results in Figure 4.

Queried-rule locating head. Attention head (19,11)'s *query* activation has a large intervened logit difference according to Figure 4(a). Therefore, its query (and attention patterns) are QUERY-dependent and have strong IE. Furthermore, at the QUERY position, we find that *on average*, its attention weight is above 85% at the "conclusion" variable of the rule being queried. It is responsible for *locating* the queried rule, and storing that rule's information at the QUERY position.[4]

Queried-rule mover head. Attention head (23,6)'s *value* activations have large intervened logit difference, and intriguingly, its query and key activations do *not* share that tendency. This already suggests that its attention pattern performs a fixed action on both the original and altered prompts, and only the value information is sensitive to QUERY. Furthermore, within the relevant context (excluding the 4 in-context examples given), (23,6) assigns around 50% attention weight to the QUERY position, and its attention weight at QUERY is about 5 times larger than the second largest one on average. Recalling the role of the shallower layers, we find evidence that (23,6) moves the QUERY and queried-rule information to the ":" position.[5]

Fact processing heads. Attention heads (26,12)'s *query* activations have large intervened logit differences. Within the relevant context, at the ":" token position, it attention weight on the sentence stating the correct fact is around 60% on average.[6]

Decision head. Attention heads (28,12) and (30,9)'s query activations have large intervened logit differences. Their attention patterns suggest that they are "decision" heads. Within the relevant context, each head's attention weight focuses on the correct Fact variable (when the model is correct). The single token position occupies about than 80% of its total attention in the relevant context on average. In other words, it locates the correct answer token.

---

[4](17,9), (21,0), (21,7), (22,5), (23,12) exhibit similar tendencies.

[5](20,7), (24,15) also belong to this type.

[6](24,5), (25,7), (26,0) also belong to this type.

We delay detailed inspection and visualization of the attention statistics to Appendix B.4.2. Figure 3 illustrates the reasoning circuit we identify. We remark on two intriguing properties:

1. Compared to the attention blocks, the MLPs are relatively unimportant to correct prediction.
2. There is a sparse set of attention heads that are found to be central to the reasoning circuit: the queried-rule locating heads, queried-rule mover heads, fact-processing heads, and decision heads. We discuss circuit discovery in §3.1, and verification in §3.2. This indicates that a very small fraction of the 9B parameters are taking part in solving a specific question.

## 3.2 Verifying Sufficiency of the Discovered Circuit

A natural question now arises: is $\mathcal{C}$ *sufficient* to explain the (QUERY-sensitive) reasoning actions of the LLM? We prove sufficiency by measuring the direct effect of the input prompt on the difference in the final logits, with the complement of the identified circuit treated as the mediator (defined in §2). More concretely, we run the model on the normal prompts and only allow normal information flow in every attention head in the circuit $\mathcal{C} = \{(19, 11), (23, 6), (26, 12), (30, 9), ...\}$ (on or after QUERY), while freezing all the other attention heads to their *counterfactual* activations obtained by running on the counterfactual prompts. We expect the average *circuit-intervened* logit difference $\Delta^{\mathcal{C}}_{normal}$ approaching or surpassing the average logit difference of the (un-intervened) model run on the normal prompts, $\Delta_{normal}$. We confirm this hypothesis in Table 1.

| $\mathcal{C}^{\dagger}$ | $\mathcal{C}_{null}$ | $\mathcal{C}$ | $\mathcal{C} - QRLH$ | $\mathcal{C} - QRMH$ | $\mathcal{C} - FPH$ | $\mathcal{C} - DH$ |
|---|---|---|---|---|---|---|
| $\Delta^{\mathcal{C}^{\dagger}}_{normal}/\Delta_{normal}$ | -1.0 | **0.94** | -0.97 | -0.40 | 0.17 | -1.11 |

Table 1: $\Delta^{\mathcal{C}^{\dagger}}_{normal}/\Delta_{normal}$ for Gemma-2-9B, with different choices of $\mathcal{C}^{\dagger}$. $\mathcal{C}_{null}$ denotes the *empty* circuit, i.e. the case where no intervention is performed. We abbreviate the attention head families: $QRLH$ = queried-rule locating heads, $QRMH$ = queried-rule mover heads, $FPH$ = fact-processing heads, $DH$ = decision heads; $\mathcal{C} - DH$ = full circuit but with the decision heads removed.

Including all 13 attention heads in $\mathcal{C}$, $\Delta^{\mathcal{C}}_{normal}$ is about 94% of $\Delta_{normal}$ on the normal samples. Removing any one of the four families of attention heads from $\mathcal{C}$ renders the direct effect almost trivial. Therefore, every head family is critical to the circuit. Additionally, for Mistral-7B's circuit which we obtained via the same search process, we were able to recover about 98% of the average logit-difference, indicating transferability of the procedure; please see B.5 for more details.

Please refer to Appendix B.8.3 for deeper discussions of the experimental procedure and nuances of the results, including different versions of the circuit which we performed sufficiency tests on for Gemma-2-9B, the complexities of this verification process, and how we consider certain aspects of circuit verification a major open problem in mechanistic studies of LLM reasoning.

## 3.3 Additional Evidence: More Fine-Grained Circuit Analysis

In this sub-section, we discuss example experiments of how we further verify the functionalities of the attention head families. We present the full analysis in Appendix B.8.

**Queried-rule locating heads**. We use the queried-rule locator heads, the "first step" in the reasoning circuit, to demonstrate how we perform finer "causal surgery" on the LLM to understand sub-components of the full circuit better.

First, based on our full-circuit CMA experiments, we have formed an interpretation of the queried-rule locator heads: they rely on the QUERY information to locate the queried rule in the Rules section. This is further corroborated by the attention statistics of these attention heads, shown in Figure 5(b)(i): these attention heads place significant amount of attention weight on the correct position of the queried rule.

To further verify the "look-up" functionality, we create counterfactual prompts by swapping the location of the rules while keeping everything else untouched in the normal prompt. For instance, we would have the following normal-counterfactual pairs:

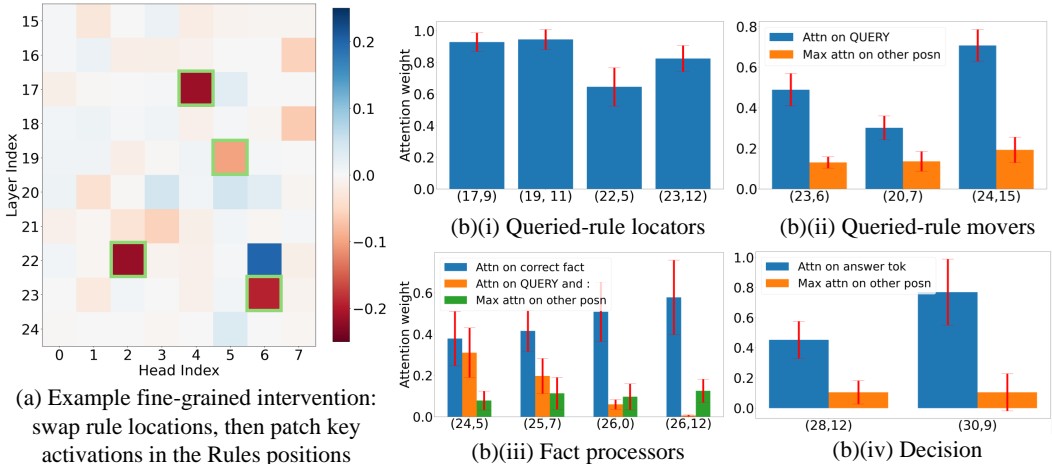

Figure 5: Finer-grained examination of circuit components in the Gemma-2-9B model. (a) shows the (zoomed-in) result of a finer-grained activation patching experiment, aimed at providing further causal evidence of the rule-locating heads' functionalities. (b)(i) to (iv) show the attention statistics of the core attention head families in the circuit.

| **Rules**: $A$ or $B$ implies $C$. $D$ implies $E$.
**Facts**: $A$ is true. $B$ is false. $D$ is true.
**Question**: what is the truth value of $C$? | $\xrightarrow[\text{prompt}]{\text{altered}}$ | **Rules**: $D$ implies $E$. $A$ or $B$ implies $C$.
**Facts**: $A$ is true. $B$ is false. $D$ is true.
**Question**: what is the truth value of $C$? |
|---|---|---|

Clearly, this rule location swap does not cause any change to the answer (here, $C$ is true in both versions). However, the queried-rule locator heads' functionality suggests that they heavily rely on the key activations (in the Rules section) to perform their role correctly. To provide further causal evidence for this interpretation, we should set the *mediator* as *the key activations of the attention heads in the Rules section*, and observe their indirect effects on the model's output logits, namely the drop in the logit difference between the answer of the queried rule and that of the alternative rule.

As expected, heads with the largest IE are the queried-rule locator heads, as shown in Figure 5(a). The keys with strong indirect effects are (17,4), (19,5), (22,2) and (23,6). By noting that Gemma-2-9B uses Grouped Query Attention which results in key and value activations shared per two heads, we see that keys with strong indirect effects indeed correspond to the attention heads (17,9), (19,11), (22,5) and (23,12). In addition, we perform CMA experiments where we swap the Fact value assignments. The fact-processing heads indeed exhibit the strongest IE.

A possible limitation in generalizing this specific "causal surgery" to contexts with a large number of clauses is that, how distant the two clauses we interchange positions with becomes a possible *confounding* factor. Careful control on how distant the two interchanged clauses are and their absolute positions in the context is then required.

**Attention statistics**. We also show the weight of four attention head families in Figure 5(b)(i) to (iv). The main takeaway is correlational evidence showing that heads attend to the expected part of the input (blue bars), which varies by type of head, rather than other positions.

## 3.4 The Reasoning Circuit in Gemma-2-27B

We summarize a few interesting observations about Gemma-2-27B here (Appendix B.9 has details).

**Mechanical circuit differences between 27B and 9B models**. Even though the 27B model also possesses attention heads which can be roughly divided into the 4 families as in the case of the 9B model, there still are certain finer differences between their circuits: the 27B model's circuit appears to have a greater degree of parallelism. In particular, a number of its fact-processing heads have direct effects on the model's output: they tend to bypass the "decision" heads. On the other hand, the 9B model's circuit is more sequential: the fact processing heads are mostly "connected" to the decision heads and exhibit weak direct effects on the model's logits.

**Logical-operator heads**. As the 27B model is required to solve the full version of the problem, on top of the four generic families of attention heads, we find that there are certain "logical-operator" heads which pay particular attention to the words "and" and "or" in the rules, and exhibit strong causal influence on the model's reasoning actions. In particular, if we generate counterfactual prompts by flipping the logical operator from "and" to "or" (or vice versa) while keeping everything else the same as the normal prompt, then we find these logical-operator heads to have strong indirect effects. This is further discussed in Appendix B.9.

### 3.5 Further Mechanistic Observations

**Subcircuit reuse**. Both Gemma-2-9B and Mistral-7B reuse their queried-rule locator heads and decision heads. Specifically, when the model needs to invoke a relevant rule (from the input) later to construct their output proof, it uses the same heads (but other input positions) to retrieve the location of the rule. Similarly, the decision heads may perform their function multiple times. In other words, this pair of components serve as the first and last reasoning step in *both* places of the model's proof. Please refer to Appendix B.8.6 for details.

**Prompt-format variation**. To provide insights on how generalizable the sub-circuits we found are, we perform CMA on Gemma-2-9B on our logic problem, but with the prompt-format changed to "Facts: ... Rules: ... Question: ... Answer: ..." (Rules and Facts sections are swapped in position). The queried-rule locators and decision heads remain almost the same in the discovered circuit as before. The mover and fact-processing heads experienced some changes, however. For example, some of the fact-processing heads focus both on relevant premise variables in the Rules section and facts in the Facts section now. This indicates that the intermediate heads are more "fluid" in their functional roles in the circuit. We discuss these results in detail in Appendix B.8.7.

**Contrast against small transformers**. To contrast against LLMs which are highly overparameterized and trained on a large variety of data, we also conducted a mechanistic analysis of a small GPT-like transformer trained from scratch on a slightly more complex version of our logic problem. We present the analysis of the 3-layer 3-head transformer in Appendix C and D, with the high-level reasoning strategy employed by the model illustrated in Figure 30; it is the smallest model capable of achieving 100% test accuracy on the problems.

The most pronounced difference we found is the less interpretable and modular nature of the attention heads in the small transformer. We show that, when a linear chain is queried, the second layer attention heads already arrive at the answer and the third layer stays "dormant", but when a logical-operator-chain is queried, layer-3 attention heads need to actively participate to resolve the right answer. In other words, the small model does not have specialized "decision heads" for all situations in the problem, unlike the LLMs. Furthermore, when the logical-operator chain is queried, even though the final-layer attention heads are proven to be the ones resolving the correct answer, their attention patterns are unstructured, scattered across token positions in the Rules and Facts sections. These observations suggest a (relative) lack of modularity and interpretability in the small model.

## 4 Related Work

**Mechanistic Interpretability**. This area explores the hidden mechanisms that enable language-modeling capabilities and other underlying phenomena from LLMs (Olsson et al., 2022; Wang et al., 2023b; Feng & Steinhardt, 2024; Wu et al., 2023; Hanna et al., 2024; McGrath et al., 2023; Singh et al., 2024; Geiger et al., 2024; Feng et al., 2024; Marks et al., 2024; Engels et al., 2024; Csordás et al., 2025; Baroni et al., 2025; Merullo et al., 2024b; Yin & Wang, 2025; Lindsey et al., 2025). Some works show how models retrieve knowledge and facts (Ferrando et al., 2025; Geva et al., 2023; Sun et al., 2025; Yao et al., 2024; Wang et al., 2024b; Lu et al., 2025), encode concepts (Todd et al., 2024; Yin & Steinhardt, 2025; Hong et al., 2025; Beaglehole et al., 2025; Li et al., 2025c) or employ certain features in CoT reasoning (Dutta et al., 2024; Troitskii et al., 2025; Wang et al., 2025a; Venhoff et al., 2025b; Chen et al., 2025; Venhoff et al., 2025a; Baek & Tegmark, 2025; Li et al., 2025b; Huang et al., 2025; Dai et al., 2025; Lee et al., 2025). These works primarily provide causal evidence of their claims through variants of *causal mediation analysis* (Vig et al., 2020; Meng et al., 2022; Hase et al., 2024; Heimersheim & Nanda, 2024; Zhang & Nanda, 2024; Mueller et al., 2025; Geiger et al., 2025). Our paper is in the sub-area of *circuit-based* mechanistic interpretability, e.g., (Elhage et al., 2021; Wang et al., 2023a). The novelty of our work is that we study larger LLMs on a compact logic

problem requiring hidden intermediate reasoning steps, which affords the analysis of how retrieval, processing, *and* decision components operate in conjunction, distinct from aforementioned works which identify certain steering features for CoT reasoning.

**Coarse-Grained Evaluation of Reasoning in LLMs**. There has been a flurry of work on training and evaluating reasoning LLMs (Muennighoff et al., 2025; Ye et al., 2025; Wang et al., 2025c; Sinha et al., 2025; Zhang et al., 2025; Shojaee et al., 2025). This extends the work on evaluating the reasoning abilities of LLMs across different tasks (Xue et al., 2024; Tafjord et al., 2021; Hendrycks et al., 2021; Chen et al., 2024; Patel et al., 2024; Berglund et al., 2024; Morishita et al., 2023; Liu et al., 2023; Fu et al., 2024; Seals & Shalin, 2024; Joren et al., 2025; Zhang et al., 2024, 2023; Saparov & He, 2023; Saparov et al., 2024; Sun et al., 2023; Luo et al., 2024; Shah et al., 2024; Arora et al., 2024; Han et al., 2024; Dziri et al., 2024; Yang et al., 2024; Loo et al., 2026; Li et al., 2025a). While these studies primarily benchmark their performance on sophisticated tasks, our work focuses on understanding "how" transformers reason on logic problems accessible to fine-grained analysis (e.g., mechanistic circuit analysis).

**Fine-Grained Analysis of How LLMs Reason**. There are fewer studies that provide fine-grained analysis of *how* pretrained LLMs reason *latently*. We build on ideas from mechanistic interpretability for arithmetic (Stolfo et al., 2023; Yu & Ananiadou, 2024; Kantamneni & Tegmark, 2025; Zhou et al., 2024; Wang et al., 2025b; Mamidanna et al., 2025), syllogistic reasoning (Kim et al., 2024), graph connectivity (Saparov et al., 2025), and indirect object identification (Wang et al., 2023a). However, none studies full propositional-logic problems with [Variable relationships]+[Variable value assignment]+[Query] aspects while considering modern LLMs. We complement existing work on mechanistic analysis for such symbolic reasoning that considers small models trained on the task (Brinkmann et al., 2024; Guo et al., 2025b; Wang et al., 2024a; Zhu et al., 2024; Lin et al., 2025).

# 5   Discussion and Conclusion

We studied the reasoning mechanisms of three pre-trained models, ranging from 7B to 27B parameters. To do so, we introduced a simple propositional logic problem that was amenable to mechanistic interpretability tools. We characterized the LLMs' reasoning circuits, showing that they contain four families of attention heads. These components implement the reasoning pathway of "QUERY→Relevant Rule→Relevant Facts→Decision." Our findings provide valuable insights into the inner workings of LLMs on in-context logical reasoning problems, going beyond prior work. The fact that we found similar circuits in three distinct LLMs suggests that certain components organically arise from pre-training (even though it is unlikely that the LLMs were trained on identical problems).

A priori, it was not clear whether we could find isolated reasoning components in LLMs. In fact, we had a different conjecture based on studying the logic problem for small models, such as the 3-layer transformers (Appendix C and D). As mentioned before, for these small models, we found less modularity in their components, and in particular, they are less *lazy*. This behavior on Transformers trained specifically on a synthetic task is also supported by the work of Ye et al. (2024). Real-world LLMs seem to behave in a different manner, where not much processing happens until the QUERY token is seen, and their attention patterns tend to be sparser and exhibit more consistent and specialized structures. Plainly, LLMs not only ingest facts from the input prompt (via in-context learning), but they also have specialized internal components that process the provided information before downstream computation.

**Limitations.** An area of improvement is to extend to more complex problems. With more computational resources, we should aim to understand what structures are consistently employed by LLMs in implementing longer logical chains. A possible direction is analyzing latent circuits at critical token positions in the CoT trace of (thinking) LLMs, building on works such as Bogdan et al. (2025), but in more formalized settings, such as on longer propositional-logic problems. This would strengthen and generalize our claims about the existence and robustness of reasoning circuits. Moreover, there are unanswered questions about how model size and family impact the types of reasoning circuits. We compare the results on Gemma with Mistral-7B in Appendix B.4.1, providing some evidence that model size may be more indicative of capabilities than model family (given certain similarities between Gemma-2-9B and Mistral-7B). Embedding-level analysis and steering are other important future directions.

## Acknowledgements

The authors wish to thank Fred Zhang for the fruitful discussions around circuit analysis of LLMs, and the anonymous reviewers for their helpful comments and suggestions for improving the paper.

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

# Appendix

## A  Methodology and Problem Details

### A.1  Appendix Table of Contents and Outline

In this section, we provide an overview of our methodology that we presented at a high level in Section 2. We then give forward pointers to our additional experimental evidence. We aim to provide an outline of the LLM analysis, guiding the reader through the rest of the appendix.

- §A.2 gives representative examples of the propositional logic problem that we study. In general, we vary the variable names, the logical operator ("or" vs. "and"), the truth values of the variables (i.e., the "Facts"), and the QUERY variable.

- §B.1 provides details of the few-shot examples and input format, and some initial results about model accuracies. In particular, we engineer the in-context examples to ensure that all models perform much better than random guessing.

- §B.2 is an expository, background section that shows how activation patching can surface circuit components. For completeness, we illustrate the main techniques on a toy 2-layer model, with figures. The main technique is to run the LLM on altered (alt) prompts, while freezing certain components to the activations on the original (orig) prompt. Throughout, we also define some notation, such as $\Delta^{\mathcal{C}}_{orig \to alt}$, which is the intervened logit difference. The relationship between the orig and alt prompts depends on the specific experiment, where we may change the query variable, the rule locations, or the truth values.

- §B.3 defines the metrics for QUERY-based activation patching.

- §B.4 through §B.7 provides the full analysis of the Mistral 7B model. We break this down into multiple subsections to present the details of the experiments, where we use the same approach for the Gemma models later on. In particular, §B.5 has the sufficient test, §B.6 contains results for varying the **rule locations**, and §B.7 has results for varying the **facts**.

- §B.8 contains the experiments for Gemma 2 9B. This provides full details for the experiments that we report in the main paper.

- §B.9 similarly contains results for Gemma 2 27B, including attention head analysis, and a discussion of the mechanical differences between Gemma 2 9B and 27B.

- §C presents a full mechanistic analysis of small transformer models (e.g., 3-layer). We present the main insights in §C and then supplement this with the full details in §D. We train several models autoregressively on the propositional logic task. Then, we explore the mechanisms through a combination of experiments (patching, probing, classification, etc), focusing on a 3-layer model. We present the high level strategy employed by the 3-layer model to solve the logic problems in Figure 30, providing a contrast to the modular and more interpretable circuit in the pre-trained LLMs.

### A.2  Propositional logic problem and examples

In this section, we provide a more detailed description of the propositional logic problem we study in this paper, and list representative examples of the problem.

At its core, the propositional logic problem requires the reasoner to (1) distinguish which chain type is being queried (LogOp or linear), and (2) if it is the LogOp chain being queried, the reasoner must know what truth value the logic operator outputs based on the two input truth values.

Below we provide a comprehensive list of representative examples of our logic problem at length 2 (i.e. each chain is formed by one rule). We use [Truth values] to denote the relevant input truth value assignments (i.e. relevant facts) to the chain being queried below.

1. Linear chain queried, [True]
   - Rules: A or B implies C. **D implies E**.
   - Facts: A is true. B is true. **D is true**.

- Question: what is the truth value of C?
- Answer: D is true. D implies E; E True.

2. Linear chain queried, [False]

   - Rules: A or B implies C. **D implies E**.
   - Facts: A is true. B is true. **D is false**.
   - Question: what is the truth value of C?
   - Answer: D is false. D implies E; E undetermined.

3. LogOp chain queried, LogOp = OR, [True, True]

   - Rules: A **or** B implies C. D implies E.
   - Facts: **A is true. B is true.** D is true.
   - Question: what is the truth value of C?
   - Answer: B is true. A or B implies C; C True.

   *Remark* A.1. In this case, the answer "A true. A or B implies C; C True" is also correct.

4. LogOp chain queried, LogOp = OR, [True, False]

   - Rules: A **or** B implies C. D implies E.
   - Facts: **A is true. B is false.** D is true.
   - Question: what is the truth value of C?
   - Answer: A is true. A or B imples C; C True.

5. LogOp chain queried, LogOp = OR, [False, False]

   - Rules: A **or** B implies C. D implies E.
   - Facts: **A is false. B is false.** D is true.
   - Question: what is the truth value of C?
   - Answer: A false B false. A or B implies C; C undetermined.

6. LogOp chain queried, LogOp = AND, [True, True]

   - Rules: A **and** B implies C. D implies E.
   - Facts: **A is true. B is true.** D is true.
   - Question: what is the truth value of C?
   - Answer: A is true. B is true. A and B implies C; C True.

7. LogOp chain queried, LogOp = AND, [True, False]

   - Rules: A **and** B implies C. D implies E.
   - Facts: **A is true. B is false.** D is true.
   - Question: what is the truth value of C?
   - Answer: B is false. A and B implies C; C undetermined.

8. LogOp chain queried, LogOp = AND, [False, False]

   - Rules: A **and** B implies C. D implies E.
   - Facts: **A is false. B is false.** D is true.
   - Question: what is the truth value of C?
   - Answer: A is false. A and B implies C; C undetermined.

   *Remark* A.2. In this case, the answer "B is false. A and B implies C; C undetermined" is also correct.

**Importance of the first answer token**. As discussed in §1, correctly writing down the first answer token is central to the accuracy of the proof. First, it requires the model to *process every part of the context properly without CoT* due to the minimal-proof requirement of the solution. Moreover, it is the answer token position which demands the *greatest number of latent reasoning steps* in the whole answer, making it the most challenging token for the model to resolve. Therefore, this token is the most interesting place to focus our study on, as we are primarily interested in how the model *internalizes* and *plans* for the reasoning problems in this work.

# B The reasoning circuit in LLMs: experimental details

## B.1 Problem format

We present six examples of the propositional-logic problem in context to the Mistral-7B model, and four examples to the Gemma-2-9B model. We then prompt them for their answer to a newly appended problem. To generate the problem, we use the template and then sample variables and truth values. The two chains have independent proposition variables (leading to five distinct boolean variables). We randomly choose variables from capital English letters (A–Z) and truth values (true or false).

*Remark* B.1. To ensure fairness, for the disjunction problem, we provide the LLM with equal number of in-context examples for each query type (the OR chain and the linear chain) in random order. Please note that, while the premise variables in the linear chain examples could be set FALSE when not queried, the actual question (the seventh example which the model needs to answer) always sets the truth value assignment of the linear chain to be TRUE, preventing the model from taking a shortcut and bypassing the "QUERY→Relevant Rule" portion of the reasoning path.

Additionally, the models perform much better than random on these problems, with high accuracy for Mistral (above 70%) and Gemma-2-9B (above 83%). Gemma-2-27B achieves nearly perfect accuracy. This reflects a balanced evaluation, with the models queried for the LogOp and linear chains at a 50% probability each. Specifically, we tested the models on 400 samples. Mistral achieved 96% accuracy when QUERY is for the linear chain, and 70% accuracy when QUERY is for the OR chain (so they average above 70% accuracy). Similar trends hold for the other models.

## B.2 Causal mediation analysis: further explanations

This subsection complements the causal mediation analysis methodologies we presented in §2 in the main text. In particular, we illustrate how the interventions are done in the circuit discovery and verification processes, by using a 2-layer 2 attention head transformer as an example for simplicity.

Figure 6 illustrates the activation patching procedure in circuit discovery. We investigate how internal transformer components causally influence model output. In the specific example, we show how we would examine the causal influence of attention head (0,2)'s activations on the correct inference of the model.

Circuit verification, on the other hand, goes through a somewhat more complex process of interventions, as illustrated in Figure 7. Recall our main procedure (discussed in the main text).

1. We run the LLM on the *original* prompts, and cache the activations of the attention heads.

2. Now, we run the LLM on the corresponding *altered* prompts, however, we *freeze* all the attention heads' activations inside the model to their activations on the *original* prompts, *except* for those in the circuit $\mathcal{C}$ which we wish to verify (i.e. only the attention heads in $\mathcal{C}$ are allowed to run normally). We record the (circuit-intervened) altered logit differences on the altered prompts.

3. We average the circuit-intervened altered logit differences across the samples, namely $\frac{1}{N}\sum_{n=1}^{N} \Delta_{orig\to alt}^{\mathcal{C}}$[7], and check whether they approach the "maximal" altered logit difference, namely $\Delta_{alt}$[8].

*Remark* B.2. As the reader can observe, we do *not* freeze the MLPs in our intervention experiments. We note that the MLPs do not move information between the residual streams at different token positions, as they only perform processing of whatever information present at the residual stream. Therefore, similar to Wang et al. (2023a), we consider the MLPs as part of the "direct" path between two attention heads, and allow information to flow freely through them, instead of freezing them and disrupting the information flow between attention heads.

---

[7]This specific term reflects, on average, how much the model favors outputting the answer token for the altered prompts over the original prompts *after* the circuit interventions.

[8]Recall that this term is obtained by running the LLM on the altered prompts without any modification to its internal activations at all. This specific term reflects, on average, how much the (un-intervened) model favors outputting the answer tokens for the altered prompts over those of the original prompts, when it is run on the altered prompts.

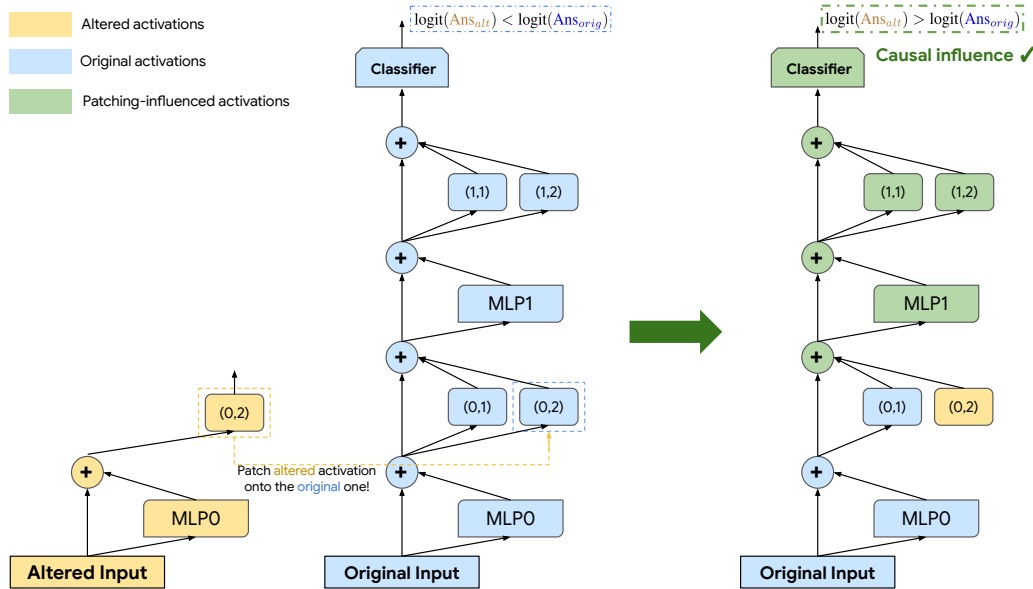

Figure 6: Illustration of how activation patching is performed in the *circuit-discovery* process (necessity-based patching), using a 2-layer, 2-head transformer as a simplified example here. An attention head is denoted as $(\ell, h)$.

In this specific illustrated example, we are studying the causal influence of attention head (0,2)'s activation on the correct inference of the model. After caching the altered activations of (0,2) (shown on the left), we run the model on the original prompt and cache the activations (shown in the middle), then replace the original activation of head (0,2) by its altered activations, and let the rest of the layers be computed normally (shown on the right) — they now operate out of distribution, and are colored in green.

In this specific example, the intervened run outputs logits which reflect "belief altering": that is, the probability for the answer token of the original prompt now is lower than the answer token for the altered prompt. This indicates that head (0,2) has causal influence on the corrent inference of the model.

### B.3 Finer details of activation patching

**Activation patching experiments: metrics**. We rely on a calibrated version of the logit-difference metric often adopted in the literature for the QUERY-based activation patching experiments (aimed at keeping the score's magnitude between 0 and 1). In particular, we compute the following metric for head $(\ell, h)$ at token position $t$:

$$\frac{\frac{1}{N} \sum_{n \in [N]} \Delta_{orig \to alt, n; (\ell, h, t)} - \Delta^{\dagger}_{orig}}{\Delta_{alt} - \Delta^{\dagger}_{orig}}. \tag{1}$$

where $\Delta^{\dagger}_{orig} = \frac{1}{N} \sum_{n \in [N]} \text{logit}(\boldsymbol{X}_{orig,n})[y_{alt,n}] - \text{logit}(\boldsymbol{X}_{orig,n})[y_{orig,n}]$, and $\Delta_{alt} = \frac{1}{N} \sum_{n \in [N]} \text{logit}(\boldsymbol{X}_{alt,n})[y_{alt,n}] - \text{logit}(\boldsymbol{X}_{alt,n})[y_{orig,n}]$. The closer to 1 this score is, the stronger the model's "belief" is altered; the closer to 0 it is, the closer the model's "belief" is to the original unaltered one.

Each of our experiments are done on 60 samples unless otherwise specified — we repeat some experiments (especially the attention-head patching experiments) to ensure statistical significance when necessary.

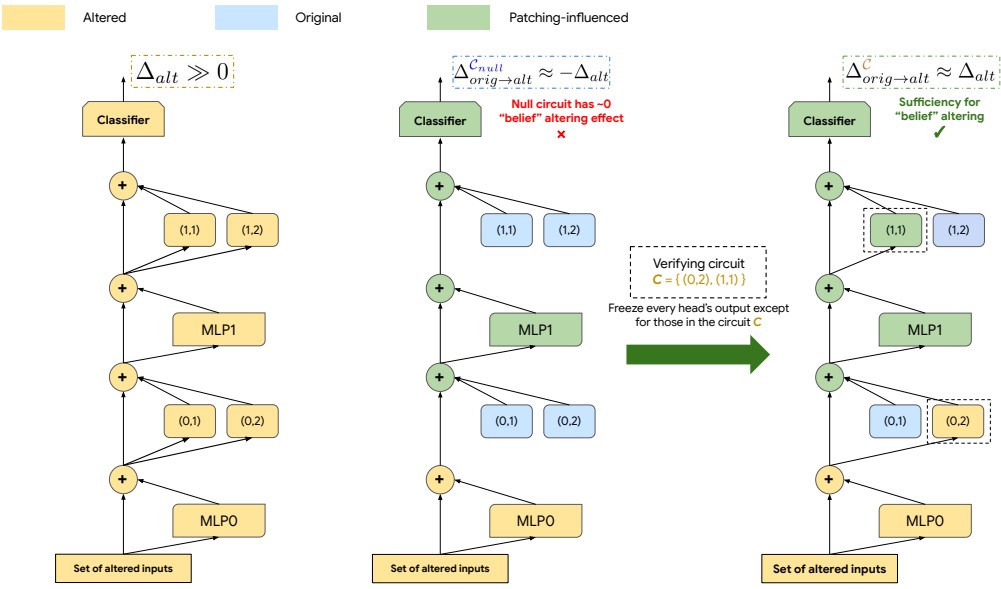

Figure 7: Illustration of how activation patching is performed in the *circuit-verification* process (sufficiency-based patching). We use a 2-layer 2-head transformer as a simplified example here. We use $(\ell, h)$ to denote an attention head.

In this specific illustrated example, we are verifying whether the circuit $\mathcal{C} = \{(0, 2), (1, 1)\}$ consisting of the two attention heads is *sufficient* for altering the "belief" of the model.

We first obtain $\Delta_{alt}$, the average altered logit difference, by running the model on the altered prompts without interventions. We also run the model on the original prompts and cache the attention heads' activations (these "original" activations are colored in blue in the figure).

The naive "baseline" for sufficiency verification is the null circuit $\mathcal{C}_{null} = \emptyset$ (shown in the middle): we freeze all the attention heads to their original activations when running the model on the altered prompts. This null circuit, as shown in this example, barely alters the model's "belief" from the original, as $\Delta^{\mathcal{C}_{null}}_{orig \to alt} \approx -\Delta_{alt}$, i.e. the model still strongly favors outputting the answer tokens for the original prompts over those of the altered prompts on average.

In contrast, if we unfreeze the attention heads in the circuit $\mathcal{C}$ when running the model (shown on the right), we observe that the model's circuit-intervened logit difference approaches the "maximal" altered logit difference $\Delta_{alt}$. This indicates that the attention heads in $\mathcal{C}$ are sufficient for correctly manipulating the information flow (and processing the information) for reaching the right answer.

## B.4 Reasoning circuits in Mistral-7B

### B.4.1 Attention head group patching

Starting in this subsection §B.4.1 and ending at §B.7, we present and visualize the attention heads with the highest average intervenes logit differences, along with their standard deviations (error bars). Furthermore, we provide further causal evidence for sub-families of attention heads in the circuit.

We note that *Grouped-Query Attention* used by Mistral-7B adds subtlety to the analysis of which attention heads have strong causal influence on the LLM's correct output. (In Mistral-7B-v0.1, each attention layer has 8 key and value activations, and 32 query activations. Therefore, heads $(\ell, h \times 4)$ to $(\ell, h \times 4 + 3)$ share the same key and value activation.) Patching a single head might not yield a high logit difference, since other heads in the same group (which possibly perform a similar function) could overwhelm the patched head and maintain the model's previous "belief". Therefore, we also run a *coarser-grained* experiment which simultaneously patches the attention heads sharing the same key and value activations, shown in Figure 8(b). This experiment reveals that heads belonging to the group (9, 24 - 27) also have high intervened logit difference. Combining with the observation that (9,25;26) have somewhat positive scores in the single-head patching experiments, and by examining these two head's attention patterns (which shall be discussed in detail in the immediate next subsection), we determine that they also should be included in the circuit.

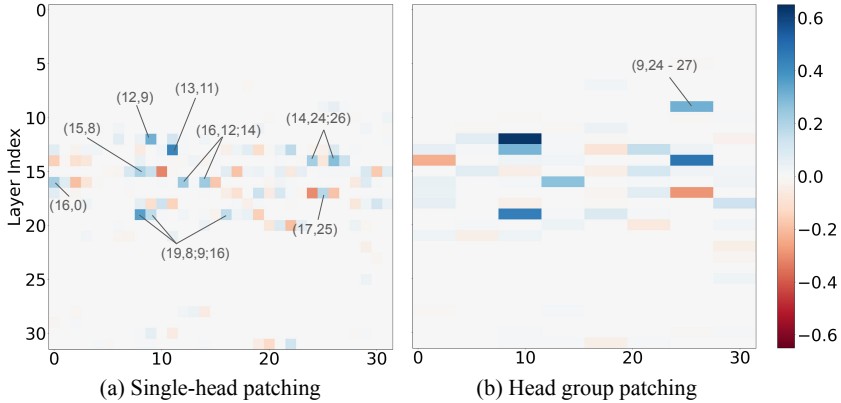

|  | (a) Single-head patching | (b) Head group patching |

Figure 8: Attention head patching, highlighting the ones with the highest intervened logit difference; $x$-axis is the head index. (a) shows single-head patching results (same as the one shown in the main text, repeated here for the reader's convenience), and (b) shows a coarser-grained head patching in groups. In (b), we only highlight the head groups that are not captured well by (a).

### B.4.2 Attention patterns of QUERY-sensitive attention heads

In this subsection, we provide finer details on the attention patterns of the attention heads we discovered. Note that the attention weights percentage we present in this section are calculated by dividing the observed attention weight at a token position by the total amount of attention the head places in the relevant context, i.e. the portion of the prompt which excludes the 6 in-context examples.

**Queried-rule locating heads**. Figure 9 presents the average attention weight the queried-rule locating heads place on the "conclusion" variable and the period "." immediately after the queried rule at the QUERY token position (i.e. the query activation of the heads come from the residual stream at the QUERY token position) — (12,9) is an exception to this recording method, where we only record its weight on the conclusion variables alone, and already observe very high weight on average. The heads (12,9), (14,24), (14,26), (9,25), (9,26) indeed place the majority of their attention on the correct position *consistently* across the test samples. The reason for counting the period after the correct conclusion variable as "correctly" locating the rule is that, it is known that LLMs tend to use certain "register tokens" to record information in the preceding sentence.

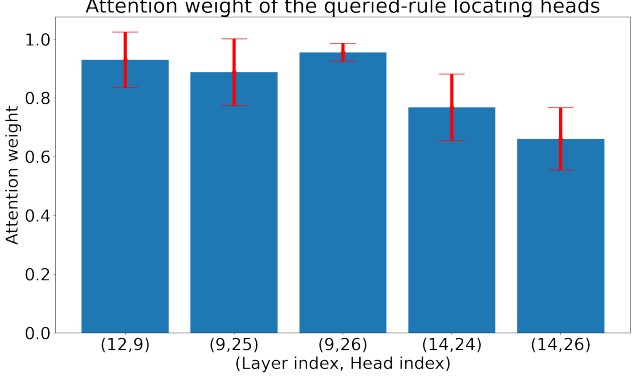

Figure 9: Mistral 7B. Average attention weights of the queried-rule locating heads, along with the standard deviations. The weights are calculated by dividing the actual attention weight placed on the correct "conclusion" variable of the rule and the period "." immediately after, by the total amount of attention placed in the relevant context (i.e. the prompt excluding the 6 in-context examples). *Head (12,9) is an exception: we only record its attention right on the conclusion variable, and still observe* $93.0 \pm 9.4\%$ *"correctly placed" attention on average.*

We can observe that head (12,9) has the "cleanest" attention pattern out of the ones identified, placing on average $93.0 \pm 9.4\%$ of it attention on the correct conclusion variable alone. The more diluted attention patterns of the other heads likely contribute to their weaker intervened logit difference score shown in §3.1 in the main text.

**Queried-rule mover heads**. Figure 10 shows the attention weight of the queried-rule mover heads. While they do not place close to 100% attention on the QUERY location consistently (when the query activation comes from the residual stream from token ":", right before the first answer token), the top-1 attention weight consistently falls on the QUERY position, and the second largest attention weight is much smaller. In particular, head (13,11) places $54.2 \pm 12.5\%$ attention on the QUERY position on average, while the second largest attention weight in the relevant context is $5.2 \pm 1.1\%$ on average (around 10 times smaller; *this ratio is computed per sample and then averaged*).

**Extra note about head (16,0)**: it does *not* primarily act like a "mover" head, as its attention statistics suggest that it processes an almost even *mixture* of information from the QUERY position and the ":" position. Therefore, while we present its statistics along with the other queried-rule mover heads here since it does allocate significant attention weight on the QUERY position on average, we do not list it as such in the circuit diagram of Figure 3. **Furthermore, we do not include it as part of the circuit** $\mathcal{C}$ **in our circuit verification experiments**.

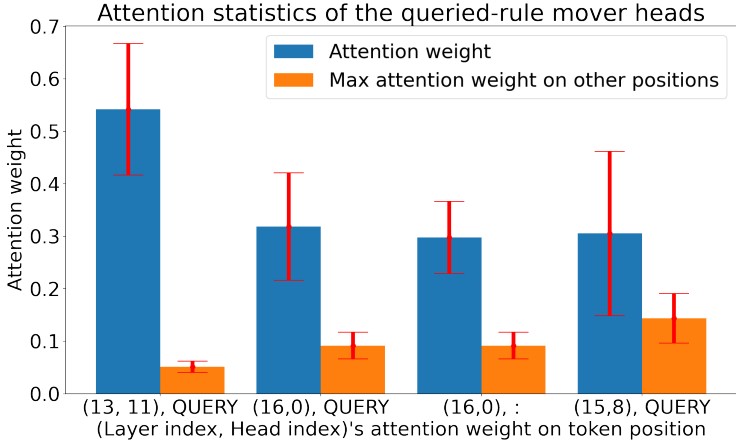

Figure 10: Mistral 7B. Average attention weights of the queried-rule mover heads, along with the standard deviations. The raw attention patterns are obtained at token position ":" (i.e. the query activation comes from the residual stream at the ":" position), right before the first answer token, and the exact attention weight (indicated by the blue bars) is taken at the QUERY position; for head (16,0), we also obtain its attention weight at the ":" position, as we found that it also allocates a large amount of attention weight to this position in addition to the QUERY position. Note: for (15,8), we found that it only acts as a "mover" head when the linear chain is being queried, so we are only reporting its attention weight statistics in this specific scenario; the other heads do not exhibit this interesting behavior, so we report those heads' statistics in all query scenarios.

**Fact processing heads**. Figure 11 below shows the attention weights of the fact processing heads; the attention patterns are obtained at the ":" position, right before the first answer token, and we sum the attention weights in the Fact section (starting at the first fact assignment, ending on the last "." in this section of the prompt). It is clear that they place significant attention on the Fact section of the relevant context. Additionally, across most samples, we find that these heads exhibit the tendency to assign lower amount of attention on the facts with FALSE value assignments across most samples, and on a nontrivial portion of the samples, they tend to place greater attention weight on the correct fact (this second ability is not consistent across all samples, however). Therefore, they do appear to perform some level of "processing" of the facts, instead of purely "moving" the facts to the ":" position.

**Decision heads**. Figure 12 shows the attention weights of the decision heads *on samples where the model outputs the correct answer (therefore, about 70% of the samples)*. The attention patterns are obtained at the ":" position. We count the following token positions as the "correct" positions:

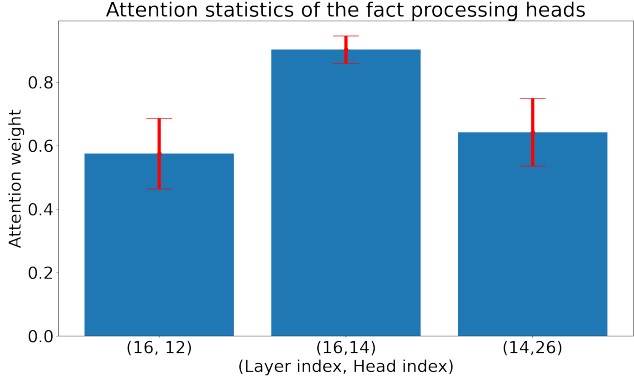

Figure 11: Mistral 7B. Average attention weights of the fact processing heads computed at the ":" token position (last position before the answer), along with the standard deviations. The weights are calculated by dividing the actual attention weight placed in the Fact section by the total amount of attention placed in the relevant context (i.e. the part of the prompt excluding the 6 in-context examples).

- In the Rules section, we count the correct answer token and the token immediately following it as correct.
- In the Facts section, we count the sentence of truth value assignment of the correct answer variable as correct (for example, "A is true.").
- Note: the only exception is head (19,8), where we only find its attention on exactly the correct tokens (not counting any other tokens in the context); we can observe that it still has the cleanest attention pattern for identifying the correct answer token.

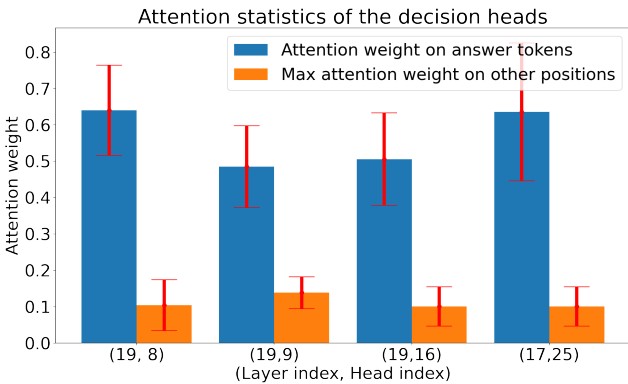

Figure 12: Mistral 7B. Average attention weights of the decision heads, along with the standard deviations. The weights are calculated by dividing the actual attention weight placed on the correct answer tokens by the total attention the model places in the relevant context.

An interesting side note worth pointing out is that, (17,25) tends to only concentrate its attention in the facts section, similar to the fact-processing heads. The reason which we do not classify it as a fact-processing head and instead as a decision head is that, in addition to finding that their attention patterns tend to concentrate on the correct fact, evidence presented in §B.7 below suggest that they are not directly responsible for locating and moving the facts information to the ":" position, while the heads (16,12;14) exhibit such tendency strongly.

## B.5    Sufficiency tests for circuit verification

In §3.2, we presented a sufficiency test of the circuit. Here, we elaborate further on the experimental procedures and finer details of the experiment.

The circuit which we perform verification on is the union of the four attention head families, $\mathcal{C} = QRLH \cup QRMH \cup FPH \cup DH$, with

- $QRLH$ = Queried-Rule Locating Heads = $\{(9, 25; 26), (12, 9), (14, 24; 26)\}$ patched at token position QUERY;
- $QRMH$ = Queried-Rule Mover Heads = $\{(13, 11; 22), (15, 8)\}$ patched at the ":" position (the last position of context);
- $FPH$ = Fact-Processing Heads = $\{(16, 12; 14), (14, 26)\}$ patched at the ":" position;
- $DH$ = Decision Heads = $\{(19, 8; 9; 16), (17, 25)\}$ patched at the ":" position.

We obtain the following results.

| $\mathcal{C}^\dagger$ | $\mathcal{C}_{null}$ | $\mathcal{C}$ | $\mathcal{C} - QRLH$ | $\mathcal{C} - QRMH$ | $\mathcal{C} - FPH$ | $\mathcal{C} - DH$ |
|---|---|---|---|---|---|---|
| $\Delta^{\mathcal{C}^\dagger}_{normal}/\Delta_{normal}$ | -1.0 | **0.98** | -1.02 | -0.99 | -0.25 | -0.89 |

Table 2: Mistral 7B. $\Delta^{\mathcal{C}^\dagger}_{normal}/\Delta_{normal}$, with different choices of $\mathcal{C}^\dagger$. $\mathcal{C}_{null}$ denotes the *empty* circuit, i.e. the case where every attention head output is frozen to its counterfactual version. We abbreviate the attention head families, for example, $DH$ = decision heads; $\mathcal{C} - DH$ = full circuit but with the decision heads removed.

An exception is that the queried-rule locating head $(14, 24)$ is also patched at the ":" position, as we observed that it tends to concentrate attention at the queried rule at this position: it does not locate the queried rule as consistently as it does at the QUERY position, however. We still chose to patch it at this position as we found that it tends to improve the altered logit difference, indicating that either the model relies on this head to pass certain additional information about the queried rule to the ":" position, or certain later parts of the circuit do rely on this head for queried-rule information. The exact function of this attention head remains part of our future study in the reasoning circuit of Mistral-7B. We likely need to examine this head's role in other reasoning problems to clearly understand what its role is at different token positions, and whether there is deeper meaning behind the fact that, their apparently redundant actions at different token positions all seem to have causal influence on the model's inference.

**Challenges of reasoning circuit sufficiency verifications**. From what we can see, verifying the sufficiency of a *reasoning* circuit is a major open problem. Part of the root of the problem lies in what exactly counts as a circuit that is truly relevant to *reasoning*: attention heads and MLPs responsible for lower-level processing such as performing change of basis of the token representations, storing information at register tokens (such as the periods "." after sentences), and so on, do not truly belong to a "reasoning" circuit in the narrow definition of the term. In our considerations, a "narrow" definition of a reasoning circuit is one which is *QUERY sensitive* and *has strong causal influence on the correct output of the model on the reasoning problems*. The first condition of QUERY sensitivity is justified by noting that the QUERY lies at the root of the reasoning chain of "QUERY→Relevant Rule(s)→Relevant Fact(s)→Decision". We do not analyze through what circuit/internal processing the "QUERY", "Relevant Rule(s)" and "Relevant Fact(s)" underwent from token level to representation level (notice that the reasoning circuit we identified starts at layer 9: it is entirely possible for the token embeddings of these important items to have undergone significant processing by the attention heads and MLPs in the lower layers). Simply setting the lower layers' embeddings to the zero vector, to their mean activations or some fixed embeddings which erase the instance-dependent information could completely break the circuit.

### B.6 Queried-rule location interventions: analyzing the queried-rule locating heads

In this experiment, we *only* swap the *location* of the linear rule with the LogOp rule in the *Rule* section of the question, while keeping everything else the same (including all the in-context examples). As an example, we alter the rules as follows. The prompts have the same answer.

| *[... in-context examples ...]* | | *[... in-context examples ...]* |
|---|---|---|
| **Rules**: $A$ or $B$ implies $C$. $D$ implies $E$. | $\xrightarrow[\text{prompt}]{\text{altered}}$ | **Rules**: $D$ implies $E$. $A$ or $B$ implies $C$. |
| **Facts**: $A$ is true. $B$ is false. $D$ is true. | | **Facts**: $A$ is true. $B$ is false. $D$ is true. |
| **Question**: what is the truth value of $C$? | | **Question**: what is the truth value of $C$? |

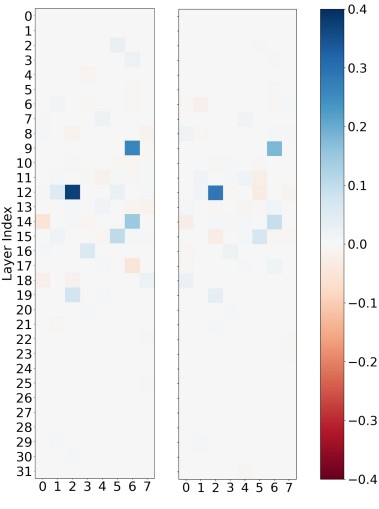
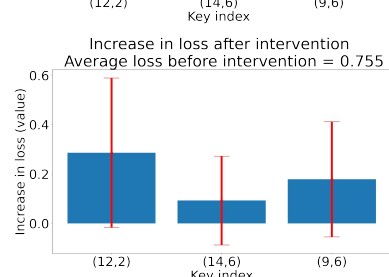

(a) Key patching - loss increase
Left: QUERY lin chain; Right: QUERY OR chain

(b) Average increase in loss
Top: QUERY lin chan; Bottom: QUERY OR chain

Figure 13: Mistral 7B. Key activations patching results. In this experiment, we swap the location of the linear rule and the LogOp rule in the Rule section and keep everything else in the prompt the same; we patch the key activations of the attention heads in the Rule section only. (a) visualizes the average increase in the cross-entropy loss with respect to the true target (the true first token of the answer) for all key indices, and (b) shows the average and standard deviation of the top three key indices with the highest loss increase. Observe that these are the keys for the queried-rule locating heads (12,9), (14,24;26) and (9,25;26) identified in §3.1.

If the *queried-rule locating heads* (with heads (12,9), (14,25;26), (9,25;26) being the QUERY-sensitive representatives) indeed perform their functions as we described, then when we run the model on the clean prompts, patching in the *altered key activations* at these heads (within the Rules section) should cause "negative" change to the model's output, since it will cause these heads to mistake the queried-rule location in the altered prompt to be the right one, consequently storing the wrong rule information at the QUERY position. In particular, the model's *cross-entropy loss* with respect to the original target should increase. This is indeed what we observe.

The average increase in cross-entropy loss exhibit a trend which corroborate the hypothesis above, shown in Figure 13. While the average cross-entropy loss on the original samples is 0.463, patching (12,9), (14,24;26) and (9,25;26)'s keys (with corresponding key indices (12,2), (14,6) and (9,6)) in the Rule section.

Patching the other *QUERY-sensitive* attention heads' keys in the Rule section, in contrast, show significantly smaller influence on the loss on average, telling us that their responsibilities are much less involved with *directly* finding or locating the queried rule via attention.

Note: this set of experiments was run on 200 samples instead of 60, since we noticed that the standard deviation of some of the attention heads' loss increase is large.

*Remark* B.3. While attention heads with key index (15,5) (i.e. heads (15, 20-23)) did not exhibit nontrivial sensitivity to QUERY-based patching (discussed in Section 3.1 in the main text), patching this key activation does result in a nontrivial increase in loss. Examining the attention heads belonging to this group, we find that they indeed also perform the function of locating the queried rule similar to head (12,9). We find them to be less accurate and place less attention on the exact rule being queried on average, however: this weaker "queried-rule locating ability" likely contributed to their low scores in the QUERY-based patching experiments presented in the main text.

## B.7 Facts interventions: analyzing the fact-processing and decision heads

In this section, we aim to provide further validating evidence for the fact-processing heads and the decision heads. We experiment with flipping the truth value assignment for the OR chain while

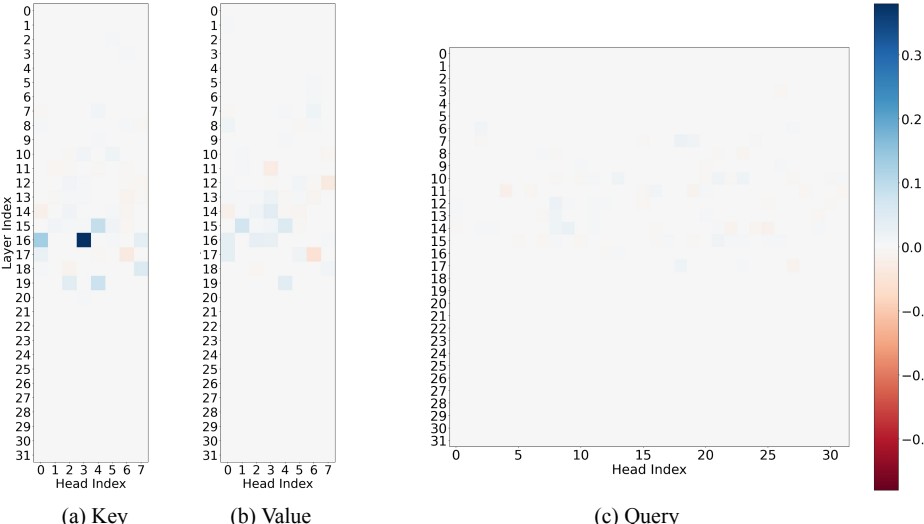

(a) Key          (b) Value          (c) Query

Figure 14: Mistral 7B. Key, value and query activation patching in the Facts section, with the metric being the calibrated intervened logit difference. The truth value assignments for the OR chain is flipped (while keeping everything else in the prompt the same), and the OR chain is always queried. Observe that only the *key* activations at index (16,3) obtain a high intervened logit difference score of approximately 0.34 (this key index corresponds to the attention heads (16, 12 - 15)). Also observe that the value and query activations in the facts section do not exhibit strong causal influence on the correct inference of the model.

keeping everything else the same in the prompt (we always query for the OR chain in this experiment). As an example, we alter the truth values as follows.

| *[... in-context examples ...]* | | *[... in-context examples ...]* |
|---|---|---|
| **Rules**: $A$ or $B$ implies $C$. $D$ implies $E$. | | **Rules**: $A$ or $B$ implies $C$. $D$ implies $E$. |
| **Facts**: $A$ **is true.** $B$ **is false.** $D$ is true. | altered prompt → | **Facts**: $A$ **is false.** $B$ **is true.** $D$ is true. |
| **Question**: what is the truth value of $C$? | | **Question**: what is the truth value of $C$? |

In this example, the variable that the answer depends on flips from $A$ to $B$ (since after altering, only $B$ is true). The (calibrated) intervened logit difference is still a good choice in this experiment, therefore we still rely on it to determine the causal influence of attention heads on the model's inference, just like in the QUERY-based patching experiments.

If the *fact-processing heads* (with (16,12;14) being the QUERY-sensitive representatives) indeed perform their function as described (moving and performing some preliminary processing of the facts as described before), then patching the *altered key activations in the Facts section* of the problem's context would cause these attention heads to obtain a nontrivial intervened logit difference, i.e. it would help in bending the model's "belief" in what the facts are (especially the TRUE assignments in the facts section), thus pushing the model to flip its first answer token. This is indeed what we observe. In Figure 14, we see that only the key activations with index (16,3) (corresponding to heads (16, 12 - 15)) obtain a much higher score than every other key index, yielding evidence that only the heads with key index (16,3) rely on the facts (especially the truth value assignments) for answer. Moreover, notice that patching the *key* activations of the *decision heads* does not yield a high logit difference on average, telling us that the decision heads do *not directly* rely on the *truth value assignment* of the variables for inference (we wish to emphasize again that, the *positions* of the variables in the Facts section are not altered, only the truth value assignments for the two variables of the OR chain are flipped).

Finally, for additional insights on the decision heads (19,8;9;16) and (17,25), we find that by patching the query activations of these decision heads at the ":" position yields nontrivial intervened logit

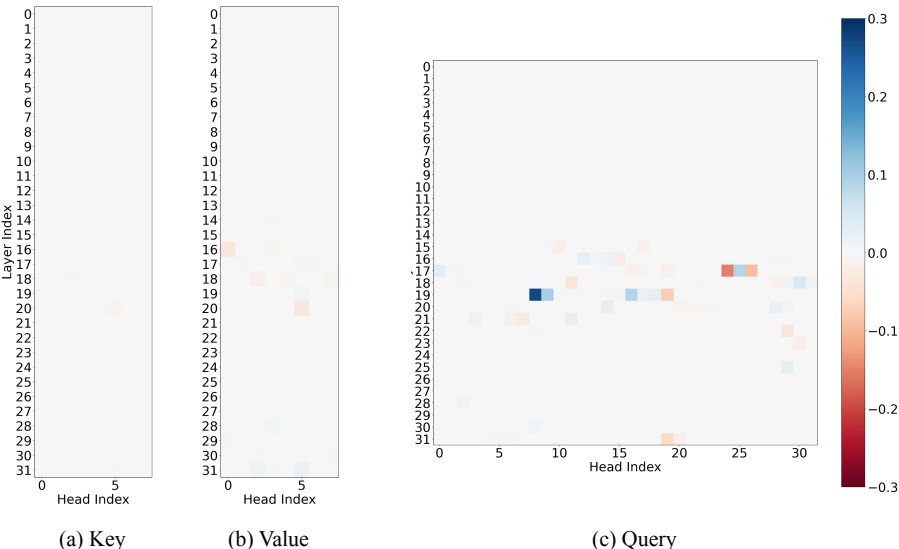

| (a) Key | (b) Value | (c) Query |
|---------|-----------|-----------|

Figure 15: Mistral 7B. Key, value and query activation patching at the ":" position (last token position in the context, right before the answer token), with the metric being the calibrated intervened logit difference. The truth value assignments for the OR chain is flipped (while keeping everything else in the prompt the same), and the OR chain is always queried. Observe that only the *query* activations at index (19,8) obtain a high intervened logit difference score of approximately 0.28; the other decision heads (19,9;16) and (17,25) also obtain nontrivial scores when their queries are patched. Also observe that the key and value activations at the ":" position do not exhibit strong causal influence on the correct inference of the model when we only flip the truth value assignments for the OR chain.

difference, as shown in Figure 15(c) ((19,8) has an especially high score of about 0.27). In other words, the query activation at the ":" position (which should contain information for flipping the answer from one variable of the OR chain to the other, as gathered by the fact-processing heads) being fed into the decision heads indeed have causal influence on their "decision" making. Moreover, patching the value activation of these heads at ":" does not yield nontrivial logit difference, further suggesting that it is their attention patterns (dictated by the query information fed into these heads) which influence the model's output logits.

### B.8 Reasoning circuit in Gemma-2-9B

In this section, we present an analysis of the reasoning circuit of Gemma-2-9B in solving the same reasoning problem which Mistral-7B was examined on from before. *We find that the discovered attention heads' attention patterns inside Gemma-2-9B bear surprising resemblance to Mistral-7B's*: according to their highly specialized attention patterns, they can also be categorized into the four families of attention heads which Mistral-7B employs to solve the problem, namely the queried-rule locating heads, queried-rule mover heads, fact-processing heads, and decision heads. While it is too early to draw precise conclusions on how similar the two circuits in the two LLMs truly are, the preliminary evidence suggests that the reasoning circuit we found in this work potentially has some degree of universality.

### B.8.1 QUERY-based attention head activation patching

We perform activation patching of the attention head output of Gemma-2-9B, by flipping the QUERY in the prompt pairs. This is the same procedure we used to discover the attention head circuit for Mistral-7B as discussed in §A.1 and B.2. We highlight the attention heads with the strongest causal influence on the model's (correct) inference in Figure 16.

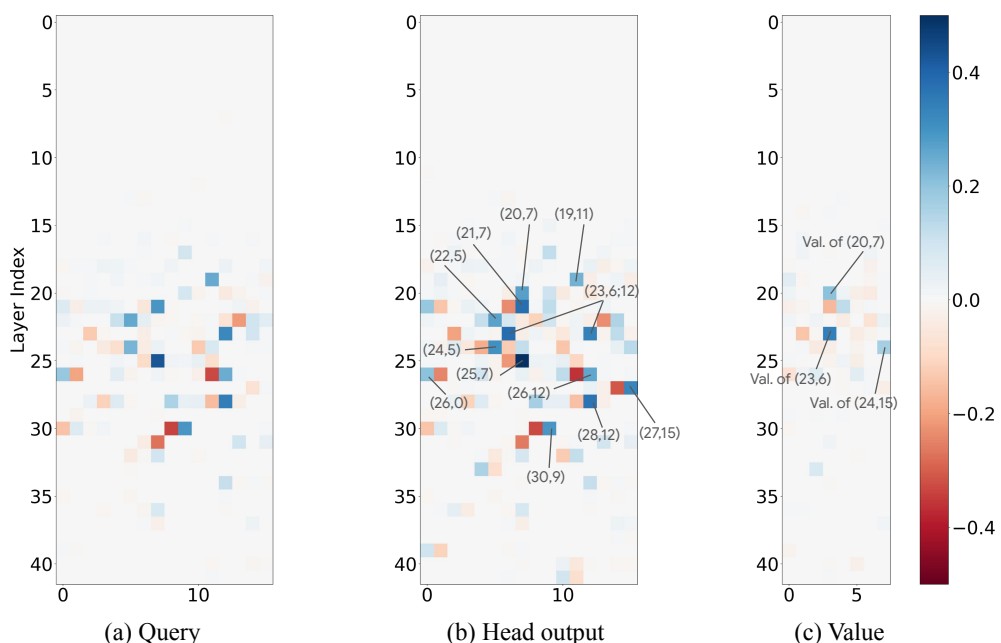

(a) Query        (b) Head output        (c) Value

Figure 16: Gemma 2 9B. QUERY-based activation patching results of Gemma-2-9B, with subcomponent patching on the query and value activations. We highlight the attention heads with the highest calibrated intervened logit difference.

### B.8.2 Attention patterns of QUERY-sensitive attention heads in Gemma-2-9B

**Queried-rule locating heads**. The queried rule locating heads inside Gemma-2-9B, namely $\{(19, 11), (21, 7), (22, 5), (23, 12)\}$, are very similar in their attention patterns to those in Mistral-7B. At the QUERY position, their attention concentrates on the conclusion token of the queried rule, and the "." which follows. Interestingly, heads (21,7), (22,5) and (23,12) also tend to place some attention on the "implies" token of the queried rule. Another intriguing difference they exhibit is redundant behavior: these attention heads are often observed to have almost exactly the same attention pattern at the "." and "Answer" token positions following the QUERY token. We visualize their attention statistics at the QUERY position in Figure 17.

**Queried-rule mover heads**. When the query activations of the queried-rule mover heads $\{(20, 7), (23, 6), (24, 15)\}$ come from the ":" residual stream, they have fixed attention patterns which focus a large portion of their attention weights on the QUERY token and two token positions

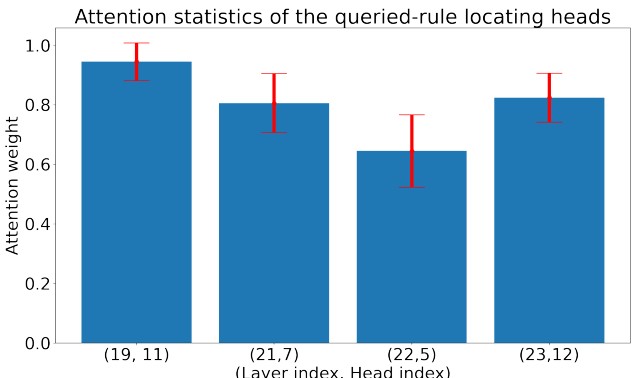

Figure 17: Gemma 2 9B. Average attention weights of the queried-rule locating heads in Gemma-2-9B, along with the standard deviations. The attention pattern is obtained at the QUERY position (i.e. query activation of the attention head is from the residual stream at the QUERY token position). We record the attention weight on the queried rule.

following it, namely the "." and "Answer" token. Their attention weights are slightly more diffuse compared to their counterparts in Mistral-7B, likely due to the queried-rule locating heads performing similar functions at the "." and "Answer" positions. Furthermore, as shown in Figure 16(c), we note that these attention heads are the only ones where patching their value activations results in a large intervened logit difference, further suggesting their role in performing a fixed "moving" action. We record their attention weights in Figure 18.

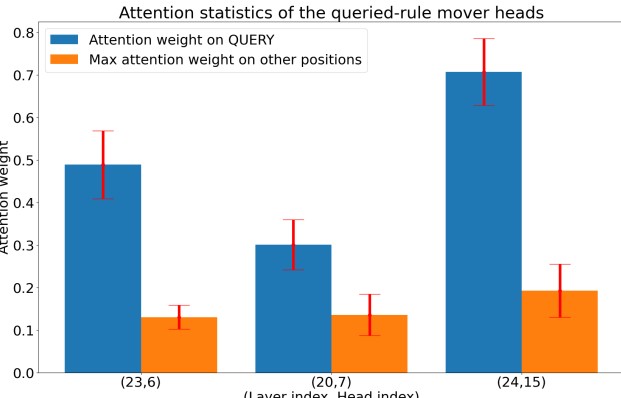

Figure 18: Gemma 2 9B. Average attention weights of the queried-rule mover heads in Gemma-2-9B, along with the standard deviations. The attention pattern is obtained at the ":" position, and we sum the attention weights at the QUERY position and the "." and "Answer" token positions which immediately follow QUERY.

**Fact-processing heads**. The fact-processing heads $\{(24,5),(25,7),(26,0),(26,12)\}$'s attention patterns at the ":" position tend to place larger weight on the correct fact for the answer, similar to the fact-processing heads in Mistral-7B. An interesting difference does exist though: heads (24,5) and (25,7) also tend to place a nontrivial amount of weight on the QUERY and ":" token positions, indicating that these heads are relying on some form of mixture of information present at those positions for processing. While it is reasonable to hypothesize that these heads are likely relying on the queried-rule information present in the QUERY and ":" residual streams, we have not confirmed this hypothesis in our current experiments. We visualize the statistics of these heads in Figure 19.

**Decision heads**. The decision heads $\{(28,12),(30,9)\}$'s attention pattern are obtained at the ":" position. They bear strong resemblance to those in Mistral-7B: they place significant attention on the correct answer token (in both the rules and facts sections, same as Mistral-7B's decision heads), and little attention weight anywhere else. This is shown in Figure 20.

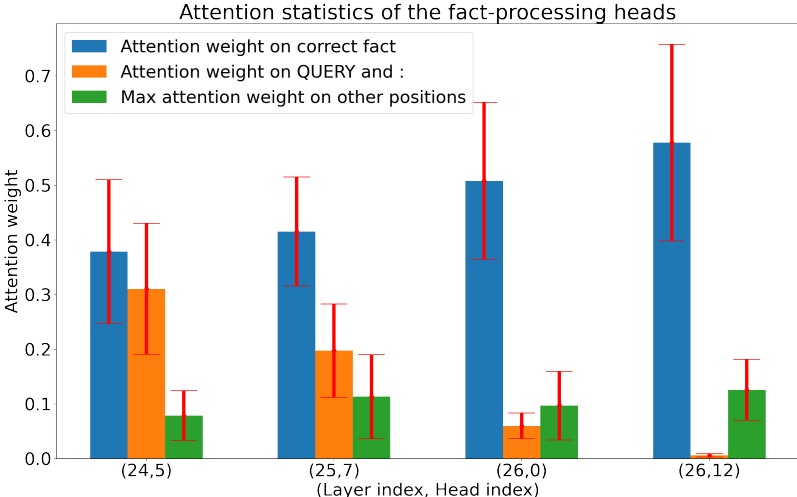

Figure 19: Gemma 2 9B. Average attention weights of the fact-processing heads in Gemma-2-9B, along with the standard deviations. The attention pattern is obtained at the ":" position. We record the attention weights at the correct fact, QUERY and ":" positions, and the maximum weight on any other position.

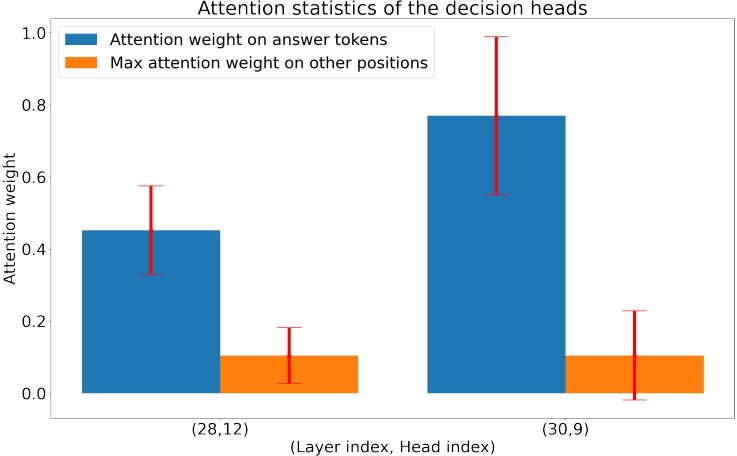

Figure 20: Gemma 2 9B. Average attention weights of the decision in Gemma-2-9B, along with the standard deviations. The attention pattern is obtained at the ":" position. We record the attention weights at the correct answer token positions.

### B.8.3 Circuit verification of Gemma-2-9B

In this sub-section, we present two versions of the circuit we performed verifications on. Version 1 was older and coarse-grained due to limited resources available at the time. Version 2 is finer-grained and performed on a larger number of samples. We leave the older version 1 here to ensure transparency to our circuit analysis process and show the complexities of circuit verifications.

**Version 1 of Circuit Verification**. This is the old version of the Gemma-2-9B circuit originally found in the first version of our work, which we observed to already have a high faithfulness score.

The circuit which we perform verification on is the union of the four attention head families, $\mathcal{C} = QRLH \cup QRMH \cup FPH \cup DH$, with

- $QRLH$ = Queried-Rule Locating Heads = $\{(19, 11), (21, 7), (22, 5), (23, 12)\}$;
- $QRMH$ = Queried-Rule Mover Heads = $\{(20, 7), (23, 6), (24, 15)\}$;

- $FPH$ = Fact-Processing Heads = $\{(24,5),(25,7),(26,0),(26,12)\}$;
- $DH$ = Decision Heads = $\{(28,12),(30,9)\}$.

*Remark.* This version of circuit verification is performed in a *coarse-grained* manner, as we patch the output of the attention heads in $\mathcal{C}$ from the QUERY position to the ":" position, instead of clearly distinguishing the token positions which each head primarily focuses on.

| $\mathcal{C}^{\dagger}$ | $\mathcal{C}_{null}$ | $\mathcal{C}$ | $\mathcal{C} - QRLH$ | $\mathcal{C} - QRMH$ | $\mathcal{C} - FPH$ | $\mathcal{C} - DH$ |
|---|---|---|---|---|---|---|
| $\Delta^{\mathcal{C}^{\dagger}}_{orig \to alt}/\Delta_{alt}$ | -1.0 | **0.94** | -0.97 | -0.40 | 0.17 | -1.11 |

Table 3: Gemma 2 9B. $\Delta^{\mathcal{C}^{\dagger}}_{orig \to alt}/\Delta_{alt}$ for Gemma-2-9B, with different choices of $\mathcal{C}^{\dagger}$. $\mathcal{C}_{null}$ denotes the *empty* circuit, i.e. the case where no intervention is performed. We abbreviate the attention head families, for example, $DH$ = decision heads; $\mathcal{C} - DH$ = full circuit but with the decision heads removed.

We find that by patching all 13 attention heads in $\mathcal{C}$, $\Delta^{\mathcal{C}}_{orig \to alt}$ is about 94% of the "maximal" average logit difference $\Delta_{alt}$ on the altered samples. Moreover, removing any one of the four families of attention heads from $\mathcal{C}$ in the circuit interventions renders the "belief altering" effect of the intervention almost trivial.

**Version 2 of Circuit**. With more resources available, we conducted a finer-grained test of the circuit, covering a larger number of attention heads which have nontrivial causal scores in our circuit search process, and restricted the token positions which the families are allowed to operate on, similar to our Mistral experiments before. We also run the verification test on a larger number of samples (240 samples, instead of 80). The following updates are made.

1. *Token position restrictions*. For the four families of heads, we now *restrict the token positions* on which they are allowed to operate normally to be (i.e. the token positions on which they are unfrozen):
   - QUERY for *queried-rule locating* heads;
   - QUERY and ":" for the *mover* heads, since they have diffuse attention on these positions.
   - The last token position ":" for the *fact-processing and decision heads*, since we observed their primary processing for writing down the answer token to take place at the last token position.

2. *Additional QRLHs*. We add heads (17,9), (21,0) to the QRLH family. (17,9) did not exhibit strong causal score in the initial search when we set the mediator to be the output of the attention head, but it surfaced with high causal score in our rule-location-swap experiment (where the key activation in the Rules section is the mediator), and its attention statistics strongly suggest that it is locating the queried rule. (21,0) is included for completeness this time, since it has moderately high causal score in the circuit search, and its attention pattern resembles those of the QRLH family closely.

3. *The "post-processing" heads*. With these token position restrictions, the 4 families of heads alone recover 86% of the logit difference – a high score, but still lower than before by a nontrivial margin. We find that for two more attention heads deeper in the model which have high causal scores in the search process, (27,15) and (33,4), un-freezing them at the QUERY and ":" positions (along with the rest of the circuit as we described above) helps us recover 95% of the un-intervened average logit difference.
   - We term these heads the "post-processing" heads. Their attention patterns are somewhat similar to the mover heads' – consistently heavy attention at the last token position ":", with small amount of attention on QUERY – yet they are deeper in the model, and patching their value activations alone does not yield high causal scores. This suggests that they are not merely moving information, but boosting the answer signal given by the core circuit (from the 4 families of heads). The exact role of these two post-processing heads remains mysterious, and requires future investigations.
   - These two heads were not included in the Version 1 circuit as they have less interpretable functional roles in the circuit, and lower causal scores than the dominant heads in the four core circuit families (especially head (33,4)). Another reason to include them now

is that they seem to have counterparts in the larger Gemma-2-27B model: some of the heads in the deeper layers in that model also exhibit the tendency to only focus on the last token position, and have nontrivial causal scores.

**Further remarks**.

1. To see the nuances with the sufficiency tests of these circuits more clearly, we encourage the interested reader to examine, and ideally work with our open-sourced Jupyter notebook "LLM Analysis Part 1 - Circuit search, interpretation, and verification.ipynb" in our GitHub repository, which covers the full workflow from environment setup to the details of circuit tests. By ablating different combinations of the circuit components and token positions (and with/without the post-processing heads), the reader should be able to gain a more concrete understanding of the reasoning circuits.

2. We find it surprising that two LLMs (Mistral-7B and Gemma-2-9B) which are trained with different procedures and data ended up relying on attention-head circuits which bear strong resemblance to each other's. In the current literature, it is unclear how one can rigorously quantify the similarity of two nontrivial circuits inside different LLMs, however, this subsection does yield preliminary evidence that, the reasoning circuit we discover potentially has some degree of *universality* to it, and is likely an emergent trait of LLMs.

### B.8.4 Analysis of queried-rule locating heads

Similar to the Mistral-7B experiments in verifying the queried-rule locating heads, we *only* swap the *location* of the linear rule with the LogOp rule in the *Rule* section of the question, while keeping everything else the same (including all the in-context examples). As an example, we alter "Rules: A or B implies C. D implies E." to "Rules: D implies E. A or B implies C." while keeping everything else the same. The two prompts have the same answer. As the results have already been visualized in the main text, we do not repeat the results here.

The basic intuition is that, if the *queried-rule locating heads* indeed perform their functions as we described, then when we run the model on the clean prompts, patching in the *altered key activations* at these heads (within the Rules section) should cause "negative" change to the model's output, since it will cause these heads to mistake the queried-rule location in the altered prompt to be the right one, consequently storing the wrong rule information at the QUERY position. In particular, the model's logit difference between the two possible answers (to the LogOp chain and linear chain) should decrease. Indeed, that is what we observe.

### B.8.5 Analysis of fact-processing heads and decision heads

Similar to the Mistral-7B experiments, in this section, we aim to provide further validating evidence for the fact-processing heads and the decision heads. We experiment with flipping the truth value assignment for the OR chain while keeping everything else the same in the prompt (we always query for the OR chain in this experiment). As an example, we alter "Rules: A or B implies C. D implies E. Facts: **A is true. B is false.** D is true. Query: please state the truth value of C." to "Rules: A or B implies C. D implies E. Facts: **A is false. B is true.** D is true. Query: please state the truth value of C.". In this example, the answer is flipped from A to B. The (calibrated) intervened logit difference between the two variables in the OR chain is still a good choice in this experiment.

If the *fact-processing heads* indeed perform their function as described (selecting the correct fact to invoke, and moving such facts to the ":" position for next-word prediction), then patching the *altered key activations in the Facts section* of the problem's context would cause these attention heads to obtain a nontrivial intervened logit difference, i.e. it would help in bending the model's "belief" in what the facts are (especially the TRUE assignments in the facts section), thus pushing the model to flip its first answer token. This is indeed what we observe. In Figure 21, we see that only the key activations with index (23,6), (25,3), (26,6) and (28,4) (corresponding to heads (23,13), (25,6), (26,12) and (28,8)) obtain a higher score than every other key index, yielding evidence that they are sensitive to the truth values assignments. Interestingly, head (23,13) did not exhibit strong causal influence in our CMA experiment in the main text, thus was not included in the circuit. We suspect that this is due to its inconsistent ability in locating the correct fact: we find that when QUERY is for the linear chain, this head tends to be correct, and allocates a large amount of attention to the sentence of the correct fact, yet, when QUERY is for the OR chain, its performance is inconsistent.

Moreover, by patching the attention head output on and after the QUERY token (up to ":"), we gain understanding of which heads output important truth-value-sensitive information for the model's inference. We find that the fact-processing heads (25,7), (26,12), (28,12), and (30,9) indeed have large intervened logit differences. Interestingly, head (27,15) also has a large score: examining its attention pattern, we find that it resembles a mover head, focusing attention on "Answer" and ":" positions. We suspect that this head is moving truth-value-sensitive information sent out by the shallower layers (e.g. from the heads (25,7), (26,12)). Due to resource constraints, we were not able to verify this hypthesis.

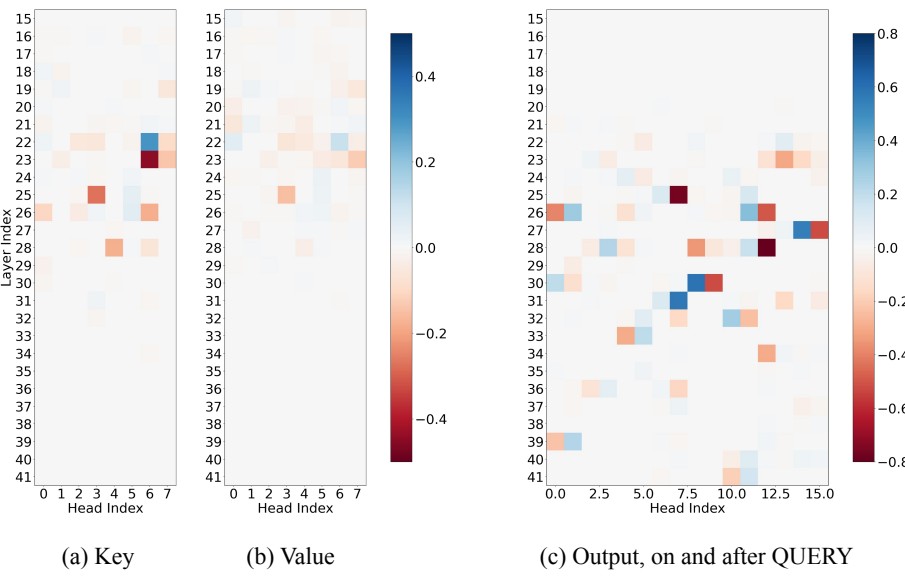

| (a) Key | (b) Value | (c) Output, on and after QUERY |

Figure 21: Gemma 2 27B. Key, value and output intervention experiment results. We note that the key and value activations are intervened in the Facts section, while the output intervention is done on token positions on and after QUERY. The former two helps us understand which attention heads are truth-value-sensitive and *rely on* truth value information for inference. The output intervention helps us know which attention heads' output are truth-value-sensitive, and *send out* truth-value-assignment-dependent information for the model's reasoning actions.

### B.8.6 Rule invocation: circuit reuse in proof writing

We localize the attention heads which invoke the correct rule for the argument. In particular, we still generate the counterfactual prompt by flipping the query token: clearly, this not only flips the first answer token (the correct fact invocation), but also the "first rule token". Consider the normal-counterfactual pairs:

| *[... in-context examples ...]* 
 **Rules**: $A$ or $B$ implies $C$. $D$ implies $E$. 
 **Facts**: $A$ is true. $B$ is false. $D$ is true. 
 **Question**: what is the truth value of $C$? 
 **Answer**: A is true. | $\xrightarrow[\text{prompt}]{\text{altered}}$ | *[... in-context examples ...]* 
 **Rules**: $A$ or $B$ implies $C$. $D$ implies $E$. 
 **Facts**: $A$ is true. $B$ is false. $D$ is true. 
 **Question**: what is the truth value of $E$? 
 **Answer**: D is true. |

What follows would be the correct rule invocation. For the normal prompt, it would be "**A** or B implies C", and for the counterfactual prompt, it would be "**D** implies E". We perform patching again at the first answer token position to localize components responsible for retrieving the correct rule to invoke; we show the results in Figure 22. We intervene on the token positions starting at QUERY, and ending at the final token position ".". Note that we perform patching accounting for GQA for Mistral-7B — we know from before that this coarser-grained patching actually helps us locate more attention heads which perform the role of rule-locating (recall Figure 8). As we see, the models reuse the rule-locator and decision heads.

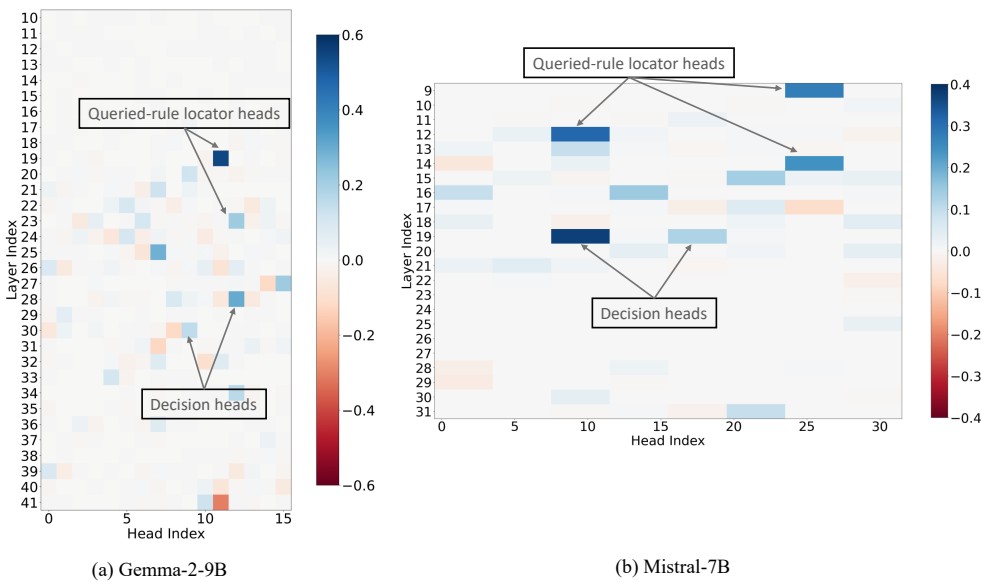

|  |  |
|---|---|
| (a) Gemma-2-9B | (b) Mistral-7B |

Figure 22: We observe subcircuit reuse when the Gemma-2-9B and Mistral-7B models invoke the correct rule.

*Remark.* With finer examination of the rule-locator heads in Gemma-2-9B, we found that the rule-locator heads' activations causally implicate the answer at both the QUERY token position and the first token of fact invocation (right after "Answer:"): indeed, both positions store information about the queried rule (reflected in their attention patterns). The decision heads, on the other hand, causally implicate the answer primarily at the last token position, right before invoking the rule — their precise functionality is more similar to that for fact invocation.

### B.8.7 Prompt format ablation

In this subsection, we aim to understand how robust our discovered subcircuits (the 4 families of heads) are to prompt format changes. In particular, we switch the Facts and Rules section of our problems. Our problem pairs now have the following form:

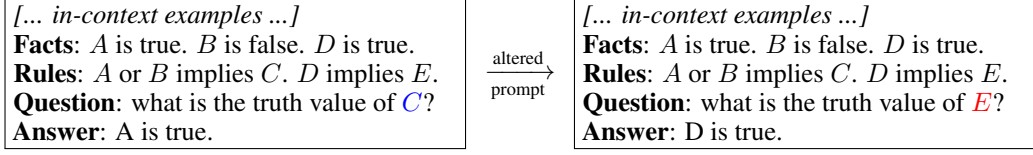

We show the CMA results, along with representative attention patterns of the surfaced attention heads. Please refer to Figure 23 for detailed visualization and explanations of our observations.

### B.9 Gemma-2-27B reasoning circuit

In this section, we analyze Gemma-2-27B's reasoning circuit. We first show the CMA result below, again relying on QUERY-based activation patching.

Additionally, we caution the reader that the experimental study in this subsection is less exhaustive in nature compared to our study of Mistral-7B and Gemma-2-9B, due to limitations in our computation budget.

### B.9.1 Attention head analysis

Most of the attention heads' attention patterns in this model's reasoning circuit (for writing down the first answer token) resemble those in Gemma-2-9B, therefore we omit another set of visualizations

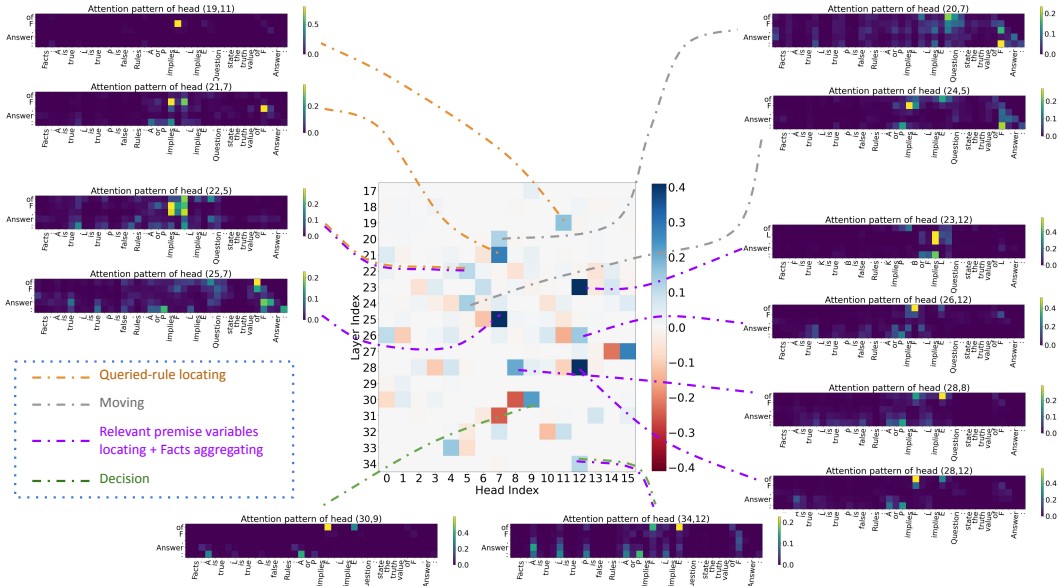

Figure 23: (Best viewed when zoomed in) CMA results on logic problems presented in the format "Facts: ... Rules: ... Question: ...", along with representative attention patterns of the discovered attention heads. Overall, the attention heads which have strong IEs remain almost the same as before. However, some of their functional roles changed. We elaborate on what stayed the same and what changed below.

First, the majority of the queried-rule locating and decision heads remain the same as before (with head (34,12) being a new hybrid decision head + fact-processing head which was not present in our circuit before). In other words, the "starting" and "ending" sub-circuits in the model remain virtually unchanged. However, the "middle-layer heads", namely the fact-processing and mover heads experienced some changes. In particular, a significant portion of these heads shifted to locating the relevant premise variables and aggregating relevant facts, instead of only performing moving actions or only focusing on the correct fact-sentence.

showing almost identical results to the 9B subsection from before (besides different layer-head indices). We focus on attention heads which exhibit *significantly new behavior* which had not been observed in the smaller models. This includes two intriguing points mentioned in the main text, (1) analyzing the logical-operator heads, and (2) showing that a portion of the fact-processing heads exhibit strong direct effect on the model's logits.

**Logical-operator heads**. We discuss how we localize these attention heads, and then show examples of their attention patterns: we find these heads' attention pattern not so easy to capture by simple bar plots showing summarized statistics.

First, to localize these attention heads, we perform the following set of experiments. We set the following restrictions to the problem context:

- We always query for the logical-operator (LogOp) chain
- We always set the truth value assignments for the two premise variables of the LogOp chain to be ["true", "false"]. We do not allow ["true", "true"] or ["false", "false"] in this analysis. The reason is that for both the situations of LogOp=AND and LogOp=OR, there is a unique first answer token (i.e. a unique fact that should be invoked in the minimal proof).

Given a normal prompt such as "Rules: A **or** B implies C. D implies E. Facts: A is true. B is false. D is true. Query: please state the truth value of C.", we generate the counterfactual prompt as "Rules: A **and** B implies C. D implies E. Facts: A is true. B is false. D is true. Query: please state the truth value of C." This can also be done the other way around of course.

We then hold the attention head output on or after QUERY as the mediator, and search for attention heads with strong indirect effects. In this setting, we measure the logit different between the two

possible answer tokens, namely the two premise variables in the LogOp chain. We show the results in Figure 24.

While a significant portion of the attention heads with nontrivial indirect effects are mover and decision heads, we find that heads (23,13), (23,31) place significant attention not only on the rule being queried, but also specifically on the logical operator of the queried rule, whenever that rule has a logical operator. Figure 25 contrasts their behavior on samples querying for the LogOp chain versus linear chain.

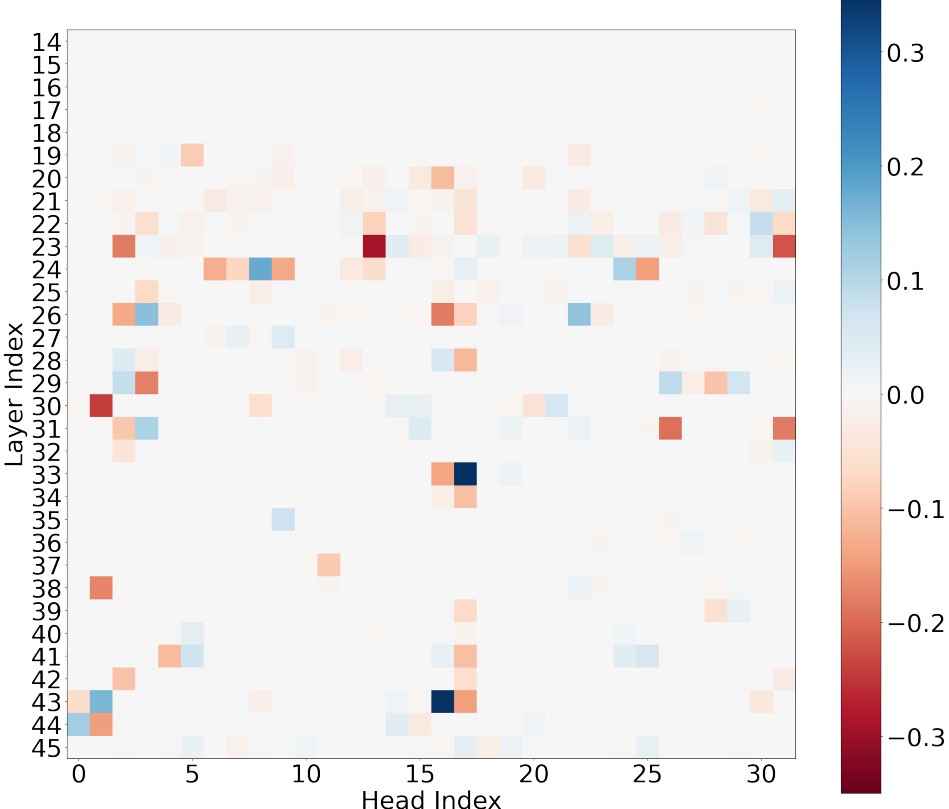

Figure 24: Gemma 2 27B. Output intervention experiment results of the logical-operator-flipping experiment. We note that the outputs are intervened on and after QUERY.

**Mechanical differences in Gemma-2-9B and Gemma-2-27B: Fact→Decision**. We illustrate in the Figure 26.

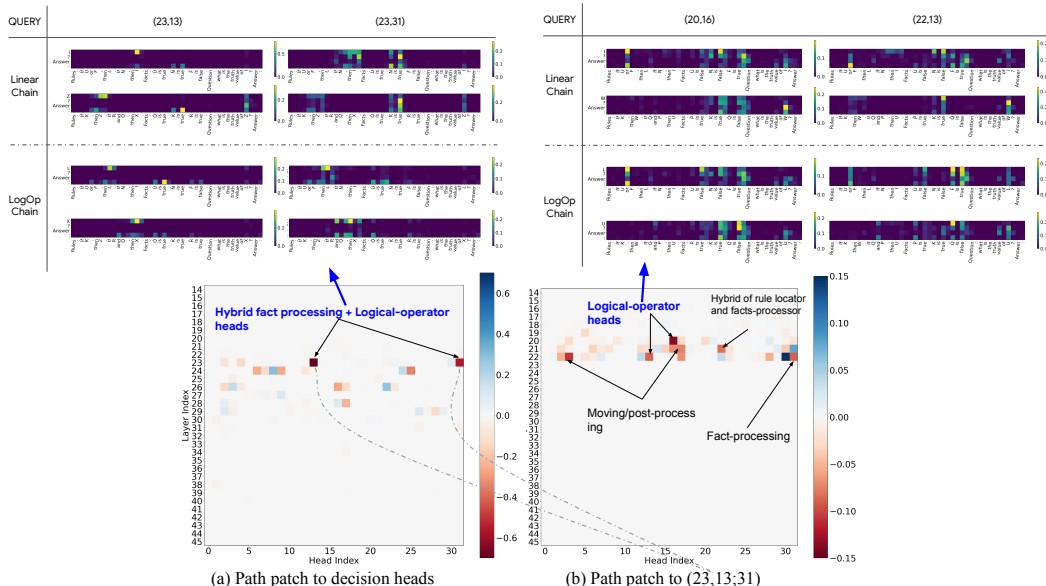

(a) Path patch to decision heads          (b) Path patch to (23,13;31)

Figure 25: Please zoom in to view the detailed path patching results and attention patterns. (a) shows the query-based path patching result to the decision heads of Gemma-2-27B, and (b) shows the query-based path patching results to the heads (23,13;31), which we found to place attention not only on the right fact(s), but also the logical operator word "and" or "or". Furthermore, we show the typical attention patterns of the logical-operator related heads on top. The path patching results are done by flipping the logical operator, as detailed in text.

The two attention heads (23,13) and (23,31) are sensitive to the change in the logical operator, and exhibit the strongest direct effects on the decision heads. For the four rows showing the example attention patterns of these two heads, the top two rows are the attention patterns of the attention heads obtained on samples querying for the linear chain, the bottom two rows are from samples querying for the LogOp chain. Each row is obtained from the same sample. Observe that in the bottom two rows, the two heads consistently place attention on the word "and" and "or" in addition to placing attention on the queried rule.

We also show typical attention patterns of (20,16) and (22,13), which have strong direct effects on (23,13;31) and also place significant attention on the logical operator. Moreover, note that they are not very precise fact-processing heads, even though they do place attention on certain facts: they do not place attention on the correct fact consistently. We hypothesize that these heads' role has more to do with informing later attention heads (particularly (23,13;31)) the logical operator present in this problem instance.

## C  The reasoning circuit in a small transformer

In this section, we study how small GPT-2-like transformers, trained solely on the logic problem, approach and solve it. While there are many parts of the answer of the transformer which can lead to interesting observations, in this work, we primarily focus on the following questions:

1. How does the transformer mentally process the context and plan its answer before writing down any token? In particular, *how does it use its "mental notes" to predict the crucial first token*?

2. How does the transformer determine the truth value of the query at the end?

We pay particular attention to the first question, because as noted in §2, *the first answer token reveals the most about how the transformer mentally processes all the context information without any access to chain of thought (CoT)*. We delay the less interesting answer of question 2 to the Appendix due to space limitations.

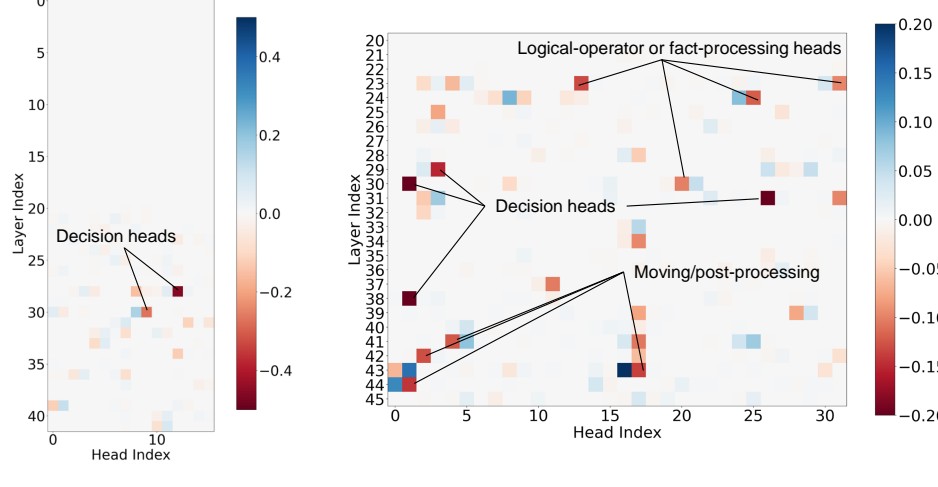

(a) Gemma-2-9B, direct effects on logits                    (b) Gemma-2-27B, direct effects on logits

Figure 26: Direct effects of attention heads on the model logits for Gemma-2-9B and Gemma-2-27B. The latter appears to have more parallel processing components than the 9B model. More specifically, for the 9B model, only the decision heads exhibit strong direct effects on the logits, the earlier ones (including fact-processing heads) do *not*. In contrast, the 27B has logical-operator and fact-processing heads exhibiting strong indirect effects on the logits, in addition to actual decision heads. Rather interestingly, if we recall the results in the previous Figure 25, we know that the logical-operator and fact-processing heads also have direct effects on the decision heads. This tells us that these intermediate heads are "connected" to both the decision heads, and also have direct influence on the logits. This is not observed in the smaller 9B model!

## C.1 Learner: a decoder-only attention-only transformer

In this section, we study decoder-only attention-only transformers, closely resembling the form of GPT-2 (Radford et al., 2019a). We train these models exclusively on the synthetic logic problem. The LogOp chain is queried 80% of the time, while the linear chain is queried 20% of the time during training. Details of the model architecture are provided in Appendix D.2.

**Architecture choice for mechanistic analysis**. We select a 3-layer 3-head transformer to initiate our analysis since it is the smallest transformer that can achieve 100% *test* accuracy; we also show the accuracies of several candidate model sizes in Figure 28 in Appendix D for more evidence. Note that a model's answer on a problem is considered accurate only if every token in its answer matches that of the correct answer. Please refer to Appendix D.2 for an illustration of the model components.

## C.2 Mechanism analysis

The model approximately follows the strategy below to predict the first answer token:

1. (Linear vs. LogOp chain) At the QUERY position, the layer-2 attention block sends out a special "routing" signal to the layer-3 attention block, which informs the latter whether the chain being queried is the linear one or not. The third layer then acts accordingly.

2. (Linear chain queried) If QUERY is for the linear chain, the third attention block focuses almost 100% of its attention weights on the QUERY position, that is, it serves a simple "message passing" role: indeed, layer-2 residual stream at QUERY position already has the correct (and linearly decodable) answer in this case.

3. (LogOp chain queried) The third attention block serves a more complex purpose when the LogOp chain is queried. In particular, the first two layers construct a partial answer, followed by the third layer refining it to the correct one.

We illustrate the overall reasoning strategy and core evidence for it in Figure 30 in Appendix D.3.

### C.2.1 Linear or LogOp chain: routing signal at the QUERY position

The QUERY token is likely the most important token in the context for the model: it determines whether the linear chain is being queried, and significantly influences the behavior of the third attention block. The transformer makes use of this token in its answer in an intriguing way.

**Routing direction at QUERY**. There exists a "routing" direction $h_{route}$ present in the embedding generated by the layer-2 attention block, satisfying the following properties:

1. $\alpha_1(X)h_{route}$ is present in the embedding when the linear chain is queried, and $\alpha_2(X)h_{route}$ is present when the LogOp chain is queried, where the two $\alpha_i(X)$'s are sample dependent, and satisfy the property that $\alpha_1(X) > 0$, and $\alpha_2(X) < 0$.
2. The "sign" of the $h_{route}$ signal determines the "mode" which layer-3 attention operates in at the ANSWER position. When a sufficiently "positive" $h_{route}$ is present, layer-3 attention acts as if QUERY is for the linear chain by placing significant attention weight at the QUERY position. A sufficiently "negative" $h_{route}$ causes layer-3 to behave as if the input is the LogOp chain: the model focuses attention on the rules and fact sections, and in fact outputs the correct first token of the LogOp chain!

We discuss our empirical evidence below to support and elaborate on the above mechanism.

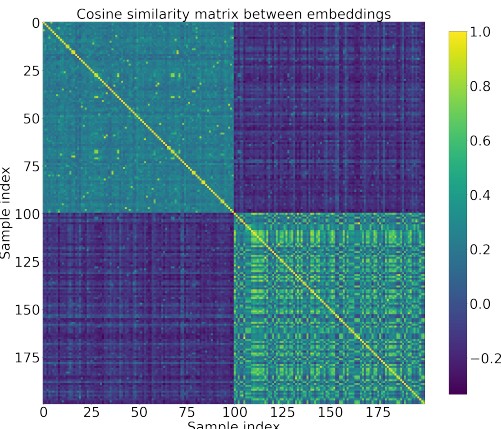

Figure 27: Small transformer (trained from scratch). Cosine similarity matrix between output embeddings from layer-2 attention block. Samples 0 to 99 query for the linear chain, samples 100 to 199 query for the LogOp chain. Observe the in-group clustering in angle (top left and bottom right), and the negative cross-group cosine similarity (top right and bottom left).

*Evidence 1a: chain-type disentanglement at QUERY.* We first observe that, at the QUERY position, the layer-2 attention block's output exhibits disentanglement in its output direction depending on whether the linear or LogOp chain is being queried, as illustrated in Figure 27.

To generate Figure 27, we constructed 200 samples, with the first half querying the linear chain and the second half querying the LogOp chain. We then extracted the layer-2 self-attention block output at the QUERY position for each sample, and calculated the pairwise cosine similarity between these outputs.

*Evidence 1b: distinct layer-3 attention behavior w.r.t. chain type.* When the linear chain is queried, the layer-3 attention heads predominantly focus on the QUERY position, with over 90% of their attention weights on the QUERY position on average (based on 1k test samples). In contrast, when the LogOp chain is queried, less than 5% of layer-3 attention is on the QUERY on average. Instead, attention shifts to the Rules and Facts sections of the context, as shown in Figure 31 in Appendix D.4.

Observations 1a and 1b suggest that given a chain type (linear or LogOp), certain direction(s) in the layer-2 embedding significantly influences the behavior of the third attention block in the

aforementioned manner. We confirm the existence and role of this special direction and reveal more intriguing details below.

*Evidence 1c: computing $\boldsymbol{h}_{route}$, and proving its role with interventions*. To *erase* the instance-dependent information, we *average* the output of the second attention block over 1k samples where QUERY is for the linear chain. We denote this *estimated* average as $\hat{\boldsymbol{h}}_{route}$ which effectively preserves the sample-invariant signal. To test the influence of $\hat{\boldsymbol{h}}_{route}$, we investigate its impact on the model's reasoning process, and we observe two intriguing properties:

1. (Linear→LogOp intervention) We generate 500 test samples where QUERY is for the linear chain. *Subtracting* the embedding $\hat{\boldsymbol{h}}_{route}$ from the second attention block's output causes the model to consistently predict the correct first token for the *LogOp chain* on the test samples. In other words, the "mode" in which the model reasons is flipped from "linear" to "LogOp".

2. (LogOp→linear intervention) We generate 500 test samples where QUERY is for the LogOp chain. *Adding* $\hat{\boldsymbol{h}}_{route}$ to the second attention block's output causes the three attention heads in layer 3 to focus on the QUERY position: greater than 95% of the attention weights are on this position averaged over the test samples. In this case, however, the model does not output the correct starting node for the linear chain on more than 90% of the test samples.

It follows that $\boldsymbol{h}_{route}$ indeed exists, and the "sign" of it determines the attention patterns in layer 3 (and the overall network's output!) in the aforementioned manner.

### C.2.2 Answer for the linear and LogOp chain

**Linear chain: answer at layer-2 residual stream at QUERY position**. At this point, it is clear to us that, *when QUERY is for the linear chain*, the third layer mainly serves a simple "message passing" role: it passes the information in the layer-2 residual stream at the QUERY position to the ANSWER position. One natural question arises: does the input to the third layer truly contain the information to determine the first token of the answer, namely the starting node of the linear chain? The answer is yes.

*Evidence 2: linearly-decodable linear-chain answer at layer 2*. We train an affine classifier with the same input as the third attention block at the QUERY position, with the target being the start of the linear chain; the training samples only query for the linear chain, and we generate 5k of them. We obtain a test accuracy above 97% for this classifier (on 5k test samples), confirming that layer 2 already has the answer linearly encoded at the QUERY position. To add further contrasting evidence, we train another linear classifier with exactly the same task as before, except it needs to predict the correct start of the *LogOp* chain. We find that the classifier achieves a low test accuracy of approximately 27%, and exhibits severe overfitting with the training accuracy around 94%.

**LogOp chain: partial answer in layers 1 & 2 + refinement in layer 3**. To predict the correct starting node of the LogOp chain, the model employs the following strategy:

1. The first two layers encode the LogOp and only a "partial answer". More specifically, we find evidence that (1) when the LogOp is an AND gate, layers 1 and 2 tend to pass the node(s) with FALSE assignment to layer 3, (2) when the LogOp is an OR gate, layers 1 and 2 tend to pass node(s) with TRUE assignment to layer 3.

2. The third layer, combining information of the two starting nodes of the LogOp chain, and the information in the layer-2 residual stream at the ANSWER position, output the correct answer.

We provide a full technical explanation of this high-level overview in Appendix D.4. Our argument mainly relies on linear probing and causal interventions at different layers and token positions in the model.

## D  Length-3 small transformer study: further technical details

In this section, we provide further technical details of our three-layer transformer experiments.

## D.1 Data definition and examples

As illustrated in Figure 1, the propositional logic problem always involve one logical-operator (LogOp) chain and one linear chain. In this paper, we study the length-3 case for the small-transformer setting, and length-2 case for the Mistral-7B-v0.1 case.

The input context has the following form:

```
RULES_START K implies D. V implies E. D or E implies A.
P implies T. T implies S. RULES_END
FACTS_START K TRUE. V FALSE. P TRUE. FACTS_END
QUERY_START A. QUERY_END
ANSWER
```

and the answer is written as

```
K TRUE. K implies D; D TRUE. D or E implies A; A TRUE.
```

In terms the the English-to-token mapping, `RULES_START`, `RULES_END`, `FACTS_START`, `FACTS_END`, `QUERY_START`, `QUERY_END ANSWER`, `.` and `;` are all unique single tokens. The logical operators `and` and `or` and the connective `implies` are unique single tokens. The proposition variables are also unique single tokens.

*Remark* D.1. The rules and facts are presented in a *random* order in the respective sections of the context in all of our experiments unless otherwise specified. This prevents the model from adopting position-based shortcuts in solving the problem.

Additionally, for more clarity, it is entirely possible to run into the scenario where the LogOp chain is queried, LogOp = OR and the two relevant facts both have FALSE truth values (or LogOp = AND and both relevant facts are TRUE), in which case the answer is not unique. For instance, if in the above example, both K and V are assigned FALSE, then both answers below are logically correct:

```
K FALSE V FALSE. K implies D; D UNDETERMINED. V implies E;
E UNDETERMINED. D or E implies A; A UNDETERMINED.
```

and

```
V FALSE K FALSE.  V implies E; E UNDETERMINED.
K implies D; D UNDETERMINED. D or E implies A; A UNDETERMINED.
```

**Problem specification**. In each logic problem instance, the proposition variables are randomly sampled from a pool of 80 variables (tokens). The truth values in the fact section are also randomly chosen. In the training set, the linear chain is queried 20% of the time; the LogOp chain is queried 80% of the time. We train every model on 2 million samples.

**Architecture choice**. Figure 28 indicates the reasoning accuracies of several candidate model variants. We observe that the 3-layer 3-head variant is the smallest model which achieves 100% accuracy. We found that 3-layer 2-head models, trained of some random seeds, do converge and obtain near 100% in accuracy (typically above 97%), however, they sometimes *fail* to converge. The 3-layer 3-head variants we trained (3 random seeds) all converged successfully.

## D.2 Small transformer characteristics, and training details

### D.2.1 Transformer definition

The architecture definition follows that of GPT-2 closely. We illustrate the main components of this model in Figure 29, and point out where the frequently used terms in the main text of our paper are in this model.

The following is the more technical definition of the model. Define input $\boldsymbol{x} = (x_1, x_2, ..., x_t) \in \mathbb{N}^t$, a sequence of tokens with length $t$. It is converted into a sequence of (trainable) token embeddings $\boldsymbol{X}_{token} = (\boldsymbol{e}(x_1), \boldsymbol{e}(x_2), ..., \boldsymbol{e}(x_t))^T \in \mathbb{R}^{t \times d_e}$, where we denote the hidden embedding dimension of the model with $d_e$. Adding to it the (trainable) positional embeddings $\boldsymbol{P} = (\boldsymbol{p}_1, \boldsymbol{p}_2, ..., \boldsymbol{p}_t)^T \in \mathbb{R}^{t \times d_e}$,

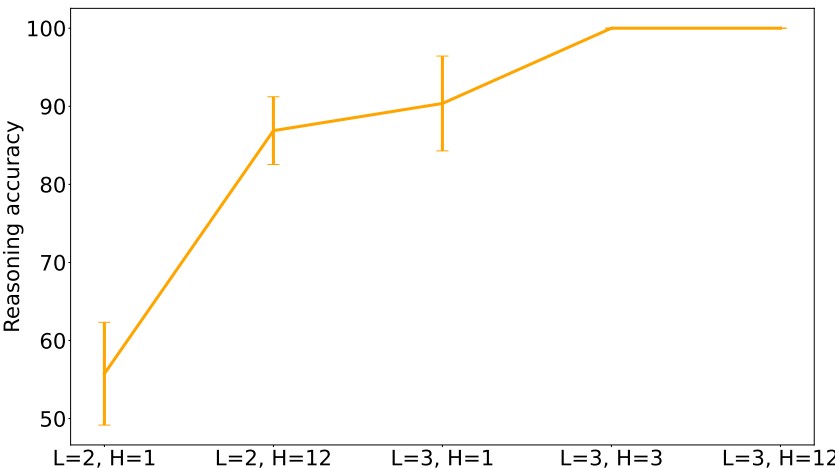

Figure 28: Small transformers (trained from scratch), varying sizes. Reasoning accuracies of several models on the length-3 problem. x-axis: model architecture (number of layers, number of heads); y-axis: reasoning accuracy. Note that the 3-layer 3-head variant is the smallest which obtains 100% accuracy on the logic problems.

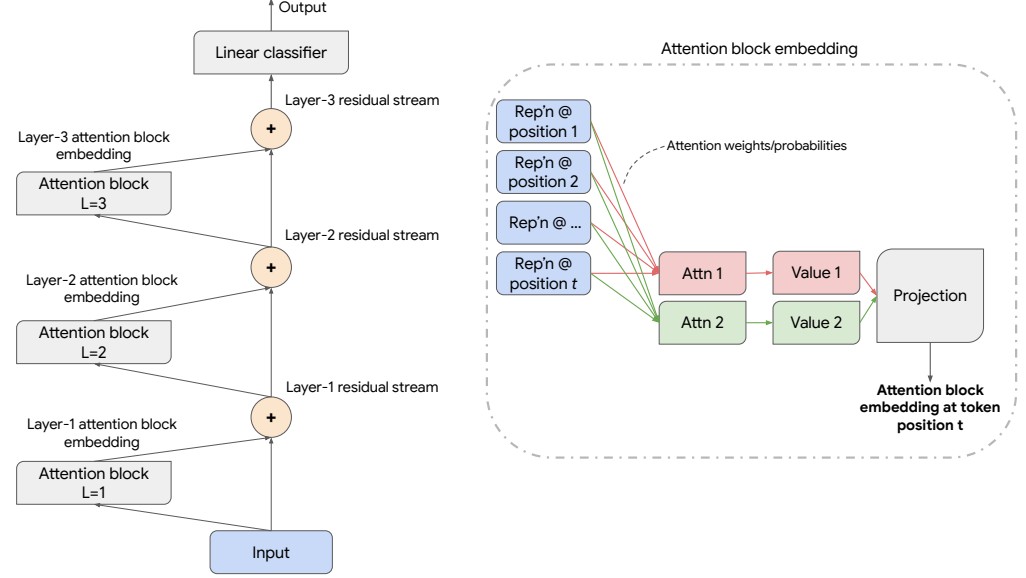

Figure 29: Illustration of the major components of a 3-layer attention-only decoder-only transformer on the left, and a rough "sketch" of what is computed inside an attention block (2 attention heads for simplicity of the sketch).

we form the zero-th layer embedding of the transformer

$$\boldsymbol{X}_0 = \boldsymbol{X}_{token} + \boldsymbol{P} = (\boldsymbol{e}(x_1) + \boldsymbol{p}_1, ..., \boldsymbol{e}(x_t) + \boldsymbol{p}_t). \tag{2}$$

This zero-th layer embedding is then processed by the attention blocks as follows.

Let the model have $L$ layers and $H$ heads. For layer index $\ell \in [L]$ and head index $j \in [H]$, attention head $\mathcal{A}_{\ell,j}$ is computed by

$$\mathcal{A}_{\ell,j}(\boldsymbol{X}_{\ell-1}) = \mathcal{S}\left(\text{causal}\left[\frac{1}{\sqrt{d_h}}\left(\boldsymbol{Q}_{\ell,j}\widetilde{\boldsymbol{X}}_{\ell-1}^T\right)^T \boldsymbol{K}_{\ell,j}\widetilde{\boldsymbol{X}}_{\ell-1}^T\right]\right)\widetilde{\boldsymbol{X}}_{\ell-1}\boldsymbol{V}_{\ell,j}^T \in \mathbb{R}^{t \times d_h}, \tag{3}$$

where $d_h = \frac{d_e}{H}$.

We explain how the individual components are computed below.

- Let us begin with how the $\mathcal{S}(...)$ term is computed.
- $\widetilde{\boldsymbol{X}}_{\ell-1} = \text{LayerNorm}(\boldsymbol{X}_{\ell-1}) \in \mathbb{R}^{t \times d_e}$, where LayerNorm denotes the layer normalization operator (Ba et al., 2016).
- $\boldsymbol{Q}_{\ell,j}, \boldsymbol{K}_{\ell,j} \in \mathbb{R}^{d_h \times d_e}$ are the key and query matrices of attention head $(\ell, j)$, where $d_h = \frac{d_e}{H}$. They are multiplied with the input $\widetilde{\boldsymbol{X}}_{\ell-1}$ to obtain the query and key activations $\boldsymbol{Q}_{\ell,j} \widetilde{\boldsymbol{X}}_{\ell-1}^T$ and $\boldsymbol{K}_{\ell,j} \widetilde{\boldsymbol{X}}_{\ell-1}$, both in the space $\mathbb{R}^{d_h \times t}$. We then perform the "scaled dot-product" of the query and key activations to obtain

$$\frac{1}{\sqrt{d_h}} \left( \boldsymbol{Q}_{\ell,j} \widetilde{\boldsymbol{X}}_{\ell-1}^T \right)^T \boldsymbol{K}_{\ell,j}, \widetilde{\boldsymbol{X}}_{\ell-1}^T \in \mathbb{R}^{t \times t}, \tag{4}$$

  which was introduced in Vaswani et al. (2017) and also used in GPT2 (Radford et al., 2019b).
- The causal mask operator causal : $\mathbb{R}^{t \times t} \to \mathbb{R}^{t \times t}$ allows the lower triangular portion of the input (including the diagonal entries) to pass through unchanged, and sets the upper triangular portion of the input to $-U$, where $U$ is a very large positive number (some papers simply denote this $-U$ as $-\infty$). In other words, given any $\boldsymbol{M} \in \mathbb{R}^{t \times t}$ and $(i, k) \in [t] \times [t]$,

$$[\text{causal}\,[\boldsymbol{M}]]_{i,k} = [\boldsymbol{M}]_{i,k} \text{, if } i \geq k;$$
$$[\text{causal}\,[\boldsymbol{M}]]_{i,k} = -U \text{, if } i < k. \tag{5}$$

- $\mathcal{S} : \mathbb{R}^{t \times t} \to \mathbb{R}^{t \times t}$ is the softmax operator, which computes the row-wise softmax output from the input square matrix. In particular, given a square input matrix $\boldsymbol{M} \in \mathbb{R}^{t \times t}$ with its upper triangular portion set to $-U$ (note that the causal mask operator indeed causes the input to $\mathcal{S}$ to have this property), we have

$$[\mathcal{S}(\boldsymbol{M})]_{i,k} = \frac{\exp\left([\boldsymbol{M}]_{i,k}\right)}{\sum_{n=1}^{i} \exp([\boldsymbol{M}]_{i,n})} \text{, if } i \geq k;$$
$$[\mathcal{S}(\boldsymbol{M})]_{i,k} = 0 \text{, if } i < k. \tag{6}$$

- To recap a bit, we have now explained how to compute the first major term in (3), namely $\mathcal{S}\left(\text{causal}\left[\frac{1}{\sqrt{d_h}} \left(\boldsymbol{Q}_{\ell,j} \widetilde{\boldsymbol{X}}_{\ell-1}^T\right)^T \boldsymbol{K}_{\ell,j} \widetilde{\boldsymbol{X}}_{\ell-1}^T\right]\right) \in [0,1]^{t \times t}$. It reflects the attention pattern (also called attention probabilities) of the attention head $(\ell, j)$ illustrated in Figure 29's right half. Intuitively speaking, the $(i, k)$ entry of this $t$ by $t$ matrix reflects how much the attention head moves the information from the previous layer $\ell - 1$ at the source token position of $k$ to the current layer $\ell$ at the target token position $i$.
- Now what about $\widetilde{\boldsymbol{X}}_{\ell-1} \boldsymbol{V}_{\ell,j}^T$? $\boldsymbol{V}_{\ell,j} \in \mathbb{R}^{d_h \times d_e}$ is the value matrix of attention head $(\ell, j)$. It is multiplied with $\widetilde{\boldsymbol{X}}_{\ell-1}$ to obtain the value activation $\widetilde{\boldsymbol{X}}_{\ell-1} \boldsymbol{V}_{\ell,j}^T \in \mathbb{R}^{t \times d_h}$.
- At this point, we have shown how the whole term in equation (3) is computed.

Having computed the output of the all $H$ attention heads in the attention block at layer $\ell$, we find the output of the attention block as follows:

$$\boldsymbol{X}_\ell = \boldsymbol{X}_{\ell-1} + \text{Concat}[\mathcal{A}_{\ell,1}(\boldsymbol{X}_{\ell-1}), ..., \mathcal{A}_{\ell,H}(\boldsymbol{X}_{\ell-1})]\boldsymbol{W}_{O,\ell}^T. \tag{7}$$

The operators are defined as follows:

- Concat$[\cdot]$ is the concatenation operator, where $\text{Concat}[\mathcal{A}_{\ell,1}(\boldsymbol{X}_{\ell-1}), ..., \mathcal{A}_{\ell,H}(\boldsymbol{X}_{\ell-1})] \in \mathbb{R}^{t \times d_e}$.
- $\boldsymbol{W}_{O,\ell} \in \mathbb{R}^{d_e \times d_e}$ is the projection matrix (sometimes called output matrix) of layer $\ell$. In our implementation, we allow this layer to have trainable bias terms too.

Finally, having computed, layer by layer, the hidden outputs $\boldsymbol{X}_{\ell,t}$ for $\ell \in [L]$, we apply an affine classifier (with softmax) to obtain the output of the model

$$\boldsymbol{f}(\boldsymbol{x}) = \mathcal{S}(\widetilde{\boldsymbol{X}}_{L,t} \boldsymbol{W}_{class}^T + \boldsymbol{b}_{class}) \tag{8}$$

This output indicates the probability vector of the next word.

In this paper, we set the dimension of the hidden embeddings $d_e = 768$.

### D.2.2 Training details

In all of our experiments, we set the learning rate to $5 \times 10^{-5}$, and weight decay to $10^{-4}$. We use a batch size of 512, and train the model for 60k iterations. We use the AdamW optimizer in PyTorch, with 5k iterations of linear warmup, followed by cosine annealing to a learning rate of 0. Each model is trained on a single V100 GPU; the full set of models take around 2 - 3 days to finish training.

### D.3 High-level reasoning strategy of the 3-layer transformer

We complement the text description of the reasoning strategy of the 3-layer transformer in the main text with Figure 30 below. It not only presents the main strategy of the model, but also summarizes the core evidence for specific parts of the strategy.

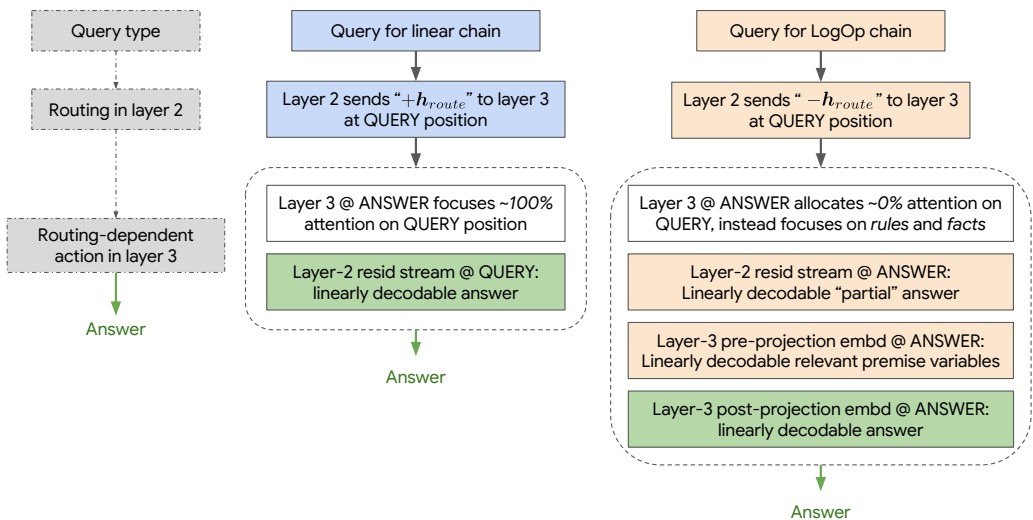

Figure 30: High-level overview of how the 3-layer transformer solves the logic problem, particularly in writing down the first answer token (the hardest place in the proof). As shown in the grey blocks on the left, the model performs "routing" in layer 2 by sending a routing signal $h_{route}$ to layer 3 (with its "sign" dependent on the query type), then the layer-3 attention block acts according to the "sign" of the routing signal sent to it. The middle (right) chain shows the strategy when the problem queries for linear (LogOp) chain.

### D.4 Answer for the LogOp Chain

*Evidence 3a: Distinct behaviors of affine predictors at different layers.* We train two affine classifiers at two positions inside the model (each with 10k samples): $W_{resid,\ell=2}$ at layer-2 residual stream, and $W_{attn,\ell=3}$ at layer-3 attention-block output, both at the position of ANSWER, with the target being the correct first token. In training, if there are two correct answers possible (e.g. OR gate, starting nodes are both TRUE or both FALSE), we randomly choose one as the target; in testing, we deem the top-1 prediction "correct" if it coincides with one of the answers. We observe the following predictor behavior on the test samples:

1. $W_{attn,\ell=3}$ predicts the correct answer 100% of the time.
2. $W_{resid,\ell=2}$ always predicts one of the variables assigned FALSE (in the fact section) if LogOp is the AND gate, and predicts one assigned TRUE if LogOp is the OR gate.

*Evidence 3b: linearly decodable LogOp information from first two layers.* We train an affine classifier at the layer-2 residual stream to predict the LogOp of the problem instance, over 5k samples (and tested on another 5k samples). The classifier achieves greater than 98% accuracy. We note that training this classifier at the layer-1 residual stream also yields above 95% accuracy.

*Evidence 3c: identification of LogOp-chain starting nodes at layer 3.* Attention heads (3,1) and (3,3), when concatenated, produce embeddings which we can linearly decode the two starting nodes of

the LogOp chain with test accuracy greater than 98%. We also find that they focus their attention in the rule section of the context (as shown in Figure 31). Due to causal attention, this means that they determine the two starting nodes from the LogOp-relevant rules. *Remark*. The above pieces of observations suggest the "partial information→refinement" process.[9] To further validate that the embedding from the first two layers are indeed causally linked to the correct answer at the third layer, we perform an activation patching experiment.

*Evidence 3d: linear non-decodability of linear chain's answer.* To provide further contrasting evidence for the linear decodability of the LopOp chain's answer, we experimentally show that it is not possible to linearly decode the answer of the *linear* chain in the model. Due to the causal nature of the reasoning problem (it is only possible to know the answer at or after the QUERY token position), and the causal nature of the decoder-only transformer, we train a set of linear classifiers on all token positions at or after the QUERY token and up to the ANSWER token, and on all layers of the residual stream of the transformer. We follow the same procedure as in Evidence 3c, except in this set of experiments, for contrasting evidence, QUERY is for the LopOp chain, while the classifier is trained to predict the answer of the Linear chain. The maximum test accuracy of the linear classifiers across all aforementioned token positions and layer indices is only 32.7%. Therefore, the answer of the Linear chain is not linearly encoded in the model when QUERY is for the LopOp chain.

*Evidence 3e: layer-2 residual stream at ANSWER is important to correct prediction.* We verify that layer-3 attention does rely on information in the layer-2 residual stream (at the ANSWER position):

- Construct two sets of samples $\mathcal{D}_1$ and $\mathcal{D}_2$, each of size 10k: for every sample $\boldsymbol{X}_{1,n} \in \mathcal{D}_1$ and $\boldsymbol{X}_{2,n} \in \mathcal{D}_2$, the context of the two samples are exactly the same, except the LogOp is flipped, i.e. if $\boldsymbol{X}_{1,n}$ has disjunction, then $\boldsymbol{X}_{2,n}$ has the conjunction operator. If layer 3 of the model has *no* reliance on the $\text{Resid}_{\ell=2}$ (layer-2 residual stream) for LogOp information at the ANSWER position, then when we run the model on any $\boldsymbol{X}_{2,n}$, patching $\text{Resid}_{\ell=2}(\boldsymbol{X}_{n,2})$ with $\text{Resid}_{\ell=2}(\boldsymbol{X}_{n,1})$ at ANSWER should *not* cause significant change to the model's accuracy of prediction. However, we observe the contrary: the accuracy of prediction degrades from 100% to 70.87%, with standard deviation 3.91% (repeated over 3 sets of experiments).

*Observation: LogOp-relevant reasoning at the third layer.* We show that the output from attention heads (3,1) and (3,3) (before the output/projection matrix of the layer-3 attention block), namely $\mathcal{A}_{3,1}(\boldsymbol{X}_2)$ and $\mathcal{A}_{3,3}(\boldsymbol{X}_2)$, when concatenated, contain linearly decodable information about the two starting nodes of the LogOp chain. We frame this as a multi-label classification problem, detailed as follows:

1. We generate 5k training samples and 5k test samples, each of whose QUERY is for the LogOp chain. For every sample, we record the *target* as a 80-dimension vector, with every entry set to 0 except for the two indices corresponding to the two proposition variables which are the starting nodes of the LogOp chain.

2. Instead of placing softmax on the final classifier of the transformer, we use the Sigmoid function. Moreover, instead of the Cross-Entropy loss, we use the Binary Cross-Entropy loss (namely the `torch.nn.functional.binary_cross_entropy_with_logits` in PyTorch, which directly includes the Sigmoid for numerical stability).

3. We train an affine classifier, with its input being the concatenated $\text{Concat}[\mathcal{A}_{3,1}(\boldsymbol{X}_2), \mathcal{A}_{3,3}(\boldsymbol{X}_2)]$ (a 512-dimensional vector) on every training sample, and with the targets and training loss defined above. We use a constant learning rate of $0.5 \times 10^{-3}$, and weight decay of $10^{-2}$. The optimizer is AdamW in PyTorch.

4. We assign a "correct" evaluation of the model on a test sample only if it correctly outputs the two target proposition variable as the top-2 entries in its logits. We observe that the classifier achieves greater than 98% once it converges.

---

[9]In fact, the observations suggest that layer 3 performs a certain "matching" operation. Take the OR gate as an example. Knowing which of the three starting nodes (for LogOp and linear chain) are TRUE, and which two nodes are the starting nodes for the LogOp chain are sufficient to determine the first token! This exact algorithm, however, is not fully validated by our evidence; we leave this as part of our future work.

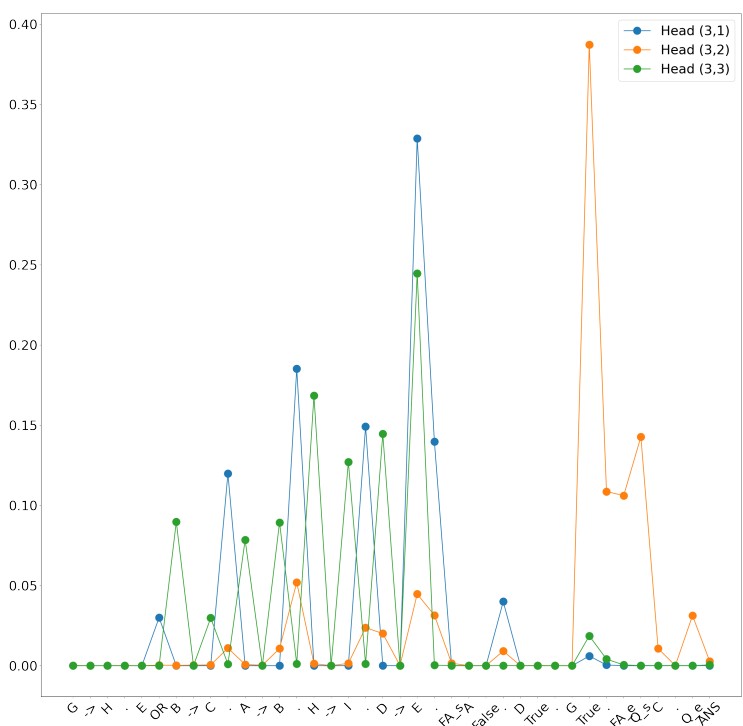

Figure 31: Attention statistics, averaged over 500 samples, all of which query for the LogOp chain. The x-axis is simply an example prompt that helps illustrate where the attention is really placed at. Observe that only attention head (3,2) pays significant attention to the fact section. The other two heads focus on the rule section. Note that none of them concentrate attention on the QUERY token. Reminder: due the the design of the problem, the rule, fact and query sections all have consistent length for every sample!

## D.5   Extra remarks

**Observation 3 supplement: linearly-decodable linear-chain answer at layer 2**. We simply frame the learning problem as a linear classification problem. The input vector of the classifier is the same as the input to the layer-3 self-attention block, equivalently the layer-2 residual-stream embedding. The output space is the set of proposition variables (80-dimensional vector). We train the classifier on 5k training samples (all whose QUERY is for the linear chain) using the AdamW optimizer, with learning rate set to $5 \times 10^{-3}$ and weight decay of $10^{-2}$. We verify that the trained classifier obtains an accuracy greater than 97% on an independently sampled test set of size 5k (all whose QUERY is for the linear chain too).

**Remarks on truth value determination**. Evidence suggests that determining the truth value of the simple propositional logic problem is easy for the model, as the truth value of the final answer is linearly decodable from layer-2 residual stream (with 100% test accuracy, trained on 10k samples) when we give the model the context+chain of thought right before the final truth value token. This is expected, as the main challenge of this logic problem is not about determining the query's truth value, but about the model spelling out the minimal proof with careful planning. When abundant CoT tokens are available, it is natural that the model knows the answer even in its second layer.

