# OpenReview forum: "A Implies B: Circuit Analysis in LLMs for Propositional Logical Reasoning"
_NeurIPS.cc/2025/Conference — NeurIPS 2025 spotlight_

### Official Review · Reviewer_zZhs · 2025-06-11

**Clarity:** 4
**Significance:** 3
**Originality:** 4
**Rating:** 5
**Confidence:** 5

**Summary:**

This paper utilizes a synthetic propositional logic problem set to explore the reasoning pathways and circuits of transformer-based LLMs with parameter sizes of up to 27 billion. By means of counterfactual variations, it identifies modular reasoning circuits within LLMs and the "lazy reasoning" phenomenon, in contrast to smaller transformers.

**Questions:**

1. The propositional logic problem studied is minimal (5 variables, 2 rules). How do the identified circuits scale to more complex reasoning tasks with longer logical chains or nested operators? Maybe you can test the models on extended problem variants with 3+ rules or nested logical operators (e.g., “A or (B and C) implies D”). Analyze whether the four attention head families (queried-rule locating, mover, fact-processing, decision) remain modular or require additional components. If the core circuit structure is preserved in more complex cases, this would strengthen claims about the universality of the identified mechanisms. A failure to generalize would indicate limitations in handling real-world multi-step reasoning.
2. The analysis focuses almost exclusively on attention heads, but what is the role of MLPs and lower-layer embeddings in the reasoning process? For example, do MLPs contribute to logical operator processing or fact integration? Performing ablation experiments by freezing MLP activations or lower-layer embeddings (e.g., layers 1–8 in Mistral-7B) and measure the impact on reasoning accuracy is welcomed. If MLPs are shown to play a non-trivial role (e.g., ≥20% accuracy drop when frozen), this would require revising the claim that reasoning is primarily localized in attention heads. Conversely, if their impact is minimal, it validates the focus on attention-based circuits.
3. The study is limited to Mistral and Gemma models. Do similar reasoning circuits exist in other LLM families like Qwen? Replicate the analysis on at least one additional model family (e.g., LLaMA or Qwen) to verify whether the four attention head families emerge universally. Compare attention patterns and circuit sparsity across architectures. If analogous circuits are found in diverse models, this supports the hypothesis that modular reasoning components are an emergent property of transformer pretraining. Discrepancies would highlight architecture-specific reasoning strategies.
4. The contrast with 3-layer task-trained models mentions “intermingled reasoning” but lacks detailed mechanistic comparison. How exactly do the information flow and component specialization differ between small, task-specific models and large, general-purpose LLMs? Train additional small transformers (e.g., 4-layer, 6-head) on the same task and analyze their attention patterns using the same CMA framework. Compare metrics like circuit sparsity, layer-wise information flow, and reliance on early vs. late layers. If larger small models show emerging modularity (e.g., separate fact-processing and decision layers), it would challenge the claim that modularity is exclusive to LLMs. If not, it would reinforce the role of scale in enabling specialized components.
5. Recommended citation: LogiCoT: Logical Chain-of-Thought Instruction-Tuning

**Ethical Concerns:**

["NO or VERY MINOR ethics concerns only"]

**Limitations:**

The paper does not discuss potential societal implications of its findings, such as how insights into LLM reasoning circuits could be used to improve safety, bias mitigation, or adversarial robustness. While the focus is foundational, addressing these aspects would strengthen the work’s relevance to broader ML ethics discussions.

**Quality:**

3

**Strengths And Weaknesses:**

Strengths:
The paper provides one of the first systematic mechanistic analyses of reasoning in large pretrained LLMs (up to 27B parameters).
The framework for circuit discovery (QUERY-based patching, CMA) offers a replicable methodology for analyzing reasoning in other LLMs. The emphasis on the first answer token as a critical latent reasoning point provides a focal point for future studies, as it captures the earliest stages of proof construction.
Cross-scale and cross-model comparisons reveal both conserved and scale-dependent mechanisms.

Weaknesses:
The propositional logic problem studied is minimalistic (5 variables, 2 rules), which may not generalize to real-world multi-step reasoning tasks with greater syntactic/semantic complexity.
The analysis focuses almost exclusively on attention heads, sidelining the role of MLPs and lower-layer embeddings. For instance, MLP-0 is dismissed as a “nonlinear token embedding,” but its potential role in feature transformation is underexplored. Additionally, the 27B model analysis is “partial” due to computational constraints, leaving open questions about how its more parallelized circuit impacts reasoning efficiency or error modes.
Unclear generalizability to other popular open models like Qwen.
The contrast with 3-layer models is valuable but underdeveloped. The small models are trained exclusively on the synthetic task, whereas LLMs are pretrained on diverse data, making direct mechanistic comparisons challenging. More analysis is needed to explain why task-specific training leads to intermingled reasoning, while general pretraining fosters modularity.

---

> ### Author Rebuttal · Authors · 2025-07-30
>
> Thank you for your valuable feedback, we are glad that you appreciate our methodology and mechanistic insights on the large and small models. Below are our responses to your comments and questions.
>
> > The propositional logic problem studied is minimal…
>
> While extending to longer problems is beyond the scope of this paper due to our constrained computational resources (longer context lengths coupled with the large models is too demanding for our computing resources), we did conduct two new experiments to understand how much the model reuses the identified subcircuits, under variations of our current setting.
>
> **_New experiment 1 - Prompt format variation_**. We conducted the following **new experiment for testing the robustness of the circuit across prompt formats**. We swapped the fact and rule sections in the prompt area. Now all the logic problems are presented in the form “Facts: … Rules: … Question:...”, for example:
>
> _Facts_: O is false. Q is true. I is true. _Rules_: I implies C. Q or O implies H. _Question_: what is the truth value of H? _Answer_: Q is true. Q or O implies H; H is true.
>
> We found evidence that on **Gemma-2-9B, the circuit for predicting the first answer token stays almost exactly the same**! In particular, we find that it is the same heads exhibiting high IE scores in the activation patching sweep, with slightly different scores. Due to the time limit, we have not performed the sufficiency test (as described in Section 3.2 and Appendix B.5, B.8.3) thoroughly yet, so we cannot report DE measurements here, but we think even the _necessity-based evidence_ is quite interesting to present here!
>
> **_New experiment 2 - Rule-invocation subcircuit reuse_**. We conduct a further analysis on the second most important place in the model’s generated proof, the **first token for invoking the rule** – recall that the model needs to invoke the correct fact, then the correct rule to infer the correct truth value of the queried proposition variable. We found that in both Gemma-2-9B and Mistral-7B, **the _queried-rule locators_ and _decision heads_ again exhibit strong indirect effects at this proof position, indicating _reuse_ of functional sub-circuits in their proof writing**.
>
> We will add the more detailed and rigorous version of these results to our updated paper.
>
> > “The contrast with 3-layer task-trained models mentions ‘intermingled reasoning’ but lacks detailed mechanistic comparison…”
>
> Thank you for this excellent point. Due to the page limit we had to delay our mechanistic analysis of the 3-layer 3-head transformer to _Appendix D_ (with its backbone algorithm summarized in _Figure 29_) – we do see some major differences between how the small models and the LLMs solve the simple propositional problems. We will add further comparisons to our updated paper. For now, we highlight some interesting differences below.
>
> The most pronounced difference is the _less interpretable and modular_ nature of the attention heads (especially their attention patterns) in the small transformer. Since the problem has many token positions which are “redundant” (e.g. “is”, the periods, spaces, etc.), the small transformer’s attention heads tend to make use of these positions in a much less predictable way than the large models.
>
> For example, while we show, through a set of probing and ablation experiments, that when the logical-operator chain is queried, it is only at the _final_ layer of the attention heads that the model arrives at the right token to write down – meaning that they are effectively “_decision heads_” – their attention pattern statistics are fairly messy, scattered across token positions in the Rules section. This stands in great contrast to the _modularity_ of the attention head families and the _interpretability_ of their attention patterns found in the LLMs. A cautionary remark should be emphasized, however: such modularity is _not_ necessarily observable in _all_ LLMs. For instance, smaller and older LLMs such as Gemma-2-2B and the GPT2 models have very low proof accuracies, they might not even possess “reasoning circuits” as we see in the Gemma-2-($\ge$9B) and Mistral-7B models.
>
> > Recommended citation: LogiCoT: Logical Chain-of-Thought Instruction-Tuning
>
> Thank you for this reference, we will add it to our updated paper.
>
> **Discussion on future directions**. We sincerely appreciate your comments on the possible ways of extending the current analysis. Increasing the complexity of the propositional problems by increasing its length and involving nested operators is indeed a feasible direction. However, even without increasing the problems’ lengths (with heavier compute demands), we believe that there are meaningful directions to pursue.
>
> First, the _negation_ operator is an interesting one to study: its analysis might indicate whether certain “polarized” behaviors arise in the model, and whether the model reuses certain causal pathways when it needs to latently negate a (simple) clause.
>
> First-order logic problems are another possible extension direction of problem setting. We are especially interested in _how transformers latently bind quantifiers to variables, and how such binding affects the way they reason about sets_ – a foundational problem in how LLMs reason.

---

### Official Review · Reviewer_rGZP · 2025-06-18

**Clarity:** 3
**Significance:** 3
**Originality:** 3
**Rating:** 5
**Confidence:** 4

**Summary:**

This paper presents a circuit-based analysis of how Gemma-2-9b, Gemma-2-27b and Mistral-7b perform (a constrained form of) single-step logical deduction. Each task consists of two rules (implications, with a binary disjunction or conjunction as the precondition), three facts (truth values of atoms) and one query (question about the truth value of one atom, which only depends on one of the two rules). The findings indicate that all of these models contain circuits which decompose the task as locating the relevant rule, identifying the query token, processing the relevant facts and then making a decision. In contrast, the authors find that training small transformers from scratch on these tasks does not yield such circuits, indicating that this behaviour arises at scale (of the model or the training data).

**Questions:**

Suggestions:
- Please include all **main** numerical results in the main text of the paper (in particular for section 3.2).

Questions:
- Can you elaborate a bit more on the significance of the decision head? It appears that it is attending sharply to the atom for which a known fact can be used in one of the rules to derive the conclusion. Calling this a "decision" seems a bit far fetched - it almost seems more like "fact locating" to me?
- What motivated your selection of models? In particular, did you consider studying a smaller model (e.g. 1b or 2b)? If it is feasible, comparing Gemma-2-2b to the 9b and 27b models would be particularly interesting (although it is very good that you also included a model belonging to a different family).

Criteria under which my evaluation score would increase or decrease: My ability to judge the significance and novelty of this work is somewhat limited as this is not my primary area of research. As such I have predominantly focused on the clarity of the exposition of the material, which overall is already good, as well as the apparent scientific rigor of the study. As such I will only consider updating my score if the discussion with other reviewers clarifies the novelty and significance of the work, or if I feel that the answers to my questions above have further strengthened the paper.

**Ethical Concerns:**

["NO or VERY MINOR ethics concerns only"]

**Final Justification:**

I believe this paper to be contributing novel findings to the growing body of knowledge regarding the internal workings of LLMs. Like other mechanistic interpretability works it definitely has its flaws, in particular regarding the scope of the analysis, and the novelty of the method itself is unclear. However, I do believe the findings to nonetheless have significant scientific value, especially when taking the additional experiments carried out during the rebuttal period into account. Unlike some other reviewers I do not take issue with the writing or presentation of the methodology.

Overall, I do not believe this paper to be groundbreaking in any way, but I do believe that it contributes new and meaningful knowledge to the literature and would therefore like to see it accepted.

**Limitations:**

yes

**Quality:**

4

**Strengths And Weaknesses:**

Strengths:
- The paper is very clear and well written. Both the motivations and the findings are presented clearly.
- The findings are interesting, especially due to the contrast between the LLMs and the small transformers trained from scratch.
- Despite the laborious nature of this type of research, the authors do manage to find generalizable findings across three models (although the Gemma-2-27b study is somewhat limited)
- While the problem is still very toy, it feels a bit more relevant to actual usages of LLMs than most prior work in this area (that I am aware of).

Weaknesses:
- Given the authors find some evidence that Gemma-2-27b carries out this task somewhat differently than Gemma-2-9b, it would have been nice to see Gemma-2-2b studied as well (to isolate the effect of scale vs. architecture).
- Focusing uniquely on Gemma-2-9b in the main text makes the narrative clear but makes it harder to get a sense of the generalizability of these findings. Focusing on one model in the text but still including the main numerical results for all models (e.g. Table 2 for Mistral-7b) in the main text would have made for a clearer paper.
- The influence of the decision head is not clear to me, which I believe is because of the focus on the model's state right before outputting the first token. However, I recognize that scaling up this analysis to arbitrary positions in the output sequence would be infeasible.
- The results figures (in particular Figure 5 and similar figures in the appendix) could use some polish. Furthermore, the legends do not clarify what the red bars are (95% CI on the mean? +- one standard deviation?).

---

> ### Author Rebuttal · Authors · 2025-07-30
>
> Thank you for your valuable feedback! We are glad that you find the paper’s writing clear, and appreciate the mechanistic insights on the large and small models. We will address your comments below.
>
> > “Given the authors find some evidence that Gemma-2-27b carries out this task somewhat differently than Gemma-2-9b, it would have been nice to see Gemma-2-2b studied as well (to isolate the effect of scale vs. architecture).”, “...did you consider studying a smaller model (e.g. 1b or 2b)?”
>
> We wish to emphasize that the main goal of this paper is to study _the latent processes behind how sufficiently capable LLMs write simple but valid (propositional logic) proofs_. We found that small LLMs (Gemma-2-2b, the GPT-2 models, etc.)’s proof accuracies are overly low even with a large number of in-context proof examples, showing that they do not appear to recognize the latent reasoning patterns. This is the primary reason for the study of larger LLMs in our paper, compared to the more standard sizes studied in many existing circuit analysis papers (mostly under 1B in size) [1-3].
>
> > Focusing uniquely on Gemma-2-9b in the main text makes the narrative clear but makes it harder to get a sense of the generalizability of these findings. Focusing on one model in the text but still including the main numerical results for all models (e.g. Table 2 for Mistral-7b) in the main text would have made for a clearer paper.
>
> Thank you for this point, we will add Table 2 to section 3.2 in the main text, and highlight that we were also able to verify a functionally similar circuit in Mistral-7B.
>
> > Can you elaborate a bit more on the significance of the decision head? It appears that it is attending sharply to the atom for which a known fact can be used in one of the rules to derive the conclusion. Calling this a "decision" seems a bit far fetched - it almost seems more like "fact locating" to me?
>
> There are two reasons why we group the decision heads instead of merging them into the family of fact-processing heads.
>
> First, we separate the two families based on their distinct attention patterns. We noted in Appendix B.4.2 and B.8.2 that the fact-processing heads’ attention weights usually do _not_ directly focus on the answer token itself, but are typically spread over the sentence containing the correct fact. In contrast, the decision heads always direct significant attention weight at the _precise answer token_, for all the LLMs we analyzed. Second, we found that the _decision_ heads tend to have _direct effects on the model logits_ in Gemma-2-9B (the main model analyzed in this paper), as we show in _Figure 25_ in the Appendix, while the fact-processing heads do not.
>
> We will surface succinct versions of the distinction between the two attention-head families to the main text in the updated version of our paper.
>
> > The influence of the decision head is not clear to me, which I believe is because of the focus on the model's state right before outputting the first token. However, I recognize that scaling up this analysis to arbitrary positions in the output sequence would be infeasible.
>
> We found that for **_rule invocation_ in the proof, the _decision heads_ again exhibit high importance in the Gemma-2-9B and Mistral-7B’s inference pathway**.
>
> More specifically, we performed another set of patching experiments measuring the causal importance of the model components for writing down the _first token of the invoked rule_ – the second most important place in the proof, after the fact-invocation token. We found that the same _decision heads_ and _rule-locator heads_ exhibit strong causal scores (measured via logit difference) in both Gemma-2-9B and Mistral-7B, with similar attention patterns to fact invocation! This suggests that the LLMs reuse functional sub-circuits for writing down the proof. We will add detailed versions of these results to our updated paper.
>
> > The results figures (in particular Figure 5 and similar figures in the appendix) could use some polish. Furthermore, the legends do not clarify what the red bars are (95% CI on the mean? +- one standard deviation?).
>
> Thank you for this point, we will make sure to add clarity to the figures in our updated paper.
>
> **References**.
>
> 1. K. Wang et al. Interpretability in the Wild: a Circuit for Indirect Object Identification in GPT-2 small. ICLR 2023.
> 2. G. Kim et al. A Mechanistic Interpretation of Syllogistic Reasoning in Auto-Regressive Language Models. ArXiv Preprint, 2025.
> 3. Y. Yao et al. Knowledge Circuits in Pretrained Transformers. NeurIPS 2024.

---

> > ### Comment · Reviewer_rGZP · 2025-08-01
> > **Response to Authors' Rebuttal**
> >
> > I thank the authors for their extensive reply to my questions and concerns, as well as to those of my fellow reviewers.
> >
> > Having read the other reviews in detail, I believe it seems fair to summarize the most important critique as boiling down to (1) questions regarding the generalizability and truthfulness of the circuits; (2) the impact of this work.
> >
> > For the first part, I am happy to see that the authors have added two additional experiments, as highlighted most succinctly in the response to zZhs but as has also been mentioned throughout the rebuttals. These additional experiments do, in my opinion, give more credence to the circuit analysis. In particular, it seems unlikely that their results would have agreed with those in the original version had the circuits been cherry-picked. However, I am a little bit unsure what the authors mean when they say that the circuits stay "*almost* exactly the same". Authors, can you please make this statement for precise?
> >
> > As for (2), this paper reads to me like an example of good science. There is a hypothesis, the truth value of which is investigated thoroughly, with clear limitations and scoping. Certainly this is not a paper that will suddenly cause a great leap forward in the capabilities of LLMs, nor is it a paper that decisively once and for all answers the question of it can be that they can reason. This paper is nonetheless a contribution to our collective understanding of these models, and while it may be modest in size it is nonetheless a contribution which I deem valuable. (Relatedly, while I very much appreciated the thoughtful and elaborate review by b3jQ, whose feedback on the presentation of the details of the casual analysis will certainly strengthen the paper, I do not agree that the casual chain is asserted without evidence. Rather, as the authors suggest in their reply to b3jQ, I read this as being a hypothesis which the rest of the paper seeks to investigate, which to me seems a good way of structuring the paper.)
> >
> > I will leave my score unchanged for now. I eagerly await the responses of b3jQ and 9qMC, who gave the most negative scores, as I am curious to hear what they think of the new findings (and updates to the presentation).
> >
> > In the meantime, I encourage the authors to clarify exactly what the precise findings of their additional experiments were, as I suspect this is something that will affect other reviewers' responses as well.

---

> > > ### Author Response · Authors · 2025-08-04
> > >
> > > Thank you for your encouraging comments on our work and rebuttals!
> > >
> > > **Circuit changes**. We studied the circuits in Gemma-2-9B with the prompt-format change by *measuring the indirect-effects (IE) scores of the attention heads’ output*. This was considered a “necessity”-based search in our paper. Interestingly, the attention heads with high IE scores from before *maintain* their high scores in this search (we will discuss the few new ones below), and we observe the same sparsity: besides these heads with high IE scores, the rest are very close to 0.
> > >
> > > Therefore, we highlight here the few new heads surfaced in our search, and the few heads from before which exhibit different functionalities below. These discussions, along with more causal and correlational analyses dissecting the nuanced differences (given more time before the camera-ready version, if we get accepted), will be added to our paper revision.
> > >
> > > 1. Head (34,12), not present in our previous circuit, surfaced with nontrivial IE score, and appears to take on the hybrid role of fact processing and decision making. More specifically, *whenever the Logical-Operator (LogOp) chain is queried*, it places heavy attention on *both* the truth values of the relevant facts in the Facts section (i.e. truth values of the two input variables in the logical-operator chain), and the two input variables in the queried rule in the Rules section. Moreover, similar to the other decision heads, it has nontrivial *direct* effects on the model logits. Therefore, it seems to directly take the step from identifying the two relevant proposition variables in the LogOp chain to making a decision on which token to write down (i.e. which fact to invoke). This is an interesting contrast against the decision heads found before: their behavior stays the same as before, i.e. they still only focus heavy attention on the right token to write down.
> > > 2. Heads (23,12), (23,13), not included in our circuit before, surfaced with nontrivial scores. They appear to have fact-processing functionalities: they place heavy attention on both the relevant proposition variables in the queried rule and the sentences assigning truth values.
> > > 3. The previous “fact-processing” head (26,12) behaves closer to a decision head now, as it places heavy attention *only* on the correct answer token to write down.
> > >
> > > **Faithfulness versus generalizability**. We think it is important to make some clarifications here about the “cherry-picked circuit” comment here. The *sufficiency* experiments in *Section 3.2, Appendix B.5 and B.8.3* are exactly aimed at countering such critique. Basically, we showed that when running the model on the normal prompt:
> > > 1. If we freeze all heads’ activations to their counterfactual version, the model behaves exactly as if it is solving the counterfactual problem, in terms of the logit-difference metric. This is reflected by the -1.0 score under the $C_{null}$ entry in Table 1 and 2.
> > > 2. If we unfreeze only the circuit heads (still “ablating” the non-circuit heads), we recover 94% of the normal performance of Gemma-2-9B (and 98% of Mistral-7B), shown under the entry $C$ in Tables 1 and 2.
> > > 3. As a further demonstration of sub-circuit *necessity* and *minimality*, we showed that removing any one family of heads renders the performance recovery to become nearly trivial.
> > >
> > > In more technical terms, as [1] defined, **we verified the *faithfulness* of the circuit**. This advanced criterion is not tested for or met in a number of mechanistic interpretability works [2-4] (they only presented IE-based necessity evidence).
> > >
> > > We think it is important to separate circuit *faithfulness* from *generalizability*. The former is a criterion for assessing discovered circuits on a chosen problem, the latter is a major open problem in the field of mechanistic interpretability (requires collective effort beyond a single paper).
> > >
> > > We do not believe that the *full* circuit in our work will generalize perfectly to other substantially different problems. On the other hand, *we do see early evidence of the generalizability of the functional sub-circuits*: [1,2] also observed *locator and mover heads* on some simple language problems, [5] found certain “insight” heads which place attention on intermediate steps towards an answer (somewhat similar to our *fact-processing heads*) in some RAG scenarios, etc.
> > >
> > > **References**.
> > > 1. K. Wang et al. Interpretability in the Wild: a Circuit for Indirect Object Identification in GPT-2 small. ICLR 2023.
> > > 2. Y. Yang et al. Emergent Symbolic Mechanisms Support Abstract Reasoning in Large Language Models. ICML 2025.
> > > 3. A. Stolfo et al. A Mechanistic Interpretation of Arithmetic Reasoning in Language Models using Causal Mediation Analysis. EMNLP 2023.
> > > 4. B. Wang et al. Grokking of Implicit Reasoning in Transformers: A Mechanistic Journey to the Edge of Generalization. NeurIPS 2024.
> > > 5. X. Zhao et al. Understanding Synthetic Context Extension via Retrieval Heads. ArXiv Preprint, 2025.

---

> > > > ### Comment · Reviewer_rGZP · 2025-08-08
> > > >
> > > > I thank the authors for their extensive, detailed reply, and for a productive discussion during the rebuttal period.
> > > >
> > > > Having considered the discussion herein, I have decided to update the Quality (to 4) and Confidence (also up to 4) scores for my final review. This reflects both the fact that many details of the scope of the findings were clarified during the discussion as well as the additional experiments carried out, which have increased my belief in the veracity and relevance of the findings. My decision not to update my already high overall score of 5 - Accept comes down to the belief that the findings, while interesting and valuable to the community, are nonetheless not groundbreaking, and the methodology itself follows prior work closely.
> > > >
> > > > I do not expect to change my scores any further.

---

> > > > > ### Author Response · Authors · 2025-08-08
> > > > >
> > > > > Thank you very much for your positive assessment of our work. We really appreciate the exchange of ideas in this discussion period, and found it very helpful towards improving our work, particularly in increasing the breadth of the experiments, and in clearly delineating the scope and limitations of our methodology. We will incorporate the suggestions from you, and all the reviewers here in our final manuscript.

---

### Official Review · Reviewer_9qMC · 2025-06-29

**Clarity:** 3
**Significance:** 2
**Originality:** 3
**Rating:** 4
**Confidence:** 4

**Summary:**

This paper studies the mechanism of how pretrained LLMs perform propositional logical reasoning tasks. The authors constructed synthetic propositional logical reasoning tasks that contain facts, rules, and a query of the truth value of a Boolean variable as questions and contain the proof and the final answer as the response. By providing several in-context examples and investigating the first answer token logit by causal mediation analysis (CMA), the authors identified a sparse circuit (which contains 13 attention heads for Gemma-2-9B) that performs propositional logical reasoning. The attention heads in the circuit can be divided into four groups according to their roles, i.e., queried-rule locating heads, queried-rule mover heads, fact processing heads, and decision heads. Analysis on different models (Mistral-7B and Gemma-2-27B models) also revealed similar components, although the larger model’s (Gemma-2-27B) circuits have a greater degree of parallelism.

**Questions:**

What is the longest reasoning chain studied in this problem (it seems that the illustrative examples in the main text can be solved by two-hop reasoning)

**Ethical Concerns:**

["NO or VERY MINOR ethics concerns only"]

**Final Justification:**

The authors' responses addressed most of my concerns.

For the first point, I agree with the authors that studying all output tokens might be computationally prohibitive (and can be a future direction), and their further analysis of the first token for invoking the correct rule is nice.

Therefore, I decided to raise my score to 4.

**Limitations:**

See the Weaknesses section.

**Paper Formatting Concerns:**

No major formatting issues.

**Quality:**

3

**Strengths And Weaknesses:**

**Strengths:**

1. The problem studied in this paper, i.e., how pretrained LLMs perform propositional logical reasoning tasks, is important, as propositional logical reasoning can be viewed as one of the most fundamental building blocks of more complex reasoning problems.

2. The authors carefully designed a synthetic task and identified the circuits performing the reasoning across different models, and found similar functional units (such as attention heads serving as queried-rule locating heads).

**Weaknesses:**

1. The analysis for the logit difference by CMA only involves the first answer token. Although predicting the first token is non-trivial and is important for generating the whole output sequence, analyzing the first token only might not be sufficient to understand the underlying mechanism of how LLMs perform propositional logical reasoning.

2. The input contains in-context examples. However, it is unclear how the in-context example will affect the underlying mechanism. For example, when there is no in-context example or when the number of in-context examples varies, will the circuit change?

3. The problem studied in this paper is somewhat restricted, and the format is not that flexible. To understand how LLMs perform logical reasoning in a more general setting and which heads play the important role, it would be important to analyze the scenarios when the input format changes while keeping the set of facts and rules unaltered (e.g., exchanging the order of the rule area and the fact area, or rules and facts are interleaved) or slightly increasing the complexity of the problem (as the illustrative examples in the main text is relatively simple and might not be sufficient to discover the full mechanism of how LLMs perform propositional logical reasoning).

4. It would be better to see some implications of the finding in this paper, such as how one can utilize the discovered circuits.

---

> ### Author Rebuttal · Authors · 2025-07-30
>
> Thank you for your thoughtful and constructive feedback! Below are our responses to your questions and suggestions.
>
> > The analysis for the logit difference by CMA only involves the first answer token. Although predicting the first token is non-trivial and is important for generating the whole output sequence, analyzing the first token only might not be sufficient to understand the underlying mechanism of how LLMs perform propositional logical reasoning.
>
>
> Thank you for this point. Indeed, it would be nice to study how the models write down all output tokens in the proof, but it is computationally infeasible across the large models we study given our resources.
>
> Nevertheless, **we conduct a further analysis** on the _second most important place_ in the model’s generated proof, the **first token for invoking the correct rule** – recall that the model needs to invoke the correct fact, then the correct rule to infer the correct truth value of the queried proposition variable. We found that in both Gemma-2-9B and Mistral-7B, **the _queried-rule locators_ and _decision heads_ again exhibit strong indirect effects at this proof position, indicating _reuse_ of functional sub-circuits in their proof writing**.
>
> We will add detailed versions of these results to our updated paper.
>
> > The problem studied in this paper is somewhat restricted, and the format is not that flexible. To understand how LLMs perform logical reasoning in a more general setting and which heads play the important role, it would be important to analyze the scenarios when the input format changes while keeping the set of facts and rules unaltered (e.g., exchanging the order of the rule area and the fact area, or rules and facts are interleaved)...
>
> Thank you for this nice suggestion!
>
> We conducted the following **new experiment for testing the robustness of the circuit across prompt formats**. As you suggest, we swapped the fact and rule sections in the prompts. Now all the logic problems are presented in the form “Facts: … Rules: … Question:...”. For example:
>
> _Facts_: O is false. Q is true. I is true. _Rules_: I implies C. Q or O implies H. _Question_: what is the truth value of H? _Answer_: Q is true. Q or O implies H; H is true.
>
> We found evidence that **on Gemma-2-9B, the circuit for predicting the first answer token stays almost exactly the same**! In particular, we find that it is the same heads exhibiting high IE scores in the activation patching sweep. Due to the time limit, we have not yet performed the sufficiency test (as described in Section 3.2 and Appendix B.5, B.8.3) thoroughly, so we cannot report DE measurements here, but we think the _necessity-based evidence_ is already quite interesting to present here! We will add the more detailed and rigorous version of these results to our updated paper.
>
> > The input contains in-context examples. However, it is unclear how the in-context example will affect the underlying mechanism. For example, when there is no in-context example or when the number of in-context examples varies, will the circuit change?
>
> Thank you for the comment – it is a good question as to if, and how much the circuit changes with respect to the number of in-context examples. It is, however, outside the scope of this paper.
>
> The primary goal of this paper is _not_ about understanding circuit _formation_ in in-context learning, but rather _how transformers output tokens that describe simple but valid mathematical proofs_: this requires us to work in a regime where the models are sufficiently high in accuracy, in which case the logit difference metric becomes a valid metric for measuring causal effects of model components. If the model is low in accuracy, then it cannot separate the two most likely candidate answer tokens well (the two which we use as normal and counterfactual answers by construction, in our QUERY-flipping experiments). Basically, we need to eliminate possible confounders introduced by having overly limited in-context proof examples, which would make the model too uncertain about the proof structure.

---

> > ### Comment · Reviewer_9qMC · 2025-08-05
> >
> > Thank the authors for the response and for addressing my concerns.
> >
> > For the first point, I agree that studying all output tokens might be computationally prohibitive (and can be a future direction), and the further analysis of the first token for invoking the correct rule is nice. I would like to see the new results in the revision.
> >
> > I will raise my score to 4.

---

> > > ### Author Response · Authors · 2025-08-05
> > >
> > > We are very grateful for your decision to raise the score for our submission! We appreciate your constructive feedback, and we will include the new mechanistic analysis, plus more nuanced dissections of the new results in our final manuscript. Please let us know if you have any further questions during the discussion period.

---

### Official Review · Reviewer_b3jQ · 2025-07-03

**Clarity:** 2
**Significance:** 2
**Originality:** 2
**Rating:** 4
**Confidence:** 3

**Summary:**

This work presents an in-depth causal mediation analysis (CMA) of how pretrained LLMs (Mistral, Gemma; up to 27B parameters) solve formal logic problems. The authors uncover fine-grained functional roles of attention heads by identifying sparse, modular circuits responsible for reasoning. They decompose the identified circuits into sub-circuits exhibiting four distinct sets of attention heads and validate these roles through various techniques (intervention, patching, ablation).

**Questions:**

- Why do exactly four attention head types emerge? Could this modularization differ under other prompts or tasks?

- Can the authors provide a minimal pseudo-code for discovering modular reasoning circuits applicable beyond the logic task?

- How statistically robust is the attention focus in Figure 4b? For decision head (30,9), attention seems diffuse across multiple tokens, yet the interpretation singles out “U”.

- In Table 1, how should we interpret negative logit differences (e.g., –0.97, –1.11)? Does the sign have causal meaning?

- Can the authors expand on this claim: IE magnitude ≠ component necessity? How does the method distinguish compensatory effects from causal roles?

- If decision heads track the correct fact variable when the model is correct, what happens when it is incorrect? How does attention shift, and what does it imply?

- In line 219 (Sec. 3.1), where are the MLPs defined?

**Ethical Concerns:**

["NO or VERY MINOR ethics concerns only"]

**Final Justification:**

My last comment to the authors is also my final justification, which I repeat here as below.

> I thank the authors for their detailed response, which has addressed most of my concerns regarding the CMA setup, experiments, and the rationale behind the four types of attention heads. The commitment to open-source the code meaningfully improves the replicability and potential broader adoption of this method (though the write-up could be more streamlined). I also appreciate the additional experiments, which partially strengthen the empirical grounding. A discussion, planned by the authors, on the token-distance effect (a potential confounder) and on the generalizability of the proposed circuit surgery to distinct reasoning domains would further reinforce the claims made in this work. Given these points, I decide to raise my score to 4 to recommend acceptance. Look forward to seeing the open-sourced tool tested on new domains, which could substantially advance our understanding of LLM circuit mechanisms.

**Limitations:**

Yes

**Quality:**

2

**Strengths And Weaknesses:**

### Pros:
 - this paper performs a fine-grained and relatively large-scale analysis of LLMs’ reasoning processes by examining symbolic logic tasks beyond single-hop inference. And it yields actionable insights into the internal modularity of large models.

 - It contributes to the growing body of mechanistic interpretability work, especially in understanding how attention heads are structured and composed for symbolic reasoning beyond one step.

### Cons:

- Causal mediation analysis methodology needs refinement:

  - the use of text-only explanations in Section 2 makes the methodology hard to follow. Critical elements such as circuit construction, intervention steps, and component attribution should be formalized using math notation, diagrams, or pseudo-code. Without these, the method lacks generalizability and reproducibility.

 - several CMA assumptions may be violated or not addressed:

   - the analysis tracks only the first answer token, assuming it encapsulates all reasoning. This may ignore important contributions to subsequent steps or generated proofs.

   - the causal graph assumed (X → M → Y) does not reflect the dense interdependencies in a transformer; identifiability assumptions (e.g., no hidden confounders, functional invariance) are likely invalid.

   - also, unadjusted upstream activations may confound the estimation of indirect effects, leading to overestimated effect sizes.

   - prompt perturbations (e.g., flipping QUERY, shuffling rules) modify multiple features (position encodings, token distance), so changes in logits cannot cleanly be attributed to the modeled causal chain.

   - this paper treats interventions as Monte Carlo over prompts, rather than stochastic model outputs, which may not be inconsistent with Pearl-style DE/IE definitions over probabilities.

- Conceptual and methodological clarity:

  - the definition of “causal chain” (e.g., “QUERY → Relevant Rule → Relevant fact(s) → Decision”) is introduced (lines 118–122) without formal backing. Why must the model explicitly follow this chain? How is it operationalized in model activations?

  - the claim that “the model must execute the causal chain” to produce the first answer token is asserted without sufficient justification. Given that transformer-based reasoning is distributed and redundant, the claim may be overly rigid or oversimplified.

  - the rationale behind exactly four types of attention heads remains ambiguous. Is this structure emergent or imposed by the design of the few-shot examples? Could alternative task formulations or data lead to different modular decompositions?

- Insufficient rigor in key sections:

  - in section 3.1, MLP references (e.g., MLP-0) are introduced abruptly without clear definitions. Terms like logit difference, patching, and attention head role should be better defined with notation as mentioned for clarity.

  - interventions (e.g., altering QUERY tokens, flipping rules) are scattered across text. A structured summary table would improve clarity.

   - figure 3 lacks explanation of the color encoding scheme; fig 4(b) lacks axis labels and a legend. The ordering of figures is also inconsistent with narrative flow.

   - table 1 includes abbreviations that are undefined in the main text, making it unreadable without referring to the appendix. All terms describing sub-circuits need be introduced and explained concisely in the main paper.

  - the method for identifying the four attention head types is underspecified. How are the roles inferred from weight visualization? How are components assigned to categories?

- Concerns with experimental interpretations:

  - section 3.2: Removing non-circuit heads may leave behavior unchanged, but that doesn’t necessarily prove their non-involvement—redundant heads may still contribute causally.

  - section 3.3: Swapping rules affects token order and embeddings, introducing confounds that may account for changes in indirect effect, rather than proving lookup mechanisms.

  - some analysis appears cherry-picked. For instance, attention head (23,6) is cited as moving QUERY to “:”, but the evidence for this movement is unclear. Further, figure 4c mentions both (23,6) and (23,3); clarification is needed.

  - large indirect effect (IE) does not imply necessity. Fragile components may inflate IE, while robust but essential parts may show small IE due to downstream compensation. Similarly, large direct effect (DE) does not prove sufficiency due to non-additivity and redundant pathways.

- presentation issues:

   - the writing requires grammar and clarity checks in several places (e.g., lines 92, 148, 210).

  - several critical technical definitions are scattered or missing, leading to cognitive overhead for readers unfamiliar with causal analysis in LLMs.

---

> ### Author Rebuttal · Authors · 2025-07-30
>
> We sincerely thank you for the thoughtful questions and suggestions! We would be happy to continue the discussion on any of our responses below.
>
> # CMA experiments, methodology and interpretation.
>
> Before delving into the technical details, we wish to emphasize that the use of _activation patching_ for localizing and interpreting hidden mechanisms in LLMs is popular in mechanistic interpretability, and has been explicitly cast as instances of Pearl’s Causal Mediation Analysis (CMA) in seminal works [1-3]. In particular, it is typically acknowledged that _activation patching offers imperfect (and potentially biased) estimates for the indirect and direct effects_, due to the LLM’s complexity and limits in computational resources.
>
> With this overall consideration of CMA in interpretability research, we address your concerns below.
>
> > the causal graph assumed (X → M → Y) does not reflect the dense interdependencies in a transformer
>
> We employ the simplified causal graph $X \to M \to Y$ as a _local abstraction_ of the LLM: in our circuit analysis, we analyze one hypothesized mediator at a time, treating the rest of the network as background, essentially approximating the indirect effects of the mediator. _This approach is common in mechanistic interpretability._
>
> > identifiability assumptions (e.g., no hidden confounders, functional invariance) are likely invalid.
>
> While the listed identifiability assumptions are largely satisfied by activation patching, some subtleties exist.
>
> First, please note that we sidestep many identifiability conditions in classical mediation analysis since we can observe the model’s “counterfactual” outcomes. By design of activation patching, no _exposure‑induced mediator‑outcome confounder_ exists. To perform activation patching, we run the model on the normal prompt, and hold the mediator’s output to its corresponding counterfactual activations, that is, we perform a hard _intervention_ $do(M = m_{counterfactual})$. For any potential $C$ in the LLM with $X \to C \to Y$ and $X \to C \to M$, the intervention cuts the link $C \to M$, removing such confounders. _Exposure-outcome_ and _exposure-mediator confounding_ are also ruled out – the LLM’s forward pass is fully deterministic and “self-contained”, as both $Y$ and $M$ are only functions of $X$. Functional invariance can be shown similarly.
>
> The _positivity_ assumption is subtler. Earlier approaches in mechanistic interpretability corrupt a mediator in a “messy” fashion, such as noise injection [4]. In comparison, our counterfactual activations are generated with much care to minimize “off-manifold” effects, as we use activation from altered prompts which differ _minimally_ from the normal one. We emphasize, however, that a balance has to be struck between efficiency of reaching reasonable conclusions and how fine-grained the differences are.
>
> > unadjusted upstream activations may confound the estimation of indirect effects…
>
> The upstream activations of the mediator, including those in the lower layers and previous token positions, and parallel activations (e.g. from the attention heads in the same layer, if the mediator is an attention head) are identical with or without intervention. Only the mediator’s output is intervened.
>
> > “large indirect effect (IE) does not imply necessity…  large direct effect (DE) does not prove sufficiency…”, “...compensatory effects”
>
> To demonstrate “necessity” of (a family of) attention heads in the circuit, we show
> 1) large IE in the single-component patching sweeps, and
> 2) when we ablate all non-circuit heads, further ablating any one of the attention families from the circuit _causes performance-recovery to drop from >90% to nearly trivial_.
>
> This shows that the identified attention head families participate heavily in the LLM’s inference pathway. This is a (stronger) “necessity” result following [2]’s definition.
>
> We believe that the rest of the comment is less about “sufficiency”, but more about the circuit’s “**faithfulness**” and “**completeness**” – more advanced criteria introduced in [5]. As shown in Section 3.2 and B.5, the discovered circuit is _nearly sufficient in recovering the model’s performance_ (94% for Gemma-2-9B, 98% for Mistral-7B) – **our circuit is faithful**. We note that **even the faithfulness criterion is not tested for or met in many existing mechanistic interpretability works** [6,7].
>
> Now, **our circuit is not necessarily _complete_**. Backup heads, albeit typically weak and noisy, have been observed in some LLMs, and is considered an open problem in mechanistic interpretability. Please note that **the completeness criterion is rarely tested for in many mechanistic interpretability works**, due to its _heavy computational demand_ of having to iteratively ablate the discovered causal mechanisms to track compensating pathways – an aspect which is particularly pronounced in our work due to the large model sizes. We will add a discussion on these more advanced criteria to our updated paper.
>
> > “Prompt perturbations… modify multiple features (position encodings, token distance)...”
>
> We first clarify that the sets of patching experiments are independent of each other. For example, when we flip the QUERY token, we do not shuffle the rules: only the QUERY _value_ is changed, _no position encoding or other features vary in the prompt_.
>
> We treat the rule-shuffling and fact-changing patching experiments as _cross-checks_ for our hypotheses of the sub-circuits in the LLMs. For example, the rule-locators, whose attention weights consistently track the queried rule, have high causal scores in both the QUERY-flipping and rule-shuffling experiments, but trivial scores in the fact-changing experiments. While the correlational and causal evidence do not fully eliminate nuanced factors such as _token distances_, the proposed circuit is still the more plausible explanation of existing evidence.
>
> > In Table 1, how should we interpret negative logit differences?
>
> The scores in Table 1 are normalized. W.r.t. the logit-difference metric, a score of -1.0 shows that the model behaves as if it is solving the counterfactual problem, even though it is run on the normal prompt. It is attained with the _null circuit_, i.e. freezing all attention head activations to their counterfactual activations. A score of 1.0 is the other extreme: the model behaves as if it is naturally run on the normal prompts. 0.0 is the mid-point.
>
> # Writing and conceptual clarity.
>
> > Clarify “MLP references…”, “interventions…”, “figure 3…”, “fig 4(b)…”, “table 1…”
>
> Thank you for these suggestions, we will address them in the updated version of our paper!
>
> > “definition of ‘causal chain’ …”, “…asserted without sufficient justification”
>
> Thank you for raising this point about the QUERY → Decision chain. We intended it to be the (human-interpretable) _hypothesis_ of how the LLM implements its internal reasoning. We will adjust our phrasing in the updated version of our paper to ensure that this point is clear.
>
> > “...rationale behind exactly four types of attention heads remains ambiguous”, “ method for identifying the four attention head types is underspecified”
>
> We primarily distinguish the attention-head families based on their attention patterns. While we do provide detailed explanations of their characteristics in Appendix B.4.2, B.8.2 and B.9.1, we will aim to surface succinct versions of these analyses into the main text for better clarity in our updated paper.
>
> > … four attention head types… modularization differ under other prompts or tasks?
>
> We believe the exact types of attention heads for different tasks can change. However, functionality-wise, there have been observations about the usage of locator and mover heads in other mechanistic works, in GPT2 models [5]. An interesting future direction is to study what functional components persist across problems and models, and what emerges with model scale.
>
> > “How statistically robust … decision heads…”,
>
> We show their (robust) attention statistics in Fig. 5(b)(iv).
>
> > “moving QUERY to ‘:’... (23,3)”
>
> Due to grouped query attention, head (23,6) has the value index (23,3). We primarily classify the mover heads based on their QUERY-invariant attention patterns, trivial query and key casual scores, and high value causal scores.
>
> **Pseudocode**.
> We will add the pseudocode for our circuit analysis algorithm to our updated paper. The following is its backbone, in consideration of character limits.
> ```
> A or B implies C. D implies E. A is true. B is false. D is true. What is the truth value of C?
>
> X  = prompt with Query = C
>
> X* = prompt with Query = E   (counterfactual)
>
> (1) run model(X)    → logits_norm
>
> (2) run model(X*)   → logits_ctfl
>
> (3) run model(X) with head h set to activation from X* → logits_interv
>
> (4) Estimate indirect effects (expression (1) in the Appendix)
> ```
> We currently manually set the threshold for top-k components to form the circuit (threshold setting is a major open problem in mechanistic interpretability). For more complex propositional problems, we can vary the QUERY token to choose which sub-part of the reasoning chain we wish to probe for.
>
> # References.
>
> 1. J. Vig et al. Investigating Gender Bias in Language Models Using Causal Mediation Analysis. NeurIPS 2020.
> 2. F. Zhang et al. Towards Best Practices of Activation Patching in Language Models: Metrics and Methods. ICLR 2024.
> 3. A. Geiger et al. Causal Abstraction: A Theoretical Foundation for Mechanistic Interpretability. JMLR 2025.
> 4. K. Meng et al. Locating and Editing Factual Associations in GPT. NeurIPS 2022.
> 5. K. Wang et al. Interpretability in the Wild: a Circuit for Indirect Object Identification in GPT-2 small. ICLR 2023.
> 6. A. Stolfo et al. A Mechanistic Interpretation of Arithmetic Reasoning in Language Models using Causal Mediation Analysis. EMNLP 2023.
> 7. B. Wang et al. Grokking of Implicit Reasoning in Transformers: A Mechanistic Journey to the Edge of Generalization. NeurIPS 2024.

---

> > ### Author Response · Authors · 2025-08-05
> >
> > Dear Reviewer b3jQ,
> >
> > With the discussion period ending soon, we just wanted to gently follow up on our rebuttal. We would be grateful to know if it addressed your concerns, and are happy to clarify any remaining points.

---

> > ### Comment · Reviewer_b3jQ · 2025-08-06
> >
> > Thanks for the clarifications so far! I still have a few lingering questions about your CMA setup and experiments:
> >
> > The authors are only looking at the first answer token. I get that following every token might be too heavy (as Reviewer 9qMC noted), but please flag this as a limitation. Also, since you only test on propositional logic, I’m not sure how well these findings carry over to, for example, mathematical reasoning. This leads to the following concern.
> >
> > The author's circuit surgery idea is great, but the write-up feels a bit cluttered. Please consider distilling it as a reusable recipe (for other tasks) since it is not open-sourced, making it unclear how this analysis tool can be replicated for others to test it on new domains, or at least the authors need to consider laying out (in words) where it may work (and where it breaks) across different tasks and difficulties.
> >
> > Still, the concern about the rationale behind exactly four types of attention heads remains. Appendix B lists four kinds of attention heads, but I can’t find why it’s four. Is that something that naturally emerged, or did the prompt design force it? Without a clear rationale and (ideally) a test on a new domain, it is hard to conclude. I may miss it, and please point to where this is explained.
> >
> > When the authors shuffle the rules, token positions will also be changed, which can tweak attention scores. It’d be good to mention this token-distance effect as a limitation? Also, the authors say the intervention cuts the $C \rightarrow M$ causal link, but we still compute M from corrupted X. How does M get formed if that link’s severed? Please clear this up.
> >
> > as to the pseudo-code, it is clearly that the authors swaps mediator activations but keep upstream activations “unchanged.” wonder if this is a standard way of doing activation patching, I understand when we intervene on the mediator via prompt pertubations, the upstream activations for the pertubed prompt will differ from the upstream activations for the original prompt, do we also need to consider the side-effect of upstream activations shift too? A bit more clarity here would help.
> >
> > Overall, I value the insights here, just looking for clearer explanations of these choices and a more streamlined presentation so we can all apply this method elsewhere.

---

> > > ### Author Response · Authors · 2025-08-08
> > > **Reminder of discussion deadline**
> > >
> > > Dear Reviewer b3jQ,
> > >
> > > As the discussion period ends today, we would greatly appreciate it if you could let us know whether our responses have addressed your concerns. We are available for any further discussion, and hope you will consider our exchanges in your final evaluation of our paper.
> > >
> > > ​Thank you for your consideration.

---

> > > > ### Comment · Reviewer_b3jQ · 2025-08-09
> > > >
> > > > I thank the authors for their detailed response, which has addressed most of my concerns regarding the CMA setup, experiments, and the rationale behind the four types of attention heads. The commitment to open-source the code meaningfully improves the replicability and potential broader adoption of this method (though the write-up could be more streamlined). I also appreciate the additional experiments, which partially strengthen the empirical grounding. A discussion, planned by the authors, on the token-distance effect (a potential confounder) and on the generalizability of the proposed circuit surgery to distinct reasoning domains would further reinforce the claims made in this work. Given these points, I decide to raise my score to 4 to recommend acceptance. Look forward to seeing the open-sourced tool tested on new domains, which could substantially advance our understanding of LLM circuit mechanisms.

---

> ### Author Response · Authors · 2025-08-06
> **Response to Reviewer b3jQ, Part 1/2**
>
> Thank you for your reply, we really appreciate your careful examination of our methodology and the imperfections of activation patching! Below are our responses to your comments.
>
> > "When the authors shuffle the rules, token positions will also be changed, which can tweak attention scores. It’d be good to mention this token-distance effect as a limitation", "when we intervene on the mediator via prompt perturbations, the upstream activations for the perturbed prompt will differ from the upstream activations for the original prompt... "
>
> Thank you for emphasizing these points.
>
> In general, this concern goes back to the typically acknowledged imperfection, or caution of using activation patching, which is the _imperfect isolation of causal features_ due to _multiple_ differing features in the altered/counterfactual prompt from the normal prompt, as discussed in [1] and [2]. While our main QUERY-flipping experiments minimize this issue (which our rebuttal's discussions centered around, regarding the identifiability assumptions), the _rule-location shuffling_ experiment indeed introduces _token distance_ as a possible confounding factor, distinct from the focal factor of the experiment, the queried-rule location. More technically speaking, token distance is possibly an _exposure-mediator confounder_ for the rule-shuffling experiment.
>
> Nevertheless, we think it is well-justified to prioritize compute on the rule-shuffling experiments over finer-grained ones for cross-checking the rule locators, by noting that _we are already keeping all the major non-focal factors fixed across the normal-counterfactual pairs_, which includes QUERY value and position, fact values and positions, proposition variables' choices and relations, choice of logical operator, etc. Moreover, since our problem setting relies on prompts which are short, we do _not_ expect token/clause distance to influence the qualitative conclusions on the rule-locator heads in our setting – i.e. they only surface in the QUERY-flipping and rule-shuffling experiments, but get trivial scores in the fact-changing experiments.
>
> However, it is possible that when the problem becomes very long (e.g. large number of clauses), token/clause distance becomes a factor of importance. Therefore, we will flag this factor in our final manuscript for cautioning the design of the cross-checking experiments in _more general settings_.
>
> >  I get that following every token might be too heavy (as Reviewer 9qMC noted), but please flag this as a limitation.
>
> We will flag this limitation in our final manuscript.
>
> Just to ensure that we are on the same page, we did conduct two more experiments to increase the breadth of the experiments: one experiment finding that for writing down the first token of the invoked rule, the rule-locator and decision heads in Gemma-2-9B and Mistral-7B again exhibit strong IE scores; one experiment finding that if we swap the Rules and Facts sections in the prompts (i.e. present Facts before Rules, instead of the original Rules before Facts), the surfaced heads overlap significantly with the circuit found before. Please refer to our responses to Reviewers rGZP and 9qMC for further details (to avoid making this response too cluttered).
>
> >  the concern about the rationale behind exactly four types of attention heads remains… Is that something that naturally emerged, or did the prompt design force it?
>
> We primarily assign the attention head families based on their distinct attention patterns with detailed discussions in Appendix B.4.2, B.8.2 and B.9.1 (with finer-grained validations from the cross-checking CMA experiments). We are not making any assumption regarding what family of heads _must_ emerge, we are just classifying them based on our experimental observations.
>
> However, in general, we understand your concern regarding how correlated the full circuit is with the prompt distribution we test on, even after we add the results in our previous point. This is a typical and valid concern with circuit-analysis on LLMs: the heavy demand for compute and labor limits analysis to narrow prompt distributions (to be fair, our work already takes a solid step towards more realistic reasoning problem structures compared to prior works [4-6]).
>
> From our perspective, the full circuit in this work likely does _not_ generalize perfectly to substantially different problems – it is possible that finer classes of heads emerge on more complex logic problems. However, we do see _early evidence of the generalizability of the functional sub-circuits_ found in our work: [4-6] observed _locator and mover_ heads on certain simple language problems, [7] found certain “insight” heads which place attention on intermediate steps towards an answer (somewhat similar to our _fact-processing_ heads) in some RAG scenarios, etc.
>
> (Continued in the next part)

---

> ### Author Response · Authors · 2025-08-06
> **Response to Reviewer b3jQ, Part 2/2**
>
> > the authors say the intervention cuts the $C \rightarrow M$ causal link, but we still compute M from corrupted X. How does M get formed if that link’s severed?
>
> We were explaining how the hard intervention $do(M = m_{counterfactual})$ eliminates (exposure-induced) _mediator-outcome confounding_. For the "counterfactual version" of $M$'s output, it is still computed naturally on the counterfactual prompt $X^*$.
>
> > wonder if this is a standard way of doing activation patching
>
> Our activation patching implementation precisely follows the standard algorithm, as described in [1-3].
>
> > Please consider distilling it as a reusable recipe (for other tasks) since it is not open-sourced
>
> We will make sure to add the more complete version of the pseudocode to our paper's appendix.
>
> We plan to open-source our code upon paper acceptance, including a main demo IPython notebook. Our code is built on top of the TransformerLens library (a well-known library in mechanistic interpretability), so it should be easily accessible to other researchers.
>
> > I’m not sure how well these findings carry over to, for example, mathematical reasoning.
>
> It is an important future direction to extend our work to more general mathematical reasoning settings, starting with simple, but focused topics such as how LLMs process negation and quantifiers (towards _first-order logic_ problems). They are, however, beyond the scope of this work.
>
> **References**.
> 1. F. Zhang et al. Towards Best Practices of Activation Patching in Language Models: Metrics and Methods. ICLR 2024.
> 2. S. Heimersheim et al. How to use and interpret activation patching. ArXiv Preprint, 2024.
> 3. A. Geiger et al. Causal Abstraction: A Theoretical Foundation for Mechanistic Interpretability. JMLR 2025.
> 4. K. Wang et al. Interpretability in the Wild: a Circuit for Indirect Object Identification in GPT-2 small. ICLR 2023.
> 5. Y. Yang et al. Emergent Symbolic Mechanisms Support Abstract Reasoning in Large Language Models. ICML 2025.
> 6. G. Kim et al. A Mechanistic Interpretation of Syllogistic Reasoning in Auto-Regressive Language Models. ArXiv Preprint, 2025.
> 7. X. Zhao et al. Understanding Synthetic Context Extension via Retrieval Heads. ArXiv Preprint, 2025.

---

### Decision · Program_Chairs · 2025-09-17

**Decision:**

Accept (spotlight)

**Comment:**

The paper presents a mechanistic interpretability study of how large pretrained LLMs (Gemma-2-9B, Gemma-2-27B, and Mistral-7B) perform propositional logical reasoning. Using causal mediation analysis (CMA) and activation patching, the authors identify sparse modular circuits consisting of four functional families of attention heads (rule locators, rule movers, fact processors, decision heads). They show these circuits are faithful (recovering 94–98% of model performance), scale-dependent, and consistent across model families, contrasting them with small transformers trained from scratch that do not exhibit modular reasoning.

### Strengths of the paper:

- The reviewers (zZhs, rGZP, 9qMC) emphasize the novelty and significance of providing one of the first systematic mechanistic analyses of reasoning in large pretrained models, revealing both conserved and scale-dependent mechanisms.

- The work is technically sound and rigorous, with careful use of activation patching, sufficiency/necessity tests, and cross-model comparisons. The experiments are labor-intensive but yield meaningful and generalizable findings.

- The paper is clear and well written, situating itself within mechanistic interpretability and contributing a reusable framework for reasoning circuit analysis.

- The authors commit to open-sourcing the code, improving reproducibility and impact, and their rebuttal added further experiments confirming circuit robustness under prompt format variation and rule invocation.

### Weaknesses of the paper:

- The methodology relies primarily on attention heads, with limited exploration of the role of MLPs or embeddings.

- The reasoning tasks are restricted to short propositional logic problems, raising questions about generalization to more complex or real-world reasoning settings.

- The analysis of the 27B model is partial due to computational constraints.

- Some presentation aspects (e.g., pseudo-code, figure clarity, explanation of head categories) could be streamlined to reduce cognitive load for readers.

### Primary reasons for Accept (Spotlight)

The primary reasons for recommending Accept (Spotlight) are that the paper makes a principled, timely, and rigorous contribution to mechanistic interpretability. It advances our understanding of how LLMs implement symbolic reasoning by uncovering sparse, modular circuits with clear functional roles, verified through both necessity and sufficiency evidence. The cross-model comparisons and scale-dependent analyses provide robust insights of broad interest. While limited in scope to propositional logic, the findings establish a faithful methodology and framework that can be extended in future work. The paper’s novelty, methodological rigor, and clear empirical support make it a valuable addition to the NeurIPS community.

### Summary of the discussion and rebuttal

The authors provided detailed responses that successfully addressed key concerns. For R-b3jQ, they clarified methodological rigor by explaining how activation patching constitutes a valid causal intervention, addressed confounding concerns, and demonstrated circuit faithfulness (recovering 94–98% accuracy). They also agreed to streamline the presentation with pseudo-code and clearer definitions. For R-9qMC, they expanded experiments to show that functional sub-circuits (rule locators and decision heads) are reused when altering prompt formats and during rule invocation, alleviating concerns about overfitting to the first answer token. For R-rGZP, they provided additional causal importance experiments and clarified the functional role of decision heads, strengthening confidence in the interpretability of the circuits. Overall, the rebuttal and discussion period led to multiple upgraded scores, consensus among reviewers, and reinforced the contribution as a robust, reproducible, and high-impact study of reasoning circuits in LLMs.